# Towards Trustworthy Federated Learning with Untrusted Participants

Youssef Allouah [1]   Rachid Guerraoui [1]   John Stephan [1]

## Abstract

Resilience against malicious participants and data privacy are essential for trustworthy federated learning, yet achieving both with good utility typically requires the strong assumption of a trusted central server. This paper shows that a significantly weaker assumption suffices: each pair of participants shares a randomness seed unknown to others. In a setting where malicious participants may collude with an *untrusted* server, we propose CAFCOR, an algorithm that integrates robust gradient aggregation with correlated noise injection, using shared randomness between participants. We prove that CAFCOR achieves strong privacy-utility trade-offs, significantly outperforming local differential privacy (DP) methods, which do not make any trust assumption, while approaching central DP utility, where the server is fully trusted. Empirical results on standard benchmarks validate CAFCOR's practicality, showing that privacy and robustness can coexist in distributed systems without sacrificing utility or trusting the server.

## 1. Introduction

The increasing complexity of machine learning (ML) models and the vast amounts of data required for training have driven the widespread adoption of distributed ML (Dean et al., 2012; Kairouz et al., 2021). By distributing the computational workload across multiple machines, this approach enables scalable and efficient model training. In the standard *server-based* architecture, multiple workers collaboratively optimize a shared model under the coordination of a central server. This process is typically implemented using distributed stochastic gradient descent (DSGD) (Bertsekas & Tsitsiklis, 2015), where workers compute gradients on local mini-batches and transmit them to the server, which then aggregates the updates to refine the global model. DSGD is especially valuable when privacy constraints prohibit data sharing, such as in healthcare applications where institutions hold sensitive patient records (Kaissis et al., 2020).

While DSGD inherently reduces data exposure by keeping local datasets private, significant privacy risks remain. If the trained model is publicly accessible, it becomes vulnerable to membership inference (Shokri et al., 2016) and model inversion (Fredrikson et al., 2015; Hitaj et al., 2017; Melis et al., 2019) attacks. Moreover, a *curious* server can analyze gradients and intermediate model states during training to extract sensitive information about individual data points (Phong et al., 2017; Wang et al., 2019; Zhu et al., 2019). Besides, in practical distributed environments, it is common for some workers to behave unpredictably due to system failures, corrupted data, or adversarial interference. These workers, also known as *Byzantine* (Lamport et al., 1982), can arbitrarily manipulate their updates, compromising the integrity of the learning process (Baruch et al., 2019; Xie et al., 2020). In DSGD, even a few malicious workers can significantly degrade model performance by poisoning gradients or models (Feng et al., 2015). This vulnerability underscores the need for robust aggregation techniques to mitigate the influence of malicious workers.

**Threat models.** Differential privacy (DP) (Choudhury et al., 2019; Hu et al., 2020; Noble et al., 2022) and robustness (Yin et al., 2018; Karimireddy et al., 2021; Allouah et al., 2023b) have been extensively studied in isolation in distributed machine learning. However, their intersection, which is crucial for building trustworthy learning systems, remains underexplored. Recent work shows that combining local DP (LDP), which assumes no trust, with robustness incurs a fundamental utility cost (Allouah et al., 2023c). This reinforces earlier findings that ensuring privacy under LDP, even without malicious workers, is prohibitive in terms of utility (Duchi et al., 2013). In contrast, the central DP (CDP) model, where the server is fully trusted, enables significantly better utility than LDP, particularly as the number of workers increases (Liu et al., 2021; Hopkins et al., 2022). These findings highlight the trade-offs between trust, privacy, and robustness in distributed learning, underscoring the need for alternative approaches that mitigate these limitations.

[1]EPFL, Switzerland. Alphabetical order. Correspondence to: Youssef Allouah <youssef.allouah@epfl.ch>, John Stephan <john.stephan@epfl.ch>.

*Proceedings of the 42nd International Conference on Machine Learning*, Vancouver, Canada. PMLR 267, 2025. Copyright 2025 by the author(s).

## 1.1. Contributions

In this work, we analyze an intermediate threat model that extends the privacy guarantees of LDP while achieving utility comparable to CDP, which assumes a trusted server. This model, referred to as *secret-based local differential privacy* (SecLDP), was only studied in non-adversarial settings focused on privacy (Allouah et al., 2024) or without utility guarantees in the presence of malicious workers (Sabater et al., 2022). We present the first analysis of SecLDP in adversarial distributed learning, considering an untrusted server and malicious workers, who aim to disrupt the learning as well as to compromise the privacy of honest workers by colluding with the server. Our results establish SecLDP as a viable privacy model for trustworthy distributed learning, bridging the gap between the stringent trust assumptions of LDP and the stronger utility guarantees of CDP.

**Algorithm.** We introduce CAFCOR, a privacy-preserving and robust distributed learning algorithm under the SecLDP model. CAFCOR employs a correlated noise injection mechanism inspired by secret sharing (Shamir, 1979), leveraging workers' access to shared randomness. This is efficiently achieved through a one-time pairwise agreement on secret randomness seeds using standard public-key infrastructure, as commonly done in secure distributed systems (Bonawitz et al., 2017). However, the added noise can obscure honest contributions, reducing the effectiveness of existing aggregations in filtering out malicious workers. To address this, we design a novel robust aggregation method that efficiently mitigates the impact of dimensionality on the mean estimation error, relying only on an upper bound on the number of corrupt workers.

**Bridging the gap.** We provide theoretical guarantees on the privacy-utility trade-off of CAFCOR under SecLDP, assuming an untrusted server and colluding malicious workers. Our analysis shows that CAFCOR achieves near-CDP performance under limited corruption, significantly outperforming LDP-based methods. When shared randomness is unavailable, CAFCOR seamlessly reverts to the standard LDP model while maintaining state-of-the-art performance. Extensive experiments on benchmark datasets (MNIST and Fashion-MNIST) validate our findings, confirming CAFCOR's superior performance in adversarial settings.

## 1.2. Related Work

Most works combining DP and robustness focus on local privacy models, often only achieving per-step privacy guarantees and suboptimal learning performance (Guerraoui et al., 2021; Zhu & Ling, 2022; Xiang & Su, 2022; Ma et al., 2022). Others reveal a three-way trade-off between privacy, robustness, and utility for discrete distribution estimation in non-interactive settings (Cheu et al., 2021; Acharya et al., 2021) and more general interactive learning tasks (Al-louah et al., 2023c). In the CDP model where the server is fully trusted, the privacy-utility trade-off is far superior to the LDP setting (Duchi et al., 2013; 2018). Prior work on centralized robust mean estimation with DP (Liu et al., 2021; Hopkins et al., 2022; Alabi et al., 2023) achieves near-optimal error bounds, disentangling the three-way trade-off by matching the optimal sample complexity in both private and robust settings. Empirical studies confirm this advantage in standard federated learning tasks (Xie et al., 2023).

Intermediate threat models between local and central DP, such as those leveraging cryptographic primitives or anonymization via shuffling (Jayaraman et al., 2018; Erlingsson et al., 2019; Cheu et al., 2019), can approach the privacy-utility trade-off of central DP but often incur high computational costs, rely on trusted entities, or lack robustness. Correlated noise schemes have also been explored for privacy without a trusted server; early methods like secret sharing (Shamir, 1979) ensure input security but do not privatize the output, leaving it vulnerable to inference attacks (Melis et al., 2019). More refined approaches (Imtiaz et al., 2021) combine correlated and uncorrelated Gaussian noise to protect averages, while decentralized extensions remain limited to averaging tasks (Sabater et al., 2022) and lack robustness (Allouah et al., 2024). We also note that (Sabater et al., 2022) consider malicious workers but do not provide utility guarantees in their presence, and only use cryptographic checks, e.g., to verify boundedness of all messages. On the other hand, they consider sparse topologies which induces lower communication costs, unlike our quadratic communication complexity for seed exchange, which we recall is negligible since it happens once before training and only includes integers, compared to the large-scale models exchanged iteratively during training. Recently, Gu et al. (2023) considered malicious workers and an untrusted server but relied on computationally intensive tools like zero-knowledge proofs and secure aggregation, whereas our method avoids these complexities. Moreover, they do not quantify the utility cost of their solution.

Cryptographic primitives such as homomorphic encryption and secure multi-party computation have been widely used to enhance privacy in distributed learning (Bonawitz et al., 2017; Zhang et al., 2020), offering efficient implementations. Some works have explored integrating these techniques with robustness (Corrigan-Gibbs & Boneh, 2017; Choffrut et al., 2024), but they face limitations. While cryptographic schemes efficiently support basic operations like averaging, they often incur significant computations when coupled with robust aggregation due to the non-linearity of key operations such as median and nearest neighbors. This limits their practicality in robust distributed learning.

## 2. Problem Statement

We consider the classical server-based architecture comprising $n$ workers and a central server. The workers hold local datasets $\mathcal{D}_1, \ldots, \mathcal{D}_n$, each composed of $m$ data points from an input space $\mathcal{X}$, i.e., $\mathcal{D}_i \in \mathcal{X}^m, i \in [n]$. For a given parameter vector $\theta \in \mathbb{R}^d$, a data point $x \in \mathcal{X}$ has a real-valued loss function $\ell(\theta; x)$. The empirical loss function for each $i \in [n]$ is defined for all $\theta \in \mathbb{R}^d$ as $\mathcal{L}_i(\theta) := \frac{1}{m} \sum_{x \in \mathcal{D}_i} \ell(\theta; x)$. The goal of the server is to minimize the global empirical loss function defined for all $\theta \in \mathbb{R}^d$ as $\mathcal{L}(\theta) := \frac{1}{n} \sum_{i=1}^{n} \mathcal{L}_i(\theta)$. We assume that each loss $\mathcal{L}_i$ is differentiable, and that $\mathcal{L}$ is lower bounded.

### 2.1. Robustness

We consider a setting where at most $f$ out of $n$ workers may be malicious. Such workers may send arbitrary messages to the server, and need not follow the prescribed protocol. The identity of malicious workers is a priori unknown to the server. Let $\mathcal{H} \subseteq \{1, \ldots, n\}$, with $|\mathcal{H}| = n - f$. We define for all $\theta \in \mathbb{R}^d$, $\mathcal{L}_{\mathcal{H}}(\theta) := \frac{1}{|\mathcal{H}|} \sum_{i \in \mathcal{H}} \mathcal{L}_i(\theta)$. If $\mathcal{H}$ represents the indices of honest workers, $\mathcal{L}_{\mathcal{H}}$ is referred to as the global *honest loss*. An algorithm is robust if it enables the server to approximately minimize the global honest loss, as follows.

**Definition 2.1** $((f, \varrho)$-Robustness). *A distributed algorithm $\mathcal{A}$ is $(f, \varrho)$-robust if it outputs $\hat{\theta}$ such that $\mathbb{E}\left[\mathcal{L}_{\mathcal{H}}(\hat{\theta}) - \mathcal{L}_{\star}\right] \leq \varrho$, where $\mathcal{L}_{\star} := \inf_{\theta \in \mathbb{R}^d} \mathcal{L}_{\mathcal{H}}(\theta)$, and the expectation is over the randomness of the algorithm.*

In other words, Algorithm $\mathcal{A}$ is $(f, \varrho)$-robust if it outputs in every execution a $\varrho$-*approximate* minimizer of the global honest loss function $\mathcal{L}_{\mathcal{H}}$, despite $f$ malicious workers.

### 2.2. Differential Privacy

Each honest worker $i \in \mathcal{H}$ aims to protect the privacy of their dataset $\mathcal{D}_i$ against the *honest-but-curious* (or untrusted) server. That is, the server follows the prescribed protocol, but may attempt to violate the workers' data privacy. Also, we assume that every pair of workers $i, j \in [n]$ shares a sequence of *secrets* $S_{ij}$, which represent observations of random variables only known to the workers $i, j$. In practice, these are generated locally via shared randomness seeds exchanged after one round of encrypted communications (Bonawitz et al., 2017), and conceptually one can consider the secrets to be the shared randomness seeds only. We denote by $\mathcal{S} := \{S_{ij} : i, j \in [n]\}$ the set of all secrets.

We recall that local DP (LDP) (Kasiviswanathan et al., 2011) protects against an untrusted server without assuming the existence of secrets, at the price of a poor privacy-utility trade-off (Duchi et al., 2013). Instead, to achieve a better trade-off, we consider the following relaxation of LDP into *secret-based local differential privacy* (SecLDP), adapting

the formalism of Allouah et al. (2024), where two datasets are adjacent if they differ by one worker's local dataset.

**Definition 2.2** (SecLDP). *Let $\varepsilon \geq 0$, $\delta \in [0, 1]$. Consider a randomized distributed algorithm $\mathcal{A}_{\mathcal{S}} : \mathcal{X}^{m \times n} \to \mathcal{Y}$, which outputs the transcript of all communications, given the full set of secrets $\mathcal{S}$. Algorithm $\mathcal{A}_{\mathcal{S}}$ satisfies $(\varepsilon, \delta, \mathcal{S}_{known})$-SecLDP if it satisfies $(\varepsilon, \delta)$-DP given that the subset of secrets $\mathcal{S}_{known} \subseteq \mathcal{S}$ is revealed to the curious server. That is, for all adjacent datasets $\mathcal{D}, \mathcal{D}' \in \mathcal{X}^{m \times n}$,*

$$\mathbb{P}\left[\mathcal{A}_{\mathcal{S}}(\mathcal{D}) \mid \mathcal{S}_{known}\right] \leq e^{\varepsilon} \cdot \mathbb{P}\left[\mathcal{A}_{\mathcal{S}}(\mathcal{D}') \mid \mathcal{S}_{known}\right] + \delta.$$

*We simply say that $\mathcal{A}_{\mathcal{S}}$ satisfies $(\varepsilon, \delta)$-SecLDP when $\mathcal{S}_{known}$ is clear from the context.*

The above definition captures multiple levels of trust, always considering the server to be honest-but-curious. First, the least stringent setting is when no worker colludes with the server, meaning that no secrets are disclosed to the server ($\mathcal{S}_{known} = \varnothing$). Second, the most adversarial scenario arises when all workers collude with the server ($\mathcal{S}_{known} = \mathcal{S}$), which corresponds to the standard LDP threat model. Finally, we consider an intermediate scenario where only malicious workers may collude with the server to compromise the privacy of honest workers by disclosing the secrets they share with them ($\mathcal{S}_{known} = \{S_{ij} : i \in [n] \setminus \mathcal{H}, j \in [n]\}$). However, even under this model, the server remains unaware of the malicious identities. A malicious worker may collude by revealing its secrets but has no incentive to disclose its adversarial nature, as it also aims to disrupt the learning.

### 2.3. Assumptions

Our results are derived under standard assumptions. Data heterogeneity is modeled via Assumption 2.1 (Allouah et al., 2023c), which subsumes standard assumptions in federated learning (Karimireddy et al., 2020) by bounding gradient variance in all directions. To establish the convergence guarantees of CAFCOR, we adopt the standard Assumption 2.2 on the variance of stochastic gradients. Assumption 2.3 ensures bounded gradients at any data point, a common requirement in differentially private optimization to avoid complications due to clipping (Agarwal et al., 2018).

**Assumption 2.1** (Bounded heterogeneity). *There exists $G_{\mathrm{cov}} > 0$ such that $\forall \theta \in \mathbb{R}^d$,*

$$\frac{1}{|\mathcal{H}|} \sum_{i \in \mathcal{H}} \langle v, \nabla \mathcal{L}_i(\theta) - \nabla \mathcal{L}_{\mathcal{H}}(\theta) \rangle^2 \leq G_{\mathrm{cov}}^2.$$

**Assumption 2.2** (Bounded variance). *There exists $\sigma > 0$ such that for each honest worker $w_i, i \in \mathcal{H}$, and all $\theta \in \mathbb{R}^d$,*

$$\frac{1}{m} \sum_{x \in \mathcal{D}_i} \|\nabla \ell(\theta; x) - \nabla \mathcal{L}_i(\theta)\|^2 \leq \sigma^2.$$

**Assumption 2.3** (Bounded gradients). *There exists $C > 0$ such that $\forall \theta \in \mathbb{R}^d$, $i \in \mathcal{H}$, and $x \in \mathcal{D}_i$, $\|\nabla \ell(\theta; x)\| \leq C$.*

**Algorithm 1** CAFCOR

**Input:** Initial model $\theta_0$; DP noise levels $\sigma_{\text{ind}}, \sigma_{\text{cor}}$; batch size $b$; clipping threshold $C$; learning rates $\{\gamma_t\}$; momentum coefficients $\{\beta_t\}$; number of iterations $T$.

1: **for** $t = 0 \ldots T - 1$ **do**
2:     Server broadcasts $\theta_t$ to all workers.
3:     **for every** honest worker $i \in \mathcal{H}$ **in parallel do**
4:         Sample a mini-batch $S_t^{(i)}$ of size $b$ at random from $\mathcal{D}_i$ without replacement.
5:         Average the mini-batch gradients and clip: $g_t^{(i)} = \text{Clip}\left(\frac{1}{b}\sum_{x \in S_t^{(i)}} \nabla\ell(\theta_t; x); C\right)$ (see (1)).
6:         Inject DP noises to the gradient: $\tilde{g}_t^{(i)} = g_t^{(i)} + \overline{v}_t^{(i)} + \sum_{j \in [n]\setminus i} v_t^{(ij)}$, where $\overline{v}_t^{(i)} \sim \mathcal{N}(0, \sigma_{\text{ind}}^2 I_d)$ and $v_t^{(ij)} = -v_t^{(ji)} \sim \mathcal{N}(0, \sigma_{\text{cor}}^2 I_d)$.
7:         Send $m_t^{(i)} = \beta_{t-1} m_{t-1}^{(i)} + (1 - \beta_{t-1})\tilde{g}_t^{(i)}$ if $t \geq 1$, else $m_0^{(i)} = 0$.
8:     **end for**
9:     Server aggregates: $R_t = \text{CAF}(m_t^{(1)}, \ldots, m_t^{(n)})$, where the CAF aggregation is given by Algorithm 2.
10:    Server updates the model: $\theta_{t+1} = \theta_t - \gamma_t R_t$.
11: **end for**
12: **return** $\hat{\theta}$ uniformly sampled from $\{\theta_0, \ldots, \theta_{T-1}\}$.

## 3. Algorithm: CAFCOR

We now introduce CAFCOR, summarized in Algorithm 1. CAFCOR extends distributed stochastic gradient descent (DSGD) by incorporating a correlated noise scheme and momentum-based updates at the worker level, and a novel robust aggregation scheme, called CAF, at the server level.

The server sets an initial model $\theta_0$ and broadcasts $\theta_t$ to workers at each iteration $t \geq 0$. Each honest worker $i \in \mathcal{H}$ then randomly samples a mini-batch $S_t^{(i)}$ of size $b \leq m$ from their local dataset $\mathcal{D}_i$ without replacement. Worker $i \in \mathcal{H}$ then computes per-example gradients $\nabla\ell(\theta_t; x)$ for $x \in S_t^{(i)}$, averages them, and then applies gradient clipping to bound the sensitivity, ensuring the gradient norm does not exceed $C$: $g_t^{(i)} = \text{Clip}\left(\frac{1}{b}\sum_{x \in S_t^{(i)}} \nabla\ell(\theta_t; x); C\right)$, where

$$\text{Clip}(g; C) := g \cdot \min\{1, C/\|g\|\}. \quad (1)$$

Each honest worker $i \in \mathcal{H}$ perturbs $g_t^{(i)}$ with noise to obtain $\tilde{g}_t^{(i)} = g_t^{(i)} + \sum_{j \in [n]\setminus i} v_t^{(ij)} + \overline{v}_t^{(i)}$, where $\overline{v}_t^{(i)} \sim \mathcal{N}(0, \sigma_{\text{ind}}^2 I_d)$ is an independent Gaussian noise term and $\{v_t^{(ij)}\}_{j \in [n]\setminus i}$ are pairwise-canceling correlated Gaussian noise terms, i.e., $v_t^{(ij)} = -v_t^{(ji)} \sim \mathcal{N}(0, \sigma_{\text{cor}}^2 I_d)$. These noise terms are generated through a one-time pairwise agreement protocol executed before CAFCOR. This is efficiently achieved using a public-key infrastructure, where workers establish shared randomness seeds in a single encrypted communication round (Bonawitz et al., 2017). To ensure

**Algorithm 2** CAF: Covariance bound-Agnostic Filter

**Input:** vectors $x_1, \ldots, x_n \in \mathbb{R}^d$; bound on number of corrupt inputs $0 \leq f < \frac{n}{2}$.

1: Initialize $c_1, \ldots, c_n = 1$, $\mu_\star = \frac{1}{n}\sum_{i=1}^n x_i$, $\sigma_\star = \infty$.
2: **while** $\sum_{i=1}^n c_i > n - 2f$ **do**
3:     Compute $\hat{\mu}_c = \sum_{i=1}^n c_i x_i / \sum_{i=1}^n c_i$.
4:     Compute empirical covariance $\hat{\Sigma}_c = \sum_{i=1}^n c_i (x_i - \hat{\mu}_c)(x_i - \hat{\mu}_c)^\top / \sum_{i=1}^n c_i$.
5:     Compute maximum eigenvalue $\hat{\sigma}_c^2$ of $\hat{\Sigma}_c$ and an associated eigenvector $\hat{v}_c$.
6:     **if** $\hat{\sigma}_c \leq \sigma_\star$ **then**
7:         Update best guess: $\mu_\star \leftarrow \hat{\mu}_c, \sigma_\star \leftarrow \hat{\sigma}_c$.
8:     **end if**
9:     Compute $\tau_i = \langle \hat{v}_c, x_i - \hat{\mu}_c \rangle^2, \forall i \in [n]$.
10:    Update weight $c_i \leftarrow c_i(1 - \frac{\tau_i}{\max_{j \in [n]} \tau_j}), \forall i \in [n]$.
11: **end while**
12: **return** $\mu_\star$

the correlated noise terms are pairwise-canceling, workers with smaller indices add the generated noise, while their paired counterparts subtract it.

Next, each honest worker $i \in \mathcal{H}$ updates their local momentum for $t \geq 1$: $m_t^{(i)} = \beta_{t-1} m_{t-1}^{(i)} + (1 - \beta_{t-1})\tilde{g}_t^{(i)}$, with $m_0^{(i)} = 0$ and $\beta_{t-1} \in [0, 1]$ is the momentum coefficient, and then sends it to the server. If worker $i$ is malicious, i.e., not honest, then they may send an arbitrary value for their momentum $m_t^{(i)}$. The server aggregates the momentums using the CAF aggregation, detailed in Algorithm 2, to obtain $R_t = \text{CAF}(m_t^{(1)}, \ldots, m_t^{(n)})$. It then updates the model: $\theta_{t+1} = \theta_t - \gamma_t R_t$, where $\gamma_t \geq 0$ is the learning rate. After $T$ iterations, the server outputs $\hat{\theta}$ sampled uniformly from $\{\theta_0, \ldots, \theta_{T-1}\}$.

**Covariance-bound Agnostic Filter (CAF).**[1] The CAF aggregation, outlined in Algorithm 2, aggregates $d$-dimensional vectors $x_1, \ldots, x_n \in \mathbb{R}^d$ given only an upper bound $0 \leq f < \frac{n}{2}$ on the number of corrupt inputs. CAF operates without requiring any additional assumptions on the distribution of honest inputs. The algorithm iteratively decreases the weights $c_i \in [0, 1]$ of input vectors, each initially assigned a unit weight, until the total sum of weights falls below $n - 2f$. Weight reductions are determined based on each vector's contribution to variance along the worst-case direction. Specifically, let $\hat{\mu}_c = \sum_{i=1}^n c_i x_i / \sum_{i=1}^n c_i$ denote the current weighted mean and let $\hat{\Sigma}_c = \sum_{i=1}^n c_i (x_i - \hat{\mu}_c)(x_i - \hat{\mu}_c)^\top / \sum_{i=1}^n c_i$ denote the corresponding empirical covariance matrix. The largest eigenvalue of $\hat{\Sigma}_c$ is denoted $\hat{\sigma}_c^2$, with associated eigenvector $\hat{v}_c \in \mathbb{R}^d$. Each vector's weight is reduced in

---

[1] Named for being agnostic to any bound on the honest input covariance, as discussed in Section 4.2.

proportion to its influence on variance along the worst direction, quantified by $\tau_i = \langle \hat{v}_c, x_i - \hat{\mu}_c \rangle^2$. CAF tracks the weighted average corresponding to the covariance matrix with the smallest top eigenvalue encountered across all iterations. Once the sum of weights falls below $n - 2f$, CAF outputs this tracked weighted average as its final result.

## 4. Theoretical Analysis

We now present the theoretical guarantees of CAFCOR in terms of privacy and utility, as well as their trade-off.

### 4.1. Privacy Analysis

Theorem 4.1 establishes the privacy guarantee of CAFCOR after $T$ iterations, with the proof deferred to Appendix A. As discussed in Section 4.2, the noise magnitudes $\sigma_{\text{ind}}^2$ and $\sigma_{\text{cor}}^2$ are key factors in the privacy-utility trade-off. Our objective is to minimize these terms while ensuring privacy.

**Theorem 4.1.** *Let* $\delta \in (0, 1), \varepsilon \in (0, \log(1/\delta)), q \leq f < \frac{n}{2}$. *Algorithm 1 satisfies* $(\varepsilon, \delta)$-*SecLDP against an honest-but-curious server colluding with q malicious workers, if*

$$\frac{(n-q)\sigma_{\text{cor}}^2 + \sigma_{\text{ind}}^2}{1 + \frac{\sigma_{\text{cor}}^2}{(f-q)\sigma_{\text{cor}}^2 + \sigma_{\text{ind}}^2}} \geq \frac{16C^2 T \log(1/\delta)}{\varepsilon^2}.$$

Theorem 4.1 guarantees SecLDP for any number of colluding malicious workers $q \leq f < \frac{n}{2}$, if the noise scales as $\sigma_{\text{ind}}^2 = \sigma_{\text{cor}}^2 = \Theta(\frac{1}{n\varepsilon^2})$, ignoring dependencies on other parameters. Notably, if at least one malicious worker $j$ does not collude with the server (i.e., $j$ retains its shared secrets and $q < f$), the independent noise term can be entirely removed ($\sigma_{\text{ind}} = 0$), while maintaining $\sigma_{\text{cor}}^2 = \Theta(\frac{1}{n\varepsilon^2})$ to satisfy SecLDP. The key insight is that each honest worker shares correlated noise with all other workers, including non-colluding malicious worker $j$. As a result, the server cannot cancel the correlated noise associated with worker $j$'s hidden secrets. This effectively serves as an independent noise source for honest workers, eliminating the need for an explicit $\sigma_{\text{ind}}$ in the correlated noise scheme. Finally, if the shared randomness assumption among workers is unavailable, CAFCOR naturally falls back to the LDP setting by setting $\sigma_{\text{cor}} = 0$ in Algorithm 1. Conceptually, this corresponds to a scenario where all secrets are revealed, allowing the server to eliminate correlated noise and leaving only independent noise for privacy. Consequently, $\sigma_{\text{ind}}^2$ must scale as $\Theta(\frac{1}{\varepsilon^2})$, which is $n$ times larger than under SecLDP.

The proof of Theorem 4.1, deferred to Appendix A, relies upon computing the Rényi divergence induced by the correlated noise scheme, before converting back to DP following standard conversion results (Mironov, 2017). Indeed, we derive the exact expression of the aforementioned Rényi divergence in Lemma A.4, which we use directly in our Rényi-based privacy accountant for SecLDP in prac-

tice. This exact computation was not feasible in previous related works, as they either considered different communication topologies (Sabater et al., 2022; Allouah et al., 2024). The major difference with the aforementioned analyses is granularity: we consider $0 \leq q \leq f$ workers who may poison training but are not curious, while prior works considered one extreme or the other, i.e., $q \in \{0, f\}$. This granularity induces improved empirical performance (Figure 1).

### 4.2. Utility Analysis

We first study the robustness of the CAF aggregation (Algorithm 2) in Proposition 4.1, and then analyze the convergence of CAFCOR (Algorithm 1) in Theorem 4.2. The proof is deferred to Appendix B.

**Proposition 4.1.** *Let* $0 \leq f < \frac{n}{2}$ *and* $\kappa := \frac{6f}{n-f}\left(1 + \frac{f}{n-2f}\right)^2$. *Let* $x_1, \ldots, x_n \in \mathbb{R}^d$, *and* $S \subseteq [n]$ *be any subset of size* $n - f$. *Let* $\overline{x}_S := \frac{1}{|S|}\sum_{i \in S} x_i$, *and* $\lambda_{\max}$ *denote the maximum eigenvalue operator. The output* $\hat{x} = \text{CAF}(x_1, \ldots, x_n)$ *satisfies:*

$$\|\hat{x} - \overline{x}_S\|^2 \leq \kappa \cdot \lambda_{\max}\left(\frac{1}{|S|}\sum_{i \in S}(x_i - \overline{x}_S)(x_i - \overline{x}_S)^\top\right).$$

Proposition 4.1 establishes an important property satisfied by CAF. The above corresponds to the high-dimensional robustness criterion studied by Allouah et al. (2023c), shown to be stronger than existing ones in the distributed learning literature, e.g., that of Karimireddy et al. (2022). Furthermore, we recall from Allouah et al. (2023c) that standard robust aggregations, such as the trimmed mean and median (Yin et al., 2018), can only achieve a weaker variant of Proposition 4.1, where the right-hand side of the inequality involves the trace instead of the maximum eigenvalue operator. Since the trace of a $d \times d$ matrix can be up to $d$ times its largest eigenvalue, standard robust aggregations only satisfy Proposition 4.1 with an additional multiplicative factor of $d$ in $\kappa$ (4.1), leading to a looser robustness bound.

Moreover, the so-called SMEA aggregation of Allouah et al. (2023c) also satisfies Proposition 4.1 up to constants. However, SMEA is computationally impractical as its time complexity grows exponentially with $f$. In contrast, CAF is efficient, running in polynomial time $\mathcal{O}(f(nd^2 + d^3))$. Indeed, at each iteration, at least one input ($i \in \text{argmax}_{j \in [n]} \tau_j$) is assigned a weight of zero. Thus, CAF terminates within at most $2f$ iterations, each requiring the computation of the top eigenvector and the formation of the empirical covariance matrix. To further improve efficiency, we implement a CAF variant with reduced complexity $\mathcal{O}(fnd \log d)$ by approximating the top eigenvector using the power method. This approach avoids explicit covariance matrix formation and instead relies on matrix-vector products. We adopt this more efficient variant in our experiments in Section 5, as it can be easily shown to satisfy Proposition 4.1 up to constants.

CAF builds on recent algorithmic advances in robust statistics (Diakonikolas et al., 2017). A major difference is that CAF does not require a bound on the covariance of honest inputs. In contrast, prior robust mean estimators have relied on such bounds as a crucial design requirement (Diakonikolas et al., 2017; Steinhardt et al., 2018). CAF operates instead under a more practical assumption, requiring only a bound on the number of corrupt inputs $f$, a standard assumption in robust ML (Yin et al., 2018; Karimireddy et al., 2021; Farhadkhani et al., 2022). By eliminating covariance constraints, CAF is more suited to our setting, where information about honest inputs such as gradients or momentums may not be available or could compromise privacy.

Next, Theorem 4.2 establishes the convergence rate of CAF-COR, with the proof deferred to Appendix C.

**Theorem 4.2.** *Suppose that Assumptions 2.1, 2.2, and 2.3 hold, and that $\mathcal{L}_{\mathcal{H}}$ is L-smooth. Recall $\kappa$ from (4.1). Let*

$$\bar{\sigma}^2 := \frac{\sigma_b^2 + d(f\sigma_{\text{cor}}^2 + \sigma_{\text{ind}}^2)}{n - f}$$
$$+ 4\kappa\left(\sigma_b^2 + 108\left(n\sigma_{\text{cor}}^2 + \sigma_{\text{ind}}^2\right)\left(1 + \frac{d}{n-f}\right)\right),$$

*where $\sigma_b^2 := 2(1 - \frac{b}{m})\frac{\sigma^2}{b}$. We also denote $\mathcal{L}_\star := \inf_{\theta \in \mathbb{R}^d} \mathcal{L}_{\mathcal{H}}(\theta)$, $\mathcal{L}_0 := \mathcal{L}_{\mathcal{H}}(\theta_0) - \mathcal{L}_\star$. Consider Algorithm 1 with $T \geq 1$, the learning rates $\gamma_t$ specified below, and momentum coefficients $\beta_t = 1 - 24L\gamma_t$. The following holds:*

1. *$\mathcal{L}_{\mathcal{H}}$ is $\mu$-strongly convex: If $\gamma_t = \frac{10}{\mu(t+240\frac{L}{\mu})}$, then*

$$\mathbb{E}\left[\mathcal{L}_{\mathcal{H}}(\theta_T) - \mathcal{L}_*\right] \lesssim \frac{\kappa G_{\text{cov}}^2}{\mu} + \frac{L\bar{\sigma}^2}{\mu^2 T} + \frac{L^2 \mathcal{L}_0}{\mu^2 T^2},$$

2. *$\mathcal{L}_{\mathcal{H}}$ is non-convex: If $\gamma_t = \min\left\{\frac{1}{24L}, \frac{\sqrt{3\mathcal{L}_0}}{16\bar{\sigma}\sqrt{LT}}\right\}$, then*

$$\mathbb{E}\|\nabla\mathcal{L}_{\mathcal{H}}(\hat{\theta})\|^2 \lesssim \kappa G_{\text{cov}}^2 + \frac{\bar{\sigma}\sqrt{L\mathcal{L}_0}}{\sqrt{T}} + \frac{L\mathcal{L}_0}{T},$$

*where $\lesssim$ denotes inequality up to absolute constants and the expectation is over the randomness of the algorithm.*

Our result recovers the state-of-the-art convergence analysis in robust optimization in the local privacy model (Allouah et al., 2023c) when setting $\sigma_{\text{cor}} = 0$. The key difference with the latter analysis is the role of the correlated noise magnitude $\sigma_{\text{cor}}$, which appears in the dominating term inside $\bar{\sigma}$, in both strongly convex and non-convex settings. Moreover, our convergence rates exhibit the standard dependence on the number of iterations in smooth stochastic optimization: $\mathcal{O}(\frac{1}{T})$ for the strongly convex case and $\mathcal{O}(\frac{1}{\sqrt{T}})$ for the non-convex case. The presence of a non-vanishing term in $T$ is fundamental in robust optimization in heterogeneous data settings (Karimireddy et al., 2022). Finally, our upper bound in the strongly convex case also applies to non-convex loss functions that satisfy the Polyak-Lojasiewicz inequality (Karimi et al., 2016).

### 4.3. Privacy-Utility Trade-off

Corollary 4.1 below, which quantifies the privacy-utility trade-off of CAFCOR, follows directly from combining the results of Theorems 4.1 and 4.2.

**Corollary 4.1.** *Let $\delta \in (0,1), \varepsilon \in (0, \log(1/\delta))$ and $n \geq (2+\eta)f$, for some absolute constant $\eta > 0$. Consider Algorithm 1 in the strongly convex setting of Theorem 4.2. If $\sigma_{\text{cor}}^2 = \sigma_{\text{ind}}^2 = \frac{32C^2 T \log(1/\delta)}{\varepsilon^2(n-f)}$, then Algorithm 1 is $(\varepsilon, \delta)$-SecLDP against an honest-but-curious server colluding with all malicious workers, and $(f, \varrho)$-robust where $\varrho = \mathcal{O}\left(\frac{(f+1)C^2 d\log(1/\delta)}{n^2\varepsilon^2} + \frac{fC^2\log(1/\delta)}{n\varepsilon^2} + \frac{f}{n}G_{\text{cov}}^2\right)$, asymptotically in $T$ and ignoring absolute constants.*

To analyze the privacy-utility trade-off, we examine the first term of $\varrho$ from Corollary 4.1. This term dominates the second whenever $d \geq n$, a condition that is typically met in practice, given that modern deep learning models have dimensions in the order of millions (He et al., 2016), while the number of clients in federated learning, particularly in cross-silo settings, is often much smaller (Kairouz et al., 2021). The final term in the error stems solely from data heterogeneity and becomes negligible when data is not highly heterogeneous. Overall, CAFCOR achieves a privacy-utility trade-off of $\widetilde{\mathcal{O}}(\frac{(f+1)d}{n^2\varepsilon^2})$. Notably, our analysis assumes the worst case where all malicious workers collude with the server to breach privacy ($q = f$ in Theorem 4.1). If they do not (i.e., $q = 0$), the trade-off still holds, but constants improve with the choice of $\sigma_{\text{ind}} = 0$, as discussed after Theorem 4.1 and empirically validated in Section 5.

For comparison, recall that the minimax optimal privacy-utility trade-off in LDP is $\widetilde{\Theta}(\frac{d}{\varepsilon^2})$ (Duchi et al., 2018), which is $n$ times larger than the corresponding rate in CDP given by $\widetilde{\Theta}(\frac{d}{n^2\varepsilon^2})$ (Bassily et al., 2014), both under user-level dataset adjacency. Therefore, our rate of $\widetilde{\mathcal{O}}(\frac{(f+1)d}{n^2\varepsilon^2})$ derived under SecLDP naturally bridges these two extremes. Notably, as the number of malicious workers $f$ decreases, our rate converges to CDP. Specifically, our rate matches CDP when $f = \widetilde{\mathcal{O}}(1)$ and remains strictly better than LDP as long as $f$ is sublinear in $n$. Furthermore, under the LDP privacy model ($\sigma_{\text{cor}} = 0$), such as when shared randomness among workers is unavailable, CAFCOR recovers the state-of-the-art privacy-utility trade-off of $\widetilde{\Theta}(\frac{d}{n\varepsilon^2})$.

CAF plays a crucial role in ensuring that Algorithm 1 achieves the derived trade-off. In contrast, employing standard robust aggregations, such as the trimmed mean or median (Yin et al., 2018), significantly degrades theoretical performance. Specifically, after replacing CAF with the trimmed mean, the resulting convergence rate is $\widetilde{\mathcal{O}}(\frac{fd}{n\varepsilon^2})$ for $f > 0$ (see Appendix C.5). This rate is $n$ times worse than that achieved with CAF and only matches LDP when $f = \widetilde{\mathcal{O}}(1)$. For larger $f$, performance further degrades, with a rate strictly worse than LDP.

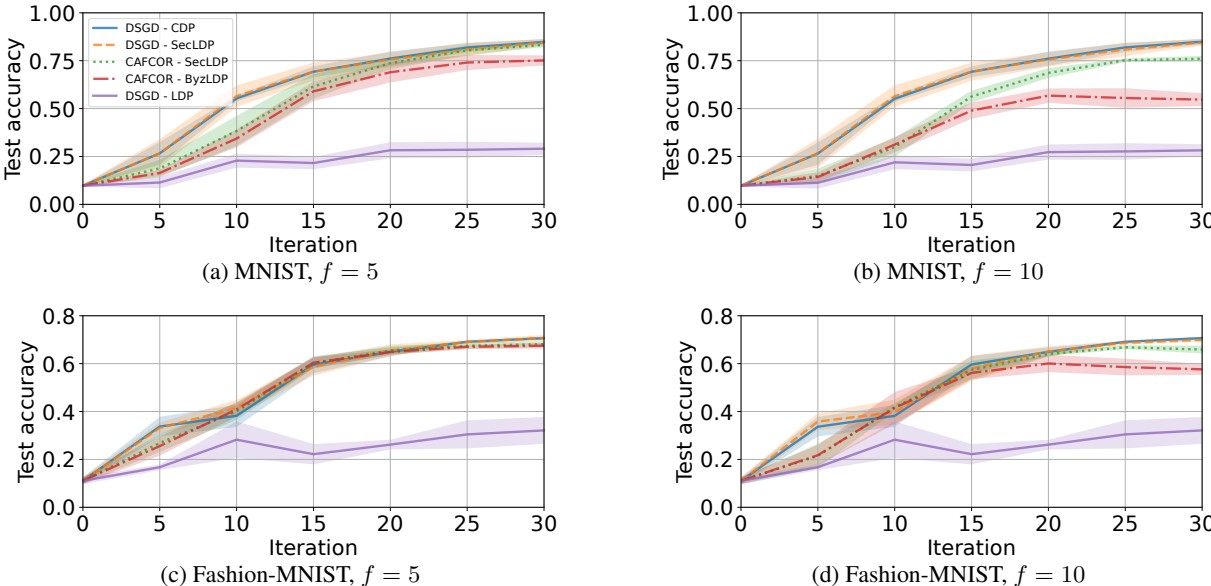

Figure 1: Comparison of CAFCOR and DSGD under four privacy threat models on (Fashion-)MNIST. There are $n = 100$ workers, including $f = 5$ and 10 malicious workers executing ALIE (Baruch et al., 2019). A homogeneous data distribution is considered among honest workers. User-level DP is used and privacy budgets for MNIST are $\varepsilon = 26.4$ and 27.8 for $f = 10$ and $f = 5$, respectively. For Fashion-MNIST, the privacy budget is $\varepsilon = 39.6$. Throughout, we set $\delta = 10^{-4}$.

## 5. Experimental Evaluation

We empirically evaluate the performance of CAFCOR in a distributed environment under varying threat models. We also compare CAFCOR to existing robust and private algorithms, demonstrating its empirical advantages.

**Datasets and models.** We consider two widely used image classification datasets: MNIST (LeCun & Cortes, 2010) and Fashion-MNIST (Xiao et al., 2017). While these tasks are relatively simple in standard, non-adversarial settings, they become challenging with the introduction of malicious workers (Karimireddy et al., 2021; Farhadkhani et al., 2022) and the differential privacy constraint (Noble et al., 2022). MNIST images are normalized using a mean of 0.1307 and standard deviation of 0.3081, while Fashion-MNIST images are horizontally flipped. To simulate data heterogeneity, we sample data for honest workers using a Dirichlet distribution of parameter $\alpha$ (Hsu et al., 2019). We consider $\alpha \in \{0.1, 1\}$, a homogeneous distribution scenario, and an extreme heterogeneity scenario where data is sorted by label and evenly assigned to honest workers. On both datasets, we train the same convolutional neural network with a model size of 431,080 parameters, using the negative log-likelihood loss.

**Attacks.** We consider four state-of-the-art attacks: Fall of Empires (FOE) (Xie et al., 2020), A Little is Enough (ALIE) (Baruch et al., 2019), Label Flipping (LF) and Sign Flipping (SF) (Allen-Zhu et al., 2020).

**Reproducibility.** To estimate the privacy budgets achieved at the end of training, we use Opacus (Yousefpour et al., 2021). Our experiments are conducted with five random seeds (1-5) to ensure reproducibility. All reported results include standard deviation across these seeds. To facilitate reproducibility, we intend to publicly release our code.

### 5.1. Comparison of Threat Models

In Figure 1, we compare CAFCOR to the DSGD baseline across four privacy regimes: LDP, CDP, SecLDP (assuming no collusion), and a novel threat model we call ByzLDP discussed in Section 2, where all malicious workers collude with the server. We execute CAFCOR in a distributed system with $n = 100$ workers among which $f \in \{5, 10\}$ are malicious. On MNIST, we use batch size $b = 50$, learning rate $\gamma = 0.075$, momentum parameter $\beta = 0.85$, and clipping parameter $C = 2.25$. For Fashion-MNIST, we use $b = 100$, $\gamma = 0.3$, $\beta = 0.9$, and $C = 1$. For both datasets, we train for $T = 30$ iterations and apply $\ell_2$-regularization at $10^{-4}$. We adopt user-level DP across all threat models. On MNIST, the privacy budgets reach $\varepsilon = 26.4$ and 27.8 for $f = 10$ and $f = 5$, respectively. On Fashion-MNIST, the privacy budget is $\varepsilon = 39.6$ for both values of $f$. In Figure 1, we consider malicious workers executing the ALIE attack. Results for the remaining attacks are deferred to Appendix D.1, where similar trends are observed.

As illustrated in Figure 1, the performance of DSGD under LDP is significantly lower than under CDP. Under SecLDP, CAFCOR achieves performance closely match-

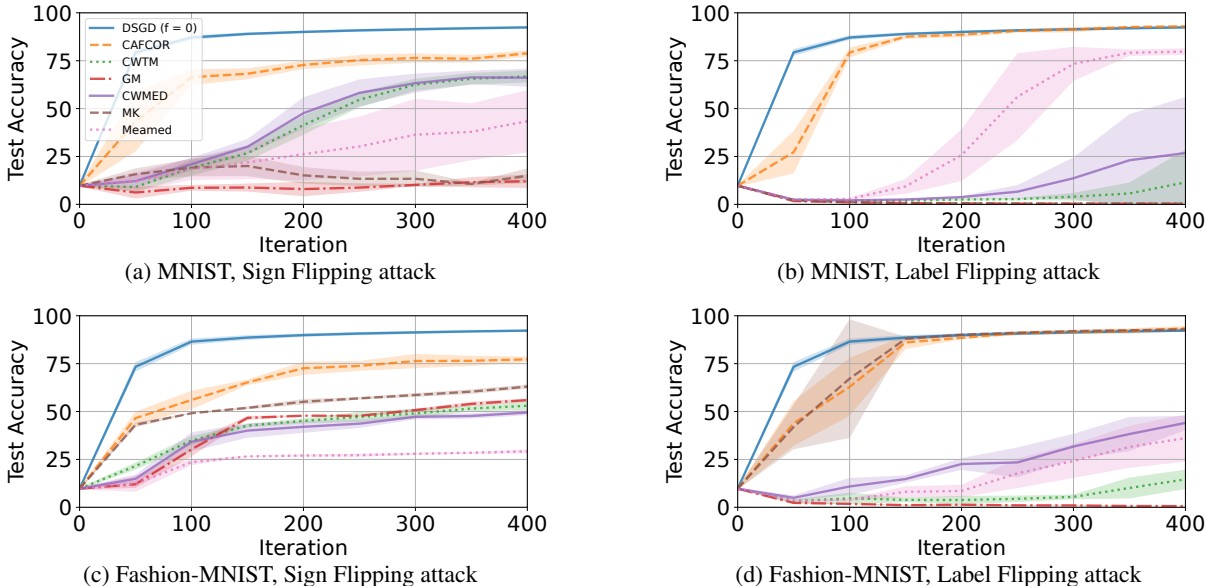

Figure 2: Performance of CAFCOR versus standard robust algorithms. There are $f = 5$ malicious workers among $n = 15$ on MNIST and $\alpha = 1$, and $f = 3$ malicious workers among $n = 13$ on Fashion-MNIST and $\alpha = 0.1$. The CDP privacy model is considered under example-level DP, and the privacy budget is $(\varepsilon, \delta) = (13.7, 10^{-4})$ throughout.

ing CDP and substantially outperforming LDP on both datasets. For $f = 5$, CAFCOR-SecLDP exhibits identical final accuracies to the adversary-free DSGD baseline under CDP: 83% on MNIST and 72% on Fashion-MNIST. When $f = 10$, CAFCOR-SecLDP has a slight decrease in accuracy compared to DSGD-CDP but remains significantly superior to DSGD-LDP. Under the stronger ByzLDP threat model, CAFCOR maintains performance comparable to SecLDP for $f = 5$. On Fashion-MNIST, the difference is negligible, demonstrating effective mitigation of malicious worker collusion. On MNIST, however, accuracy decreases by 8% due to the more stringent privacy constraints. For $f = 10$, the gap between CAFCOR-SecLDP and CAFCOR-ByzLDP is more pronounced, particularly on MNIST (see Figure 1b). Yet, CAFCOR-ByzLDP still significantly outperforms DSGD-LDP, where training struggles to exceed 30% accuracy. These findings underscore the practical utility of SecLDP and ByzLDP, and the robustness of CAFCOR under stringent privacy constraints. For completeness, we also evaluate CAFCOR under the CDP and LDP privacy models in Appendix D.1, demonstrating that the performance of CAFCOR matches the DSGD baseline under both models.

### 5.2. Comparison with Prior Robust Aggregations

To highlight the performance of our aggregation, we replace CAF in Algorithm 1 with standard alternatives in robust distributed learning: coordinate-wise trimmed mean (CWTM) and median (CWMED) (Yin et al., 2018), geometric median (GM) (Chen et al., 2017), Multi-Krum (MK) (Blanchard et al., 2017), and mean around median (Meamed) (Xie et al., 2018). We consider a system with $n - f = 10$ honest

workers and $f \in \{1, 3, 5\}$ malicious workers, with heterogeneous distributions with $\alpha \in \{0.1, 1\}$ and an extreme distribution. Training is conducted with $b = 100$, $C = 5$, and $T = 400$ iterations for both datasets. Figure 2 presents results for LF and SF attacks with $f = 5$, $\alpha = 1$ on MNIST, and $f = 3$, $\alpha = 0.1$ on Fashion-MNIST. Example-level DP is applied under the CDP privacy model, yielding a privacy budget of $(\varepsilon, \delta) = (13.7, 10^{-4})$ across all configurations.

Figure 2 demonstrates the clear superiority of CAFCOR (and CAF) over competing aggregations across all configurations, even under the least stringent privacy model (CDP). While MK matches CAFCOR on Fashion-MNIST under the LF attack, its performance significantly deteriorates under the SF attack and on MNIST. Meamed performs relatively well under LF on MNIST but struggles under LF and SF attacks on Fashion-MNIST and under the SF attack on MNIST, with CAFCOR outperforming it consistently. Other aggregations, such as CWTM, CWMED, and GM, exhibit poor performance across both datasets and attacks, particularly under LF, where their utility drops significantly and the best competing aggregation among them achieves an accuracy just above 50%. In contrast, CAF achieves results nearly indistinguishable from adversary-free DSGD under the LF attack on both datasets (90% in accuracy) and delivers the best performance under SF, reaching an accuracy close to 80%. This pattern holds consistently across the remaining attacks, as detailed in Appendix D.2.

# 6. Conclusion

Achieving good utility while simultaneously ensuring both robustness against malicious parties and strong privacy in distributed learning typically requires trusting the central server, an often unrealistic assumption. We address this challenge with CAFCOR, a novel algorithm leveraging shared randomness among workers to enable correlated noise injection prior to robust gradient aggregation. Even under collusion between malicious workers and an untrusted server, CAFCOR achieves a strong privacy-utility trade-off, significantly outperforming local DP (where there are no trust assumptions) while approaching the performance of central DP (where the server is trusted). Our results show that privacy and robustness can coexist without sacrificing utility, yet open questions remain. Notably, what is the optimal utility under SecLDP, and does CAFCOR achieve it?

# Acknowledgements

YA acknowledges support by SNSF grant 200021_200477.

# Impact Statement

This paper presents work whose goal is to advance the field of Machine Learning. There are many potential societal consequences of our work, none of which we feel must be specifically highlighted here.

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

## Organization of the Appendix

Appendix A presents the privacy analysis of CAFCOR. Appendix B contains the robustness analysis of Proposition 4.1. Appendix C provides the convergence analysis of CAFCOR. Appendix D contains additional experimental results.

## A. Privacy Analysis: Proof of Theorem 4.1

**Definition A.1** ($\alpha$-Rényi divergence). *Let $\alpha > 0, \alpha \neq 1$. The $\alpha$-Rényi divergence between two probability distributions $P$ and $Q$ is defined as*

$$D_\alpha(P \parallel Q) := \frac{1}{\alpha - 1} \log \mathbb{E}_{X \sim Q} \left( \frac{P(X)}{Q(X)} \right)^\alpha.$$

**Lemma A.1** ((Gil et al., 2013)). *Let $\alpha > 0, \alpha \neq 1, \mu_1, \mu_2 \in \mathbb{R}^n$, and $\Sigma \in \mathbb{R}^{n \times n}$. Assume that $\Sigma$ is positive definite. The $\alpha$-Rényi divergence between the multivariate Gaussian distributions $\mathcal{N}(\mu_1, \Sigma)$ and $\mathcal{N}(\mu_2, \Sigma)$ is*

$$D_\alpha(\mathcal{N}(\mu_1, \Sigma) \parallel \mathcal{N}(\mu_2, \Sigma)) = \frac{\alpha}{2}(\mu_1 - \mu_2)^\top \Sigma^{-1}(\mu_1 - \mu_2).$$

We now define *secret-based local Rényi differential privacy* (SecRDP), a strong variant of SecLDP based on Rényi DP, paraphrasing Allouah et al. (2024).

**Definition A.2** (SecRDP). *Let $\varepsilon \geq 0$, $\delta \in [0, 1]$, $\alpha > 1$. Consider a randomized decentralized algorithm $\mathcal{A}_{\mathcal{S}} : \mathcal{X}^{m \times n} \to \mathcal{Y}$, which outputs the transcript of all communications, given the full set of secrets $\mathcal{S}$. Algorithm $\mathcal{A}_{\mathcal{S}}$ is said to satisfy $(\alpha, \varepsilon, \mathcal{S}_{known})$-SecRDP if $\mathcal{A}_{\mathcal{S}}$ satisfies $(\alpha, \varepsilon)$-RDP, given that the subset of secrets $\mathcal{S}_{known} \subseteq \mathcal{S}$ is revealed to the adversary. That is, for every adjacent datasets $\mathcal{D}, \mathcal{D}' \in \mathcal{X}^{m \times n}$, we have*

$$D_\alpha(\mathcal{A}_{\mathcal{S}}(\mathcal{D}) \mid \mathcal{S}_{known} \parallel \mathcal{A}_{\mathcal{S}}(\mathcal{D}') \mid \mathcal{S}_{known}) \leq \varepsilon,$$

*where the left-hand side is the Rényi divergence (Definition A.2) between the probability distributions of $\mathcal{A}(\mathcal{D})$ and $\mathcal{A}(\mathcal{D}')$, conditional on the secrets $\mathcal{S}_{all}$ revealed to the adversary. We simply say that $\mathcal{A}_{\mathcal{S}}$ satisfies $(\alpha, \varepsilon)$-SecRDP if it satisfies $(\alpha, \varepsilon, \mathcal{S}_{known})$-SecRDP and $\mathcal{S}_{known}$ is clear from the context.*

Both SecLDP and SecRDP preserve the properties of DP and RDP, respectively, since these relaxations only condition the probability space of the considered distributions.

**Lemma A.2** (RDP Adpative Composition, (Mironov, 2017)). *If $\mathcal{M}_1$ that takes the dataset as input is $(\alpha, \varepsilon_1)$-RDP, and $\mathcal{M}_2$ that takes the dataset and the output of $\mathcal{M}_1$ as input is $(\alpha, \varepsilon_2)$-RDP, then their composition is $(\alpha, \varepsilon_1 + \varepsilon_2)$-RDP.*

**Lemma A.3** (RDP to DP conversion, (Mironov, 2017)). *If $\mathcal{M}$ is $(\alpha, \varepsilon)$-RDP, then $\mathcal{M}$ is $(\varepsilon + \frac{\log(1/\delta)}{\alpha - 1}, \delta)$-DP for all $\delta \in (0, 1)$.*

We first prove a lemma analyzing a single iteration of Algorithm 1 within the SecRDP formalism.

**Lemma A.4.** *Let $\alpha > 1$ and $q \leq f$. Each iteration of Algorithm 1 satisfies $(\alpha, \alpha\varepsilon)$-SecRDP (Definition A.2) against the honest-but-curious server, colluding with $q$ malicious workers, where*

$$\varepsilon = \frac{2C^2}{(n - q)\sigma_{\text{cor}}^2 + \sigma_{\text{ind}}^2} \left[ 1 + \frac{\sigma_{\text{cor}}^2}{(f - q)\sigma_{\text{cor}}^2 + \sigma_{\text{ind}}^2} \right]. \tag{2}$$

*Proof.* Let $\alpha > 1$, $q > 1$, and $\mathcal{I} \subseteq [n] \setminus \mathcal{H}$ be an arbitrary group of $|\mathcal{I}| = q$ malicious workers. Recall that we denote by $\mathcal{S}_{\mathcal{I}} := \{s_{jk} : j \in [n], k \in \mathcal{I}\}$ the set of secrets revealed by the malicious workers in $\mathcal{I}$ to the server. We will prove that Algorithm 1 satisfies $(\alpha, \varepsilon, \mathcal{S}_{\mathcal{I}})$-SecRDP, which protects against the honest-but-curious server, in addition to $q$ colluding malicious workers. For ease of exposition, we consider the one-dimensional case $d = 1$. Extending the proof to the general case is straightforward.

Formally, at each iteration of Algorithm 1, users possess private inputs (gradients) in the form of vector $x \in [-C, C]^n$, given that gradients are clipped at threshold $C$. Each user $i \in [n]$ shares the following privatized quantity:

$$\widetilde{x}_i := x_i + \sum_{\substack{j=1 \\ j \neq i}}^{n} v_{ij} + \bar{v}_i, \tag{3}$$

where $v_{ij} = -v_{ji} \sim \mathcal{N}(0, \sigma_{\text{cor}}^2)$ for all $j \in [n]$, and $\bar{v}_i \sim \mathcal{N}(0, \sigma_{\text{ind}}^2)$.

Our goal is to show that the mechanism producing $\widetilde{X}_{\mathcal{H}} := [\widetilde{x}_i]_{i \in \mathcal{H}}$ satisfies SecRDP when a single entry of $X := [x_i]_{i \in \mathcal{H}}$ is arbitrarily changed; i.e., one user's input differs. To do so, we first rewrite (3) to discard the noise terms known to the colluding curious users who can simply substract them to get for every $i \in \mathcal{H}$:

$$\widetilde{x}_i = x_i + \sum_{\substack{j=1 \\ j \notin \mathcal{I} \cup \{i\}}}^{n} v_{ij} + \bar{v}_i = x_i + \sum_{j \in \mathcal{H}} v_{ij} + \sum_{\substack{j=1 \\ j \notin \mathcal{H} \cup \mathcal{I}}}^{n} v_{ij} + \bar{v}_i = x_i + \sum_{j \in \mathcal{H}} v_{ij} + \bar{u}_i, \tag{4}$$

where $\bar{u}_i := \sum_{\substack{j=1 \\ j \notin \mathcal{H} \cup \mathcal{I}}}^{n} v_{ij} + \bar{v}_i \sim \mathcal{N}(0, (f-q)\sigma_{\text{cor}}^2 + \sigma_{\text{ind}}^2)$. We now rewrite the above in matrix form as:

$$\widetilde{X}_{\mathcal{H}} = X_{\mathcal{H}} + K N_{\mathcal{H}} + \bar{N}, \tag{5}$$

where $K \in \mathbb{R}^{(n-f) \times \frac{(n-f)(n-f-1)}{2}}$ is the oriented incidence matrix of the complete graph over $\mathcal{H}$, $N_{\mathcal{H}} = [v_{ij}]_{1 \leq i < j \leq n-f} \in \mathbb{R}^{\frac{(n-f)(n-f-1)}{2}}$ is the vector of pairwise noises, and $\bar{N} := [\bar{u}_i]_{i \in \mathcal{H}}$. Now, consider two input vectors $X_A, X_B \in [-C, C]^{n-f}$ which differ maximally in an arbitrary coordinate $i \in [n-f]$ without loss of generality:

$$X_A - X_B = 2C e_i \in \mathbb{R}^{n-f}, \tag{6}$$

where $e_i$ is the vector of $\mathbb{R}^{n-f}$ where the only nonzero element is 1 in the $i$-th coordinate.

We will then show that the $\alpha$-Rényi divergence between $\widetilde{X}_A$ and $\widetilde{X}_B$, which are respectively produced by input vectors $X_A$ and $X_B$, is bounded. To do so, by looking at Equation (5), we can see that $\widetilde{X}_A, \widetilde{X}_B$ follow a multivariate Gaussian distribution of means $X_A, X_B$ respectively and of variance

$$\Sigma := \mathbb{E}(\widetilde{X}_A - X_A)(\widetilde{X}_A - X_A)^\top = \mathbb{E}(\widetilde{X}_B - X_B)(\widetilde{X}_B - X_B)^\top = \sigma_{\text{cor}}^2 L + \left((f-q)\sigma_{\text{cor}}^2 + \sigma_{\text{ind}}^2\right) I_{n-q} \in \mathbb{R}^{(n-f) \times (n-f)}, \tag{7}$$

where $L = K K^\top \in \mathbb{R}^{(n-f) \times (n-f)}$ is the Laplacian matrix of the complete graph over $\mathcal{H}$. Note that $\Sigma$ is positive definite when $(f-q)\sigma_{\text{cor}}^2 + \sigma_{\text{ind}}^2 > 0$, because $L$ is positive semi-definite. Also, recalling the expression $L := (n-f)I_{n-f} - \mathbf{1}\mathbf{1}^\top$ of the Laplacian of the complete graph over $\mathcal{H}$, we have

$$\begin{aligned} \Sigma &= \sigma_{\text{cor}}^2 L + \left((f-q)\sigma_{\text{cor}}^2 + \sigma_{\text{ind}}^2\right) I_{n-f} = \sigma_{\text{cor}}^2 \left((n-f)I_{n-f} - \mathbf{1}\mathbf{1}^\top\right) + \left((f-q)\sigma_{\text{cor}}^2 + \sigma_{\text{ind}}^2\right) I_{n-f} \\ &= \left((n-q)\sigma_{\text{cor}}^2 + \sigma_{\text{ind}}^2\right) I_{n-f} - \sigma_{\text{cor}}^2 \mathbf{1}\mathbf{1}^\top. \end{aligned} \tag{8}$$

Moreover, recall the Sherman-Morrison formula, i.e., for any matrix $A$ and vectors $u, v$ such that $v^\top A^{-1} u \neq -1$, we have $(A + uv^\top)^{-1} = A^{-1} - \frac{A^{-1}uv^\top A^{-1}}{1 + v^\top A^{-1} u}$. Using this formula, we have

$$\begin{aligned} \Sigma^{-1} &= \left[\left((n-q)\sigma_{\text{cor}}^2 + \sigma_{\text{ind}}^2\right) I_{n-f} - \sigma_{\text{cor}}^2 \mathbf{1}\mathbf{1}^\top\right]^{-1} \\ &= \frac{1}{(n-q)\sigma_{\text{cor}}^2 + \sigma_{\text{ind}}^2} I_{n-f} + \frac{1}{[(n-q)\sigma_{\text{cor}}^2 + \sigma_{\text{ind}}^2]^2} \frac{\sigma_{\text{cor}}^2}{1 - \frac{(n-f)\sigma_{\text{cor}}^2}{(n-q)\sigma_{\text{cor}}^2 + \sigma_{\text{ind}}^2}} \mathbf{1}\mathbf{1}^\top \\ &= \frac{1}{(n-q)\sigma_{\text{cor}}^2 + \sigma_{\text{ind}}^2} I_{n-f} + \frac{1}{(n-q)\sigma_{\text{cor}}^2 + \sigma_{\text{ind}}^2} \frac{\sigma_{\text{cor}}^2}{(f-q)\sigma_{\text{cor}}^2 + \sigma_{\text{ind}}^2} \mathbf{1}\mathbf{1}^\top. \end{aligned} \tag{9}$$

Finally, following Lemma A.1, the $\alpha$-Rényi divergence between the distributions of $\widetilde{X}_A$ and $\widetilde{X}_B$ is

$$\begin{aligned} D_\alpha(\widetilde{X}_A \| \widetilde{X}_B) &= \frac{\alpha}{2}(X_A - X_B)^\top \Sigma^{-1}(X_A - X_B) \\ &= 2\alpha C^2 \left[\frac{1}{(n-q)\sigma_{\text{cor}}^2 + \sigma_{\text{ind}}^2} + \frac{1}{(n-q)\sigma_{\text{cor}}^2 + \sigma_{\text{ind}}^2} \frac{\sigma_{\text{cor}}^2}{(f-q)\sigma_{\text{cor}}^2 + \sigma_{\text{ind}}^2}\right] \\ &= \frac{2\alpha C^2}{(n-q)\sigma_{\text{cor}}^2 + \sigma_{\text{ind}}^2} \left[1 + \frac{\sigma_{\text{cor}}^2}{(f-q)\sigma_{\text{cor}}^2 + \sigma_{\text{ind}}^2}\right]. \end{aligned} \tag{10}$$

$\square$

We now prove the SecLDP property of the full Algorithm 1 by using the composition properties of RDP and converting to DP (Mironov, 2017).

**Theorem 4.1.** *Let $\delta \in (0, 1), \varepsilon \in (0, \log(1/\delta)), q \leq f < \frac{n}{2}$. Algorithm 1 satisfies $(\varepsilon, \delta)$-SecLDP against an honest-but-curious server colluding with $q$ malicious workers, if*

$$\frac{(n-q)\sigma_{\text{cor}}^2 + \sigma_{\text{ind}}^2}{1 + \frac{\sigma_{\text{cor}}^2}{(f-q)\sigma_{\text{cor}}^2 + \sigma_{\text{ind}}^2}} \geq \frac{16C^2 T \log(1/\delta)}{\varepsilon^2}.$$

*Proof.* Recall from Lemma A.4 that each iteration of Algorithm 1 satisfies $(\alpha, \alpha\varepsilon_{\text{step}})$-SecRDP against collusion at level $q$ for every $\alpha > 1$ where

$$\varepsilon_{\text{step}} \leq \frac{2C^2}{(n-q)\sigma_{\text{cor}}^2 + \sigma_{\text{ind}}^2} \left[ 1 + \frac{\sigma_{\text{cor}}^2}{(f-q)\sigma_{\text{cor}}^2 + \sigma_{\text{ind}}^2} \right]. \tag{11}$$

Thus, following the composition property of RDP from Lemma A.3, the full Algorithm 1 satisfies $(\alpha, T\alpha\varepsilon_{\text{step}})$-SecRDP for any $\alpha > 1$. From Lemma A.3, we deduce that Algorithm 1 satisfies $(\varepsilon'(\alpha), \delta)$-SecLDP for any $\delta \in (0, 1)$ and any $\alpha > 1$, where

$$\varepsilon'(\alpha) = T\alpha\varepsilon_{\text{step}} + \frac{\log(1/\delta)}{\alpha - 1} \leq \alpha T \frac{2C^2}{(n-q)\sigma_{\text{cor}}^2 + \sigma_{\text{ind}}^2} \left[ 1 + \frac{\sigma_{\text{cor}}^2}{(f-q)\sigma_{\text{cor}}^2 + \sigma_{\text{ind}}^2} \right] + \frac{\log(1/\delta)}{\alpha - 1}.$$

Optimizing the above bound over $\alpha > 1$ yields the solution

$$\alpha_\star = 1 + \frac{\sqrt{\log(1/\delta)\left((n-q)\sigma_{\text{cor}}^2 + \sigma_{\text{ind}}^2\right)}}{C\sqrt{2T\left(1 + \frac{\sigma_{\text{cor}}^2}{(f-q)\sigma_{\text{cor}}^2 + \sigma_{\text{ind}}^2}\right)}},$$

which in turn leads to the following bound

$$\varepsilon_\star = \varepsilon'(\alpha_\star) \leq T \frac{2C^2}{(n-q)\sigma_{\text{cor}}^2 + \sigma_{\text{ind}}^2} \left[ 1 + \frac{\sigma_{\text{cor}}^2}{(f-q)\sigma_{\text{cor}}^2 + \sigma_{\text{ind}}^2} \right] + 2C\sqrt{2T \log(1/\delta) \frac{1 + \frac{\sigma_{\text{cor}}^2}{(f-q)\sigma_{\text{cor}}^2 + \sigma_{\text{ind}}^2}}{(n-q)\sigma_{\text{cor}}^2 + \sigma_{\text{ind}}^2}}.$$

Now, recall that we assume

$$\frac{1}{(n-q)\sigma_{\text{cor}}^2 + \sigma_{\text{ind}}^2} \left[ 1 + \frac{\sigma_{\text{cor}}^2}{(f-q)\sigma_{\text{cor}}^2 + \sigma_{\text{ind}}^2} \right] \leq \frac{\varepsilon^2}{16C^2 T \log(1/\delta)}.$$

Therefore, using the assumption $\varepsilon \leq \log(1/\delta)$, Algorithm 1 satisfies $(\varepsilon_\star, \delta)$-SecLDP where

$$\varepsilon_\star \leq \frac{\varepsilon^2}{8\log(1/\delta)} + \frac{\varepsilon}{\sqrt{2}} \leq \varepsilon.$$

This concludes the proof. $\square$

# B. Robustness Analysis: Proof of Proposition 4.1

**Proposition 4.1.** *Let $0 \leq f < \frac{n}{2}$ and $\kappa := \frac{6f}{n-f}\left(1 + \frac{f}{n-2f}\right)^2$. Let $x_1, \ldots, x_n \in \mathbb{R}^d$, and $S \subseteq [n]$ be any subset of size $n - f$. Let $\overline{x}_S := \frac{1}{|S|}\sum_{i \in S} x_i$, and $\lambda_{\max}$ denote the maximum eigenvalue operator. The output $\hat{x} = \mathrm{CAF}(x_1, \ldots, x_n)$ satisfies:*

$$\|\hat{x} - \overline{x}_S\|^2 \leq \kappa \cdot \lambda_{\max}\left(\frac{1}{|S|}\sum_{i \in S}(x_i - \overline{x}_S)(x_i - \overline{x}_S)^\top\right).$$

*Proof.* Let $n \geq 1, 0 \leq f < n/2$, $x_1, \ldots, x_n \in \mathbb{R}^n$, and $S \subseteq [n], |S| = n - f$. Denote $\overline{x}_S := \frac{1}{|S|}\sum_{i \in S} x_i$ and $\sigma_S^2 := \lambda_{\max}\left(\frac{1}{|S|}\sum_{i \in S}(x_i - \overline{x}_S)(x_i - \overline{x}_S)^\top\right)$.

First, we remark that Algorithm 2 terminates after at most $2f$ iterations. Indeed, at each iteration, at least one weight $c_i, i \in [n]$, becomes zero—this is true for any $i \in [n]$ such that $i \in \mathrm{argmax}_{j \in [n]} \langle \hat{v}_c, x_j - \hat{\mu}_c\rangle^2$—for the remaining iterations. Therefore, since the weights $c_i, i \in [n]$, are all in $[0, 1]$, it is guaranteed that $\sum_{i=1}^n c_i \leq n - 2f$ after $2f$ iterations.

Second, we show that there exists an iteration such that the maximum eigenvalue $\hat{\sigma}_c^2$ is at most $\frac{2n(n-f)}{(n-2f)^2}\sigma_S^2$. For the sake of contradiction, assume that for every iteration we have $\hat{\sigma}_c^2 > \frac{2n(n-f)}{(n-2f)^2}\sigma_S^2$. It is well-known from the analysis of the original Filter algorithm (Diakonikolas et al., 2017; Steinhardt et al., 2018) (see (Zhu et al., 2022, Appendix E.3) for details) that the following invariant thus holds:

$$\sum_{i \in S}(1 - c_i) \leq \sum_{i \in [n]\setminus S}(1 - c_i). \tag{12}$$

That is, the mass removed from the points in $S$ is never greater than that removed from the points outside $S$. Importantly, manipulating the bound above yields:

$$\sum_{i=1}^n c_i \geq n - 2f + 2\sum_{i \in [n]\setminus S} c_i \geq n - 2f. \tag{13}$$

This is exactly the opposite of the termination condition, i.e., the algorithm does not terminate. This is a contradiction, since we have shown first that the algorithm terminates. Therefore, we have shown that there exists an iteration such that the maximum eigenvalue $\hat{\sigma}_c^2$ is at most $\frac{2n(n-f)}{(n-2f)^2}\sigma_S^2$.

Finally, recall that the output $\mu_\star$ is the weighted mean of the original inputs with respect to weights corresponding to $\sigma_\star^2$ the smallest maximum eigenvalue across the algorithm. Thus, thanks to the last shown result, we have $\sigma_\star^2 \leq \frac{2n(n-f)}{(n-2f)^2}\sigma_S^2$. Therefore, using (Zhu et al., 2022, Lemma 2.2), we have

$$\|\mu_\star - \overline{x}_S\| \leq \sqrt{\frac{f}{n-f}}(\sigma_\star + \sigma_S) \leq \sqrt{\frac{f}{n-f}}\left(1 + \frac{\sqrt{2n(n-f)}}{n-2f}\right)\sigma_S. \tag{14}$$

We conclude by taking squares and using Jensen's inequality. $\qquad\square$

CAF draws inspiration from recent algorithmic advances in robust statistics (Diakonikolas et al., 2017). However, a key distinction is that CAF does not rely on knowing a bound on the covariance of honest inputs, which has been a crucial requirement in the design of efficient robust mean estimators (Diakonikolas et al., 2017; Steinhardt et al., 2018). In the aforementioned literature, knowing such a bound allows for an alternative termination condition in Algorithm 2, where the algorithm halts once the weighted covariance is sufficiently small relative to the known bound, and the final output is computed as the mean weighted by the last iteration's computed weights (Diakonikolas et al., 2017).

# C. Utility Analysis: Proof of Theorem 4.2 and Corollary 4.1

## C.1. Preliminaries

We recall that we assume the loss function to be differentiable everywhere, throughout the paper.

**Definition C.1** (L-smoothness). *A function $\mathcal{L}\colon \mathbb{R}^d \to \mathbb{R}$ is L-smooth if, for all $\theta, \theta' \in \mathbb{R}^d$, we have*

$$\mathcal{L}(\theta') - \mathcal{L}(\theta) - \langle \nabla \mathcal{L}(\theta), \theta' - \theta \rangle \leq \frac{L}{2} \left\| \theta' - \theta \right\|^2.$$

The above is equivalent to, for all $\theta, \theta' \in \mathbb{R}^d$, having $\|\nabla \mathcal{L}(\theta') - \nabla \mathcal{L}(\theta)\| \leq L \|\theta' - \theta\|$ (see, e.g., (Nesterov et al., 2018)).

**Definition C.2** ($\mu$-strong convexity). *A function $\mathcal{L}\colon \mathbb{R}^d \to \mathbb{R}$ is $\mu$-stongly convex if, for all $\theta, \theta' \in \mathbb{R}^d$, we have*

$$\mathcal{L}(\theta') - \mathcal{L}(\theta) - \langle \nabla \mathcal{L}(\theta), \theta' - \theta \rangle \geq \frac{\mu}{2} \left\| \theta' - \theta \right\|^2.$$

Moreover, we recall that strong convexity implies the Polyak-Lojasiewicz (PL) inequality (Karimi et al., 2016) $2\mu \left(\mathcal{L}(\theta) - \mathcal{L}_\star\right) \leq \|\nabla \mathcal{L}(\theta)\|^2$. Note that a function satisfies $L$-smoothness and $\mu$-strong convexity inequality simultaneously only if $\mu \leq L$.

## C.2. Proof Outline

Our analysis of CAFCOR (Algorithm 1), is inspired from standard analyses in non-private robust distributed learning (Karimireddy et al., 2021; Farhadkhani et al., 2022; Allouah et al., 2023c). We include all proofs for completeness.

**Notation.** Recall that for each step $t$, for each honest worker $w_i$,

$$m_t^{(i)} = \beta_{t-1} m_{t-1}^{(i)} + (1 - \beta_{t-1}) \tilde{g}_t^{(i)}, \tag{15}$$

$$\tilde{g}_t^{(i)} = g_t^{(i)} + \sum_{\substack{j=1 \\ j \neq i}}^{n} v_t^{(ij)} + \overline{v}_t^{(i)}; \quad \overline{v}_t^{(i)} \sim \mathcal{N}(0, \sigma_{\mathrm{ind}}^2 I_d), v_t^{(ij)} = -v_t^{(ji)} \sim \mathcal{N}(0, \sigma_{\mathrm{cor}}^2 I_d), \tag{16}$$

where we initialize $m_0^{(i)} = 0$. Recall that we denote

$$R_t \coloneqq \mathrm{CAF}\left(m_t^{(1)}, \dots, m_t^{(n)}\right), \tag{17}$$

$$\theta_{t+1} = \theta_t - \gamma_t R_t. \tag{18}$$

Throughout, we denote by $\mathcal{P}_t$ the history from steps $0$ to $t \in \{0, \dots, T-1\}$: $\mathcal{P}_t \coloneqq \left\{\theta_0, \dots, \theta_t; m_1^{(i)}, \dots, m_{t-1}^{(i)}; i \in [n]\right\}$. By convention, $\mathcal{P}_1 = \{\theta_0\}$. We denote by $\mathbb{E}_t[\cdot]$ and $\mathbb{E}[\cdot]$ the conditional expectation $\mathbb{E}[\cdot \mid \mathcal{P}_t]$ and the total expectation, respectively. Thus, $\mathbb{E}[\cdot] = \mathbb{E}_1[\cdots \mathbb{E}_T[\cdot]]$.

**Momentum drift.** Along the trajectory $\theta_0, \dots, \theta_t$, the honest workers' local momentums may drift away from each other. The drift has three distinct sources: (i) noise injected by the correlated noise scheme, (ii) gradient dissimilarity induced by data heterogeneity, and (iii) stochasticity of the mini-batch gradients. The aforementioned drift of local momentums can be exploited by the adversaries to maliciously bias the aggregation output. Formally, we will control the growth of the drift $\Delta_t$ between momentums, which we define as

$$\Delta_t \coloneqq \lambda_{\max}\left(\frac{1}{|\mathcal{H}|} \sum_{i \in \mathcal{H}} (m_t^{(i)} - \overline{m}_t)(m_t^{(i)} - \overline{m}_t)^\top\right), \tag{19}$$

where $\lambda_{\max}$ denotes the maximum eigenvalue, and $\overline{m}_t \coloneqq \frac{1}{|\mathcal{H}|} \sum_{i \in \mathcal{H}} m_t^{(i)}$ denotes the average honest momentum. We show in Lemma C.1 below that the growth of the drift $\Delta_t$ of the momentums can be controlled by tuning the momentum coefficient $\beta_t$. The full proof can be found in Appendix C.6.2.

**Lemma C.1.** *Suppose that assumptions 2.2 and 2.3 hold. Consider Algorithm 1. For every $t \in \{0, \dots, T-1\}$, we have*

$$\mathbb{E}[\Delta_{t+1}] \leq \beta_t \, \mathbb{E}[\Delta_t] + 2(1 - \beta_t)^2 \left( \sigma_b^2 + 108 \left( n\sigma_{\text{cor}}^2 + \sigma_{\text{ind}}^2 \right) \left( 1 + \frac{d}{n-f} \right) \right) + (1 - \beta_t) G_{\text{cov}}^2,$$

*where $\overline{m}_t := \frac{1}{|\mathcal{H}|} \sum_{i \in \mathcal{H}} m_t^{(i)}$, $\sigma_b^2 := 2(1 - \frac{b}{m})\frac{\sigma^2}{b}$, and $G_{\text{cov}}^2 := \sup_{\theta \in \mathbb{R}^d} \sup_{\|v\| \leq 1} \frac{1}{|\mathcal{H}|} \sum_{i \in \mathcal{H}} \langle v, \nabla \mathcal{L}_i(\theta) - \nabla \mathcal{L}_{\mathcal{H}}(\theta) \rangle^2$.*

The dimension factor $d$ due to correlated and uncorrelated noises is divided by $n - f$, which would not have been possible without leveraging the Gaussian nature of the noise. To do this, we use a concentration argument on the empirical covariance matrix of Gaussian random variables, stated in Lemma C.6. Moreover, the correlated noise magnitude $\sigma_{\text{cor}}^2$ has a multiplicative factor $n$, which essentially reflects the fact that a correlated noise term is added per worker, for each worker. Finally, the remaining term $G_{\text{cov}}^2$ of the upper bound is only due to data heterogeneity, and also appears in Allouah et al. (2023c). Similar to the aforementioned work, an important distinction from Karimireddy et al. (2022) is that $G_{\text{cov}}^2$ is a tighter bound on heterogeneity, compared to $G^2$ the bound on the average squared distance from Assumption 2.1. This is because the drift $\Delta_t$ is not an average squared distance, but rather a bound on average squared distances of every projection on the unit ball. Controlling this quantity requires a covering argument (stated in Lemma C.4).

**Momentum deviation.** Next, we study the momentum deviation; i.e., the distance between the average honest momentum $\overline{m}_t$ and the true gradient $\nabla \mathcal{L}_{\mathcal{H}}(\theta_t)$ in an arbitrary step $t$. Specifically, we define momentum *deviation* to be

$$\delta_t := \overline{m}_t - \nabla \mathcal{L}_{\mathcal{H}}(\theta_t). \tag{20}$$

Also, we introduce the error between the aggregate $R_t$ and $\overline{m}_t := \frac{1}{|\mathcal{H}|} \sum_{i \in \mathcal{H}} m_t^{(i)}$ the average momentum of honest workers for the case. Specifically, when defining the error

$$\epsilon_t := R_t - \overline{m}_t, \tag{21}$$

we get the following bound on the momentum deviation in Lemma C.2, proof of which can be found in Appendix C.6.3.

**Lemma C.2.** *Suppose that assumptions 2.2 and 2.3 hold and that $\mathcal{L}_{\mathcal{H}}$ is L-smooth. Consider Algorithm 1. For all $t \in \{0, \dots, T-1\}$, we have*

$$\mathbb{E}\left[\|\delta_{t+1}\|^2\right] \leq \beta_t^2 (1 + \gamma_t L)(1 + 4\gamma_t L) \, \mathbb{E}\left[\|\delta_t\|^2\right] + 4\gamma_t L(1 + \gamma_t L)\beta_t^2 \, \mathbb{E}\left[\|\nabla \mathcal{L}_{\mathcal{H}}(\theta_t)\|^2\right]$$

$$+ (1 - \beta_t)^2 \frac{\overline{\sigma}_{\text{DP}}^2}{n - f} + 2\gamma_t L(1 + \gamma_t L)\beta_t^2 \, \mathbb{E}\left[\|\epsilon_t\|^2\right],$$

*where $\overline{\sigma}_{\text{DP}}^2 := \left(1 - \frac{b}{m}\right)\frac{\sigma^2}{b} + df\sigma_{\text{cor}}^2 + d\sigma_{\text{ind}}^2$.*

Above, the error due to the correlated noise scheme involves the noise magnitude $\sigma_{\text{cor}}^2$ but only multiplied by the number of malicious workers $f$. This essentially reflects the fact that all correlated noise terms cancel out upon averaging, except those shared with malicious workers, since the latter do not follow the protocol in the worst case.

**Descent bound.** Finally, we bound the progress made at each learning step in minimizing the loss $\mathcal{L}_{\mathcal{H}}$ using Algorithm 1, as in previous work in robust distributed optimization. From (18) and (17), we obtain that, for each step $t$, $\theta_{t+1} = \theta_t - \gamma_t R_t = \theta_t - \gamma_t \overline{m}_t - \gamma_t(R_t - \overline{m}_t)$, Furthermore, by (21), $R_t - \overline{m}_t = \epsilon_t$. Thus, for all $t$,

$$\theta_{t+1} = \theta_t - \gamma_t \overline{m}_t - \gamma_t \epsilon_t. \tag{22}$$

This means that Algorithm 1 can actually be treated as distributed SGD with a momentum term that is subject to perturbation proportional to $\epsilon_t$ at each step $t$. This perspective leads us to Lemma C.3, proof of which can be found in Appendix C.6.4.

**Lemma C.3.** *Assume that $\mathcal{L}_{\mathcal{H}}$ is L-smooth. Consider Algorithm 1. For any $t \in [T]$, we have*

$$\mathbb{E}\left[\mathcal{L}_{\mathcal{H}}(\theta_{t+1}) - \mathcal{L}_{\mathcal{H}}(\theta_t)\right] \leq -\frac{\gamma_t}{2}(1 - 4\gamma_t L) \, \mathbb{E}\left[\|\nabla \mathcal{L}_{\mathcal{H}}(\theta_t)\|^2\right] + \gamma_t(1 + 2\gamma_t L) \, \mathbb{E}\left[\|\delta_t\|^2\right] + \gamma_t(1 + \gamma_t L) \, \mathbb{E}\left[\|\epsilon_t\|^2\right].$$

Putting all of the previous lemmas together, we prove Theorem 4.2 in Section C.3.

## C.3. Proof of Theorem 4.2

We recall the theorem statement below for convenience. We denote

$$\mathcal{L}_\star = \inf_{\theta \in \mathbb{R}^d} \mathcal{L}_\mathcal{H}(\theta), \mathcal{L}_0 = \mathcal{L}_\mathcal{H}(\theta_0) - \mathcal{L}_\star, a_1 = 240, a_2 = 480, a_3 = 5760, \text{ and } a_4 = 270.$$

**Theorem 4.2.** *Suppose that Assumptions 2.1, 2.2, and 2.3 hold, and that $\mathcal{L}_\mathcal{H}$ is $L$-smooth. Recall $\kappa$ from (4.1). Let*

$$\overline{\sigma}^2 := \frac{\sigma_b^2 + d(f\sigma_{\text{cor}}^2 + \sigma_{\text{ind}}^2)}{n - f}$$

$$+ 4\kappa\left(\sigma_b^2 + 108\left(n\sigma_{\text{cor}}^2 + \sigma_{\text{ind}}^2\right)\left(1 + \frac{d}{n-f}\right)\right),$$

*where $\sigma_b^2 := 2(1 - \frac{b}{m})\frac{\sigma^2}{b}$. We also denote $\mathcal{L}_\star := \inf_{\theta \in \mathbb{R}^d} \mathcal{L}_\mathcal{H}(\theta), \mathcal{L}_0 := \mathcal{L}_\mathcal{H}(\theta_0) - \mathcal{L}_\star$. Consider Algorithm 1 with $T \geq 1$, the learning rates $\gamma_t$ specified below, and momentum coefficients $\beta_t = 1 - 24L\gamma_t$. The following holds:*
1. *$\mathcal{L}_\mathcal{H}$ is $\mu$-strongly convex: If $\gamma_t = \frac{10}{\mu(t + 240\frac{L}{\mu})}$, then*

$$\mathbb{E}\left[\mathcal{L}_\mathcal{H}(\theta_T) - \mathcal{L}_*\right] \lesssim \frac{\kappa G_{\text{cov}}^2}{\mu} + \frac{L\overline{\sigma}^2}{\mu^2 T} + \frac{L^2 \mathcal{L}_0}{\mu^2 T^2},$$

2. *$\mathcal{L}_\mathcal{H}$ is non-convex: If $\gamma_t = \min\left\{\frac{1}{24L}, \frac{\sqrt{3\mathcal{L}_0}}{16\overline{\sigma}\sqrt{LT}}\right\}$, then*

$$\mathbb{E}\|\nabla\mathcal{L}_\mathcal{H}(\hat{\theta})\|^2 \lesssim \kappa G_{\text{cov}}^2 + \frac{\overline{\sigma}\sqrt{L\mathcal{L}_0}}{\sqrt{T}} + \frac{L\mathcal{L}_0}{T},$$

*where $\lesssim$ denotes inequality up to absolute constants and the expectation is over the randomness of the algorithm.*

We prove Theorem 4.2 in the strongly convex case in Section C.3.1, and in the non-convex case in Section C.3.2. These proofs follow along the lines of Allouah et al. (2023c), and we include them here for completeness.

### C.3.1. STRONGLY CONVEX CASE

*Proof.* Let Assumption 2.2 hold and assume that $\mathcal{L}_\mathcal{H}$ is $L$-smooth and $\mu$-strongly convex. Let $t \in \{0, \ldots, T-1\}$. We set the learning rate and momentum schedules to be

$$\gamma_t = \frac{10}{\mu(t + a_1\frac{L}{\mu})}, \ \beta_t = 1 - 24L\gamma_t, \tag{23}$$

where $a_1 := 240$. Note that we have

$$\gamma_t \leq \gamma_0 = \frac{10}{\mu 240\frac{L}{\mu}} = \frac{1}{24L}. \tag{24}$$

To obtain the convergence result we define the Lyapunov function to be

$$V_t := \left(t + a_1\frac{L}{\mu}\right)^2 \mathbb{E}\left[\mathcal{L}_\mathcal{H}(\theta_t) - \mathcal{L}_\star + \frac{z_1}{L}\|\delta_t\|^2 + \kappa \cdot \frac{z_2}{L}\Delta_t\right], \tag{25}$$

where $a_1 = 240, z_1 = \frac{1}{16}$, and $z_2 = 2$. Throughout the proof, we denote $\hat{t} := t + a_1\frac{L}{\mu}$. Therefore, we have $\gamma_t = \frac{10}{\mu\hat{t}}$. Consider also the auxiliary sequence $W_t$ defined as

$$W_t := \mathbb{E}\left[\mathcal{L}_\mathcal{H}(\theta_t) - \mathcal{L}_\star + \frac{z_1}{L}\|\delta_t\|^2 + \kappa \cdot \frac{z_2}{L}\Delta_t\right]. \tag{26}$$

Therefore, we have

$$V_{t+1} - V_t = (\hat{t} + 1)^2 W_{t+1} - \hat{t}^2 W_t = (\hat{t} + 1)^2 W_{t+1} - (\hat{t}^2 + 2\hat{t} + 1)W_t + (2\hat{t} + 1)W_t$$
$$= (\hat{t} + 1)^2(W_{t+1} - W_t) + (2\hat{t} + 1)W_t. \tag{27}$$

We now bound the quantity $W_{t+1} - W_t$ below.

**Invoking Lemma C.1.** Upon substituting from Lemma C.1, we obtain

$$\mathbb{E}\left[\kappa \cdot \frac{z_2}{L}\Delta_{t+1} - \kappa \cdot \frac{z_2}{L}\Delta_t\right] \leq \kappa \cdot \frac{z_2}{L}\beta_t \mathbb{E}\left[\Delta_t\right] + 2\kappa \cdot \frac{z_2}{L}(1-\beta_t)^2\left(\sigma_b^2 + 108\left(n\sigma_{\mathrm{cor}}^2 + \sigma_{\mathrm{ind}}^2\right)(1+\frac{d}{n-f})\right)$$
$$+ \kappa \cdot \frac{z_2}{L}(1-\beta_t)G_{cov}^2 - \kappa \cdot \frac{z_2}{L}\mathbb{E}\left[\Delta_t\right]. \tag{28}$$

**Invoking Lemma C.2.** Upon substituting from Lemma C.2, we obtain

$$\mathbb{E}\left[\frac{z_1}{L}\|\delta_{t+1}\|^2 - \frac{z_1}{L}\|\delta_t\|^2\right] \leq \frac{z_1}{L}\beta_t^2 c_t \mathbb{E}\left[\|\delta_t\|^2\right] + 4z_1\gamma_t(1+\gamma_t L)\beta_t^2 \mathbb{E}\left[\|\nabla\mathcal{L}_{\mathcal{H}}(\theta_t)\|^2\right] + \frac{z_1}{L}(1-\beta_t)^2\frac{\overline{\sigma}_{\mathrm{DP}}^2}{n-f}$$
$$+ 2z_1\gamma_t(1+\gamma_t L)\beta_t^2 \mathbb{E}\left[\|\epsilon_t\|^2\right] - \frac{z_1}{L}\mathbb{E}\left[\|\delta_t\|^2\right], \tag{29}$$

where we introduced the following quantity for simplicity

$$c_t = (1+\gamma_t L)(1+4\gamma_t L) = 1 + 5\gamma_t L + 4\gamma_t^2 L^2. \tag{30}$$

**Invoking Lemma C.3.** Substituting from Lemma C.3, we obtain

$$\mathbb{E}\left[\mathcal{L}_{\mathcal{H}}(\theta_{t+1}) - \mathcal{L}_{\mathcal{H}}(\theta_t)\right] \leq -\frac{\gamma_t}{2}(1-4\gamma_t L)\mathbb{E}\left[\|\nabla\mathcal{L}_{\mathcal{H}}(\theta_t)\|^2\right] + \gamma_t(1+2\gamma_t L)\mathbb{E}\left[\|\delta_t\|^2\right] + \gamma_t(1+\gamma_t L)\mathbb{E}\left[\|\epsilon_t\|^2\right]. \tag{31}$$

Substituting from (28), (29) and (31) in (26), we obtain

$$W_{t+1} - W_t = \mathbb{E}\left[\mathcal{L}_{\mathcal{H}}(\theta_{t+1}) - \mathcal{L}_{\mathcal{H}}(\theta_t)\right] + \mathbb{E}\left[\frac{z_1}{L}\|\delta_{t+1}\|^2 - \frac{z_1}{L}\|\delta_t\|^2\right] + \mathbb{E}\left[\kappa \cdot \frac{z_2}{L}\Delta_{t+1} - \kappa \cdot \frac{z_2}{L}\Delta_t\right]$$
$$\leq -\frac{\gamma_t}{2}(1-4\gamma_t L)\mathbb{E}\left[\|\nabla\mathcal{L}_{\mathcal{H}}(\theta_t)\|^2\right] + \gamma_t(1+2\gamma_t L)\mathbb{E}\left[\|\delta_t\|^2\right] + \gamma_t(1+\gamma_t L)\mathbb{E}\left[\|\epsilon_t\|^2\right]$$
$$+ \frac{z_1}{L}\beta_t^2 c_t \mathbb{E}\left[\|\delta_t\|^2\right] + 4z_1\gamma_t(1+\gamma_t L)\beta_t^2 \mathbb{E}\left[\|\nabla\mathcal{L}_{\mathcal{H}}(\theta_t)\|^2\right] + \frac{z_1}{L}(1-\beta_t)^2\frac{\overline{\sigma}_{\mathrm{DP}}^2}{n-f}$$
$$+ 2z_1\gamma_t(1+\gamma_t L)\beta_t^2 \mathbb{E}\left[\|\epsilon_t\|^2\right] - \frac{z_1}{L}\mathbb{E}\left[\|\delta_t\|^2\right]$$
$$+ \kappa \cdot \frac{z_2}{L}\beta_t \mathbb{E}\left[\Delta_t\right] + 2\kappa \cdot \frac{z_2}{L}(1-\beta_t)^2\left(\sigma_b^2 + 108\left(n\sigma_{\mathrm{cor}}^2 + \sigma_{\mathrm{ind}}^2\right)(1+\frac{d}{n-f})\right) + \kappa \cdot \frac{z_2}{L}(1-\beta_t)G_{cov}^2$$
$$- \kappa \cdot \frac{z_2}{L}\mathbb{E}\left[\Delta_t\right]. \tag{32}$$

Upon rearranging the R.H.S. in (32) we obtain that

$$W_{t+1} - W_t \leq -\frac{\gamma_t}{2}\left((1-4\gamma_t L) - 8z_1(1+\gamma_t L)\beta_t^2\right)\mathbb{E}\left[\|\nabla\mathcal{L}_{\mathcal{H}}(\theta_t)\|^2\right] + \frac{z_1}{L}(1-\beta_t)^2\frac{\overline{\sigma}_{\mathrm{DP}}^2}{n-f}$$
$$- z_1\gamma_t\left(-\frac{1}{z_1}(1+2\gamma_t L) - \frac{1}{\gamma_t L}\beta_t^2 c_t + \frac{1}{\gamma_t L}\right)\mathbb{E}\left[\|\delta_t\|^2\right] + \gamma_t\left(1+\gamma_t L + 2z_1(1+\gamma_t L)\beta_t^2\right)\mathbb{E}\left[\|\epsilon_t\|^2\right]$$
$$- \kappa \cdot \frac{z_2}{L}(1-\beta_t)\mathbb{E}\left[\Delta_t\right] + 2\kappa \cdot \frac{z_2}{L}(1-\beta_t)^2\left(\sigma_b^2 + 108\left(n\sigma_{\mathrm{cor}}^2 + \sigma_{\mathrm{ind}}^2\right)(1+\frac{d}{n-f})\right) + \kappa \cdot \frac{z_2}{L}(1-\beta_t)G_{cov}^2. \tag{33}$$

By Proposition 4.1, we can bound $\mathbb{E}\left[\|\epsilon_t\|^2\right]$ as follows. Starting from the definition of $\epsilon_t$, we have

$$\|\epsilon_t\|^2 = \|R_t - \overline{m}_t\|^2 = \left\|\mathrm{CAF}(m_t^{(1)}, \ldots, m_t^{(n)}) - \overline{m}_t\right\|^2 \leq \kappa \cdot \lambda_{\max}\left(\frac{1}{|\mathcal{H}|}\sum_{i\in\mathcal{H}}(m_t^{(i)} - \overline{m}_t)(m_t^{(i)} - \overline{m}_t)^\top\right) = \kappa \cdot \Delta_t.$$

Then taking total expectations above gives the bound

$$\mathbb{E}\left[\|\epsilon_t\|^2\right] \leq \kappa \cdot \mathbb{E}\left[\Delta_t\right].$$

Using the bound above in Equation (33), and then rearranging terms, yields

$$W_{t+1} - W_t \le -\frac{\gamma_t}{2}\left((1 - 4\gamma_t L) - 8z_1(1 + \gamma_t L)\beta_t^2\right)\mathbb{E}\left[\|\nabla\mathcal{L}_{\mathcal{H}}(\theta_t)\|^2\right] + \frac{z_1}{L}(1 - \beta_t)^2\frac{\overline{\sigma}_{\mathrm{DP}}^2}{n - f}$$

$$- z_1\gamma_t\left(-\frac{1}{z_1}(1 + 2\gamma_t L) - \frac{1}{\gamma_t L}\beta_t^2 c_t + \frac{1}{\gamma_t L}\right)\mathbb{E}\left[\|\delta_t\|^2\right] + \kappa\gamma_t\left(1 + \gamma_t L + 2z_1(1 + \gamma_t L)\beta_t^2\right)\mathbb{E}\left[\Delta_t\right]$$

$$- \kappa \cdot \frac{z_2}{L}(1 - \beta_t)\mathbb{E}\left[\Delta_t\right] + 2\kappa \cdot \frac{z_2}{L}(1 - \beta_t)^2\left(\sigma_b^2 + 108\left(n\sigma_{\mathrm{cor}}^2 + \sigma_{\mathrm{ind}}^2\right)(1 + \frac{d}{n - f})\right) + \kappa \cdot \frac{z_2}{L}(1 - \beta_t)G_{cov}^2$$

$$= -\frac{\gamma_t}{2}\left((1 - 4\gamma_t L) - 8z_1(1 + \gamma_t L)\beta_t^2\right)\mathbb{E}\left[\|\nabla\mathcal{L}_{\mathcal{H}}(\theta_t)\|^2\right] + \frac{z_1}{L}(1 - \beta_t)^2\frac{\overline{\sigma}_{\mathrm{DP}}^2}{n - f}$$

$$- z_1\gamma_t\left(-\frac{1}{z_1}(1 + 2\gamma_t L) - \frac{1}{\gamma_t L}\beta_t^2 c_t + \frac{1}{\gamma_t L}\right)\mathbb{E}\left[\|\delta_t\|^2\right]$$

$$- \kappa z_2\gamma_t\left(\frac{1}{\gamma_t L}(1 - \beta_t) - \frac{1}{z_2}\left(1 + \gamma_t L + 2z_1(1 + \gamma_t L)\beta_t^2\right)\right)\mathbb{E}\left[\Delta_t\right]$$

$$+ 2\kappa \cdot \frac{z_2}{L}(1 - \beta_t)^2\left(\sigma_b^2 + 108\left(n\sigma_{\mathrm{cor}}^2 + \sigma_{\mathrm{ind}}^2\right)(1 + \frac{d}{n - f})\right) + \kappa \cdot \frac{z_2}{L}(1 - \beta_t)G_{cov}^2.$$

For simplicity, we define

$$A := \frac{1}{2}(1 - 4\gamma_t L) - 8z_1(1 + \gamma_t L)\beta_t^2, \tag{34}$$

$$B := -\frac{1}{z_1}(1 + 2\gamma_t L) - \frac{1}{\gamma_t L}\beta_t^2 c_t + \frac{1}{\gamma_t L}, \tag{35}$$

and

$$C := \frac{1}{\gamma_t L}(1 - \beta_t) - \frac{1}{z_2}\left(1 + \gamma_t L + 2z_1(1 + \gamma_t L)\beta_t^2\right), \tag{36}$$

Denote also

$$\overline{\sigma}^2 := \frac{\sigma_b^2 + d(f\sigma_{\mathrm{cor}}^2 + \sigma_{\mathrm{ind}}^2)}{n - f} + 4\kappa\left(\sigma_b^2 + 108\left(n\sigma_{\mathrm{cor}}^2 + \sigma_{\mathrm{ind}}^2\right)(1 + \frac{d}{n - f})\right).$$

Recall that, as $z_1 = \frac{1}{16}$ and $z_2 = 2$, and $\overline{\sigma}_{\mathrm{DP}}^2 = \sigma_b^2 + d(f\sigma_{\mathrm{cor}}^2 + \sigma_{\mathrm{ind}}^2)$, we have

$$\overline{\sigma}^2 \ge z_1\frac{\overline{\sigma}_{\mathrm{DP}}^2}{n - f} + 2\kappa \cdot z_2\left(\sigma_b^2 + 108\left(n\sigma_{\mathrm{cor}}^2 + \sigma_{\mathrm{ind}}^2\right)(1 + \frac{d}{n - f})\right).$$

Thus, substituting the above variables, we obtain

$$W_{t+1} - W_t \le -A\gamma_t\mathbb{E}\left[\|\nabla\mathcal{L}_{\mathcal{H}}(\theta_t)\|^2\right] - z_1 B\gamma_t\mathbb{E}\left[\|\delta_t\|^2\right] - \kappa \cdot z_2 C\gamma_t\mathbb{E}\left[\Delta_t\right]$$

$$+ \frac{1}{L}(1 - \beta_t)^2\overline{\sigma}^2 + \kappa \cdot \frac{z_2}{L}(1 - \beta_t)G_{cov}^2. \tag{37}$$

We now analyze below the terms $A$, $B$ and $C$ on the RHS of (37).

**Term $A$.** Recall from (24) that $\gamma_t \le \frac{1}{24L}$. Upon using this in (34), and the facts that $z_1 = \frac{1}{16}$ and $\beta_t^2 \le 1$, we obtain that

$$A \ge \frac{1}{2}(1 - 4\gamma_t L) - 8z_1(1 + \gamma_t L) \ge \frac{1}{2}(1 - 4 \times \frac{1}{24}) - \frac{8}{16}(1 + \frac{1}{24}) \ge \frac{1}{10}. \tag{38}$$

**Term $B$.** Substituting $c_t$ from (30) in (35) we obtain that

$$B = -\frac{1}{z_1}(1 + 2\gamma_t L) - \frac{1}{\gamma_t L}\beta_t^2\left(1 + 5\gamma_t L + 4\gamma_t^2 L^2\right) + \frac{1}{\gamma_t L}$$

$$= \frac{1}{\gamma_t L}\left(1 - \beta^2\right) - \frac{1}{z_1}\left(1 + 2\gamma_t L + 5z_1\beta_t^2 + 4z_1\beta_t^2\gamma_t L\right).$$

Using the facts that $\beta_t \leq 1$ and $\gamma_t \leq \frac{1}{24L}$, and then substituting $z_1 = \frac{1}{16}$ we obtain

$$B \geq \frac{1}{\gamma_t L}(1 - \beta_t^2) - 16\left(1 + \frac{2}{24} + \frac{5}{16} + \frac{4}{24 \times 16}\right) \geq \frac{1}{\gamma_t L}(1 - \beta_t^2) - 23 \geq \frac{1}{\gamma_t L}(1 - \beta_t) - 23 = 1. \qquad (39)$$

where the last equality follows from the fact that $1 - \beta_t = 24\gamma_t L$.

**Term $C$.** Substituting $z_1 = \frac{1}{16}, z_2 = 2$ in (36), and then using the facts that $\beta_t \leq 1$ and $\gamma_t \leq \frac{1}{24L}$, we obtain

$$C = \frac{1}{\gamma_t L}(1 - \beta_t) - \frac{1}{2}\left(1 + \gamma_t L + (2 \times 16)(1 + \gamma_t L)\beta_t^2\right) \geq \frac{1}{\gamma_t L}(1 - \beta_t) - \frac{1}{2}\left(1 + \frac{1}{24} + 32(1 + \frac{1}{24})\right)$$

$$\geq \frac{1}{\gamma_t L}(1 - \beta_t) - 18 = 6, \qquad (40)$$

where the last equality follows from the fact that $1 - \beta_t = 24\gamma_t L$.

**Combining terms $A$, $B$, and $C$.** Finally, substituting from (38), (39), and (40) in (37) (and recalling that $z_2 = 2$) we obtain that

$$W_{t+1} - W_t \leq -\frac{\gamma_t}{10} \mathbb{E}\left[\|\nabla \mathcal{L}_{\mathcal{H}}(\theta_t)\|^2\right] - z_1 \gamma_t \mathbb{E}\left[\|\delta_t\|^2\right] - 6\kappa z_2 \gamma_t \mathbb{E}\left[\Delta_t\right]$$
$$+ \frac{1}{L}(1 - \beta_t)^2 \overline{\sigma}^2 + \kappa \cdot \frac{2}{L}(1 - \beta_t)G_{cov}^2. \qquad (41)$$

Since $\mathcal{L}_{\mathcal{H}}$ is $\mu$-strongly convex, we have (Karimi et al., 2016) for any $\theta \in \mathbb{R}^d$ that

$$\|\nabla \mathcal{L}_{\mathcal{H}}(\theta)\|^2 \geq 2\mu(\mathcal{L}(\theta) - \mathcal{L}_\star). \qquad (42)$$

Plugging (42) in (41) above, and then recalling that $L \geq \mu$, yields

$$W_{t+1} - W_t \leq -\frac{\mu \gamma_t}{5} \mathbb{E}\left[\mathcal{L}_{\mathcal{H}}(\theta_t) - \mathcal{L}_\star\right] - z_1 \gamma_t \mathbb{E}\left[\|\delta_t\|^2\right] - 6\kappa z_2 \gamma_t \mathbb{E}\left[\Delta_t\right]$$
$$+ \frac{1}{L}(1 - \beta_t)^2 \overline{\sigma}^2 + \kappa \cdot \frac{2}{L}(1 - \beta_t)G_{cov}^2$$
$$\leq -\frac{\mu \gamma_t}{5} \mathbb{E}\left[\mathcal{L}_{\mathcal{H}}(\theta_t) - \mathcal{L}_\star + \frac{z_1}{\mu}\|\delta_t\|^2 + \kappa \cdot \frac{z_2}{\mu}\Delta_t\right] + \frac{1}{L}(1 - \beta_t)^2 \overline{\sigma}^2 + \kappa \cdot \frac{2}{L}(1 - \beta_t)G_{cov}^2$$
$$\leq -\frac{\mu \gamma_t}{5} \mathbb{E}\left[\mathcal{L}_{\mathcal{H}}(\theta_t) - \mathcal{L}_\star + \frac{z_1}{L}\|\delta_t\|^2 + \kappa \cdot \frac{z_2}{L}\Delta_t\right] + \frac{1}{L}(1 - \beta_t)^2 \overline{\sigma}^2 + \kappa \cdot \frac{2}{L}(1 - \beta_t)G_{cov}^2$$
$$= -\frac{\mu \gamma_t}{5} W_t + \frac{1}{L}(1 - \beta_t)^2 \overline{\sigma}^2 + \kappa \cdot \frac{2}{L}(1 - \beta_t)G_{cov}^2.$$

Upon plugging the above bound back in Equation (27), rearranging terms and substituting $1 - \beta_t = 24L\gamma_t$, we obtain

$$V_{t+1} - V_t \leq (\hat{t}+1)^2 \left[-\frac{\mu \gamma_t}{5} W_t + \frac{1}{L}(1 - \beta_t)^2 \overline{\sigma}^2 + \kappa \cdot \frac{2}{L}(1 - \beta_t)G_{cov}^2\right] + (2\hat{t} + 1)W_t$$
$$= -\left[(\hat{t}+1)^2 \frac{\mu \gamma_t}{5} - (2\hat{t}+1)\right]W_t + \frac{(\hat{t}+1)^2}{L}(24L\gamma_t)^2 \overline{\sigma}^2 + \kappa \cdot \frac{2(\hat{t}+1)^2}{L}(24L\gamma_t)G_{cov}^2.$$

Recall however that $\gamma_t = \frac{10}{\mu \hat{t}}$ as $\hat{t} = t + a_1 \frac{L}{\mu}$. Recall that we denote $a_1 = 24 \times 10 = 240$. Substituting $\gamma_t$ above yields

$$V_{t+1} - V_t \leq (\hat{t}+1)^2 \left[-\frac{\mu \gamma_t}{5} W_t + \frac{1}{L}(1 - \beta_t)^2 \overline{\sigma}^2 + \kappa \cdot \frac{2}{L}(1 - \beta_t)G_{cov}^2\right] + (2\hat{t} + 1)W_t$$
$$= -\left[2\frac{(\hat{t}+1)^2}{\hat{t}} - (2\hat{t}+1)\right]W_t + a_1^2 L \frac{(\hat{t}+1)^2}{\mu^2 \hat{t}^2} \overline{\sigma}^2 + 2a_1 \kappa \cdot \frac{(\hat{t}+1)^2}{\mu \hat{t}}G_{cov}^2.$$

Observe that $2\frac{(\hat{t}+1)^2}{\hat{t}} \geq 2(\hat{t}+1) > 2\hat{t}+1$, implying that the first term above is negative:

$$V_{t+1} - V_t \leq a_1^2 L \frac{(\hat{t}+1)^2}{\mu^2 \hat{t}^2} \overline{\sigma}^2 + 2a_1 \kappa \cdot \frac{(\hat{t}+1)^2}{\mu \hat{t}}G_{cov}^2.$$

Observe now that, as $\hat{t} = t + a_1 \frac{L}{\mu} \geq a_1 = 240$ (because $L \geq \mu$), we have $(\hat{t} + 1)^2 \leq (1 + \frac{1}{240})^2 \hat{t}^2 \leq 2\hat{t}^2$. Plugging this bound in the inequality above gives

$$V_{t+1} - V_t \leq \frac{2a_1^2 L}{\mu^2} \overline{\sigma}^2 + 4a_1 \kappa \cdot \frac{\hat{t}}{\mu} G_{cov}^2.$$

Therefore, we have for every $t \in \{0, \ldots, T-1\}$ that

$$V_{t+1} - V_0 = \sum_{k=0}^{t} (V_{k+1} - V_k) \leq (t+1) \frac{2a_1^2 L}{\mu^2} \overline{\sigma}^2 + \left( \sum_{k=0}^{t} \hat{k} \right) \frac{4a_1 \kappa}{\mu} G_{cov}^2.$$

Since $\sum_{k=0}^{t} \hat{k} = \sum_{k=0}^{t} (k + a_1 \frac{L}{\mu}) = \sum_{k=0}^{t} k + a_1 (t+1) \frac{L}{\mu} = \frac{t(t+1)}{2} + a_1 (t+1) \frac{L}{\mu}$, we obtain

$$V_{t+1} - V_0 = \sum_{k=0}^{t} (V_{k+1} - V_k) \leq (t+1) \frac{2a_1^2 L}{\mu^2} \overline{\sigma}^2 + \left( \frac{t(t+1)}{2} + a_1(t+1)\frac{L}{\mu} \right) \frac{4a_1 \kappa}{\mu} G_{cov}^2$$
$$= (t+1) \frac{2a_1^2 L}{\mu^2} \overline{\sigma}^2 + (t+1) \left( \frac{t}{2} + a_1 \frac{L}{\mu} \right) \frac{4a_1 \kappa}{\mu} G_{cov}^2.$$

However, recalling the definition (25) of $V_t$, we obtain

$$(t+1+a_1\frac{L}{\mu})^2 \, \mathbb{E}\left[ \mathcal{L}_{\mathcal{H}}(\theta_{t+1}) - \mathcal{L}_\star \right] \leq V_{t+1} \leq V_0 + (t+1)\frac{2a_1^2 L}{\mu^2}\overline{\sigma}^2 + (t+1)\left( \frac{t}{2} + a_1\frac{L}{\mu} \right)\frac{4a_1\kappa}{\mu}G_{cov}^2.$$

By rearranging terms, and using the fact that $\frac{L}{\mu} \geq 1$, we then get

$$\mathbb{E}\left[ \mathcal{L}_{\mathcal{H}}(\theta_{t+1}) - \mathcal{L}_\star \right] \leq \frac{V_0}{(t+1+a_1\frac{L}{\mu})^2} + \frac{t+1}{(t+1+a_1\frac{L}{\mu})^2}\frac{2a_1^2 L\overline{\sigma}^2}{\mu^2} + \frac{(t+1)\left( \frac{t}{2} + a_1\frac{L}{\mu} \right)}{(t+1+a_1\frac{L}{\mu})^2}\frac{4a_1\kappa}{\mu}G_{cov}^2$$
$$\leq \frac{V_0}{(t+1+a_1\frac{L}{\mu})^2} + \frac{1}{t+1+a_1\frac{L}{\mu}}\frac{2a_1^2 L\overline{\sigma}^2}{\mu^2} + \frac{4a_1\kappa}{\mu}G_{cov}^2. \tag{43}$$

It remains to bound $V_0$. By definition, we have

$$V_0 = \left( a_1\frac{L}{\mu} \right)^2 \left[ \mathcal{L}_{\mathcal{H}}(\theta_0) - \mathcal{L}_\star + \frac{z_1}{L}\|\delta_0\|^2 + \frac{z_2}{L}\Delta_0 \right].$$

By definition of $\overline{m}_t = \frac{1}{|\mathcal{H}|}\sum_{i \in \mathcal{H}} m_t^{(i)}$ and the initializations $m_0^{(i)} = 0$ for all $i \in \mathcal{H}$, we have $\Delta_0 = \lambda_{\max}\left( \frac{1}{|\mathcal{H}|}\sum_{i \in \mathcal{H}}(m_0^{(i)} - \overline{m}_0)(m_0^{(i)} - \overline{m}_0)^\top \right) = 0$. Therefore, we have

$$V_0 = \left( a_1\frac{L}{\mu} \right)^2 \left[ \mathcal{L}_{\mathcal{H}}(\theta_0) - \mathcal{L}_\star + \frac{z_1}{L}\|\delta_0\|^2 \right].$$

Moreover, by definition of $\delta_t$ in (20), we obtain that

$$\|\delta_0\|^2 = \|\overline{m}_0 - \nabla \mathcal{L}_{\mathcal{H}}(\theta_0)\|^2 = \|\nabla \mathcal{L}_{\mathcal{H}}(\theta_0)\|^2.$$

Recall that $\mathcal{L}_{\mathcal{H}}$ is $L$-smooth. Thus, $\|\nabla \mathcal{L}_{\mathcal{H}}(\theta_0)\|^2 \leq 2L(\mathcal{L}_{\mathcal{H}}(\theta_0) - \mathcal{L}^*)$ (see (Nesterov et al., 2018), Theorem 2.1.5). Therefore, substituting $z_1 = \frac{1}{16}$, we have

$$V_0 \leq \left( a_1\frac{L}{\mu} \right)^2 \left[ \mathcal{L}_{\mathcal{H}}(\theta_0) - \mathcal{L}_\star + \frac{2L}{16L}(\mathcal{L}_{\mathcal{H}}(\theta_0) - \mathcal{L}_\star) \right] =\leq \left( a_1\frac{L}{\mu} \right)^2 \frac{9}{8}(\mathcal{L}_{\mathcal{H}}(\theta_0) - \mathcal{L}_\star) \leq 2\left( a_1\frac{L}{\mu} \right)^2 (\mathcal{L}_{\mathcal{H}}(\theta_0) - \mathcal{L}_\star).$$

Plugging the above bound back in Equation (43), rearranging terms, and then recalling that $a_1 \frac{L}{\mu} \geq 0$, yields

$$\mathbb{E}\left[\mathcal{L}_{\mathcal{H}}(\theta_{t+1}) - \mathcal{L}_\star\right] \leq \frac{4a_1}{\mu}\kappa G_{cov}^2 + \frac{2a_1^2 L\overline{\sigma}^2}{\mu^2(t+1+a_1\frac{L}{\mu})} + \frac{2a_1 L^2(\mathcal{L}_{\mathcal{H}}(\theta_0) - \mathcal{L}_\star)}{\mu^2(t+1+a_1\frac{L}{\mu})^2}$$

$$\leq \frac{4a_1}{\mu}\kappa G_{cov}^2 + \frac{2a_1^2 L\overline{\sigma}^2}{\mu^2(t+1)} + \frac{2a_1 L^2(\mathcal{L}_{\mathcal{H}}(\theta_0) - \mathcal{L}_\star)}{\mu^2(t+1)^2}.$$

Specializing the inequality above for $t = T - 1$ and denoting $\mathcal{L}_0 := \mathcal{L}_{\mathcal{H}}(\theta_0) - \mathcal{L}_\star$ proves the theorem:

$$\mathbb{E}\left[\mathcal{L}_{\mathcal{H}}(\theta_T) - \mathcal{L}_\star\right] \leq \frac{4a_1}{\mu}\kappa G_{cov}^2 + \frac{2a_1^2 L\overline{\sigma}^2}{\mu^2 T} + \frac{2a_1^2 L^2 \mathcal{L}_0}{\mu^2 T^2}.$$

$\square$

### C.3.2. NON-CONVEX CASE

*Proof.* Let Assumption 2.2 hold and assume that $\mathcal{L}_{\mathcal{H}}$ is $L$-smooth. Let $t \in \{0, \ldots, T-1\}$. We set the learning rate and momentum to constant as follows:

$$\gamma_t = \gamma := \min\left\{\frac{1}{24L}, \frac{\sqrt{a_4 \mathcal{L}_0}}{2\overline{\sigma}\sqrt{a_3 LT}}\right\}, \ \beta_t = \beta := 1 - 24L\gamma, \tag{44}$$

where $a_1 := 240$. Note that we have

$$\gamma_t = \gamma \leq \frac{1}{24L}. \tag{45}$$

To obtain the convergence result we define the Lyapunov function to be

$$V_t := \mathbb{E}\left[\mathcal{L}_{\mathcal{H}}(\theta_t) - \mathcal{L}_\star + \frac{z_1}{L}\|\delta_t\|^2 + \kappa \cdot \frac{z_2}{L}\Delta_t\right], \tag{46}$$

where $z_1 = \frac{1}{16}$, and $z_2 = 2$. Note that $V_t$ corresponds to the sequence $W_t$ defined in Equation (26), and analyzed in Appendix C.3.1 under the assumption that $\gamma_t \leq \frac{1}{24L}$. Since the latter holds by Equation (45), we directly apply the bound obtained in Equation (41):

$$V_{t+1} - V_t \leq -\frac{\gamma_t}{10}\mathbb{E}\left[\|\nabla\mathcal{L}_{\mathcal{H}}(\theta_t)\|^2\right] - z_1\gamma_t\mathbb{E}\left[\|\delta_t\|^2\right] - 6\kappa z_2\gamma_t\mathbb{E}[\Delta_t]$$
$$+ \frac{1}{L}(1-\beta_t)^2\overline{\sigma}^2 + \kappa \cdot \frac{2}{L}(1-\beta_t)G_{cov}^2.$$

In turn, substituting $\gamma_t = \gamma, \beta_t = \beta$ and bounding the second and third terms on the RHS by zero, this implies that

$$V_{t+1} - V_t \leq -\frac{\gamma}{10}\mathbb{E}\left[\|\nabla\mathcal{L}_{\mathcal{H}}(\theta_t)\|^2\right] + \frac{1}{L}(1-\beta)^2\overline{\sigma}^2 + \kappa \cdot \frac{2}{L}(1-\beta)G_{cov}^2.$$

By rearranging terms and then averaging over $t \in \{0, \ldots, T-1\}$, we obtain

$$\frac{1}{T}\sum_{t=0}^{T-1}\mathbb{E}\left[\|\nabla\mathcal{L}_{\mathcal{H}}(\theta_t)\|^2\right] \leq \frac{10}{\gamma T}\sum_{t=0}^{T-1}(V_t - V_{t+1}) + \frac{10}{\gamma L}(1-\beta)^2\overline{\sigma}^2 + \kappa \cdot \frac{20}{\gamma L}(1-\beta)G_{cov}^2.$$

We now substitute $\beta = 1 - 24\gamma L$. Denoting $a_3 = 10 \times 24^2 = 5760, a_2 = 20 \times 24 = 480$, we obtain

$$\frac{1}{T}\sum_{t=0}^{T-1}\mathbb{E}\left[\|\nabla\mathcal{L}_{\mathcal{H}}(\theta_t)\|^2\right] \leq \frac{10}{\gamma T}\sum_{t=0}^{T-1}(V_t - V_{t+1}) + \frac{(10 \times 24^2)}{\gamma L}(\gamma L)^2\overline{\sigma}^2 + \kappa \cdot \frac{(20 \times 24)}{\gamma L}(\gamma L)G_{cov}^2$$

$$= \frac{10}{\gamma T}(V_0 - V_T) + a_3\gamma L\overline{\sigma}^2 + a_2\kappa G_{cov}^2. \tag{47}$$

We now bound $V_0 - V_T$. First recall that $V_T \geq 0$ as a sum of non-negative terms (see (46)). Therefore, we have

$$V_0 - V_T \leq V_0 = \mathcal{L}_{\mathcal{H}}(\theta_0) - \mathcal{L}_\star + \frac{z_1}{L} \|\delta_0\|^2 + \frac{z_2}{L} \Delta_0.$$

By definition of $\overline{m}_t = \frac{1}{|\mathcal{H}|} \sum_{i \in \mathcal{H}} m_t^{(i)}$ and the initializations $m_0^{(i)} = 0$ for all $i \in \mathcal{H}$, we have $\Delta_0 = \lambda_{\max}\left(\frac{1}{|\mathcal{H}|} \sum_{i \in \mathcal{H}} (m_0^{(i)} - \overline{m}_0)(m_0^{(i)} - \overline{m}_0)^\top\right) = 0$. Therefore, we have

$$V_0 = \mathcal{L}_{\mathcal{H}}(\theta_0) - \mathcal{L}_\star + \frac{z_1}{L} \|\delta_0\|^2.$$

Moreover, by definition of $\delta_t$ in (20), we obtain that

$$\|\delta_0\|^2 = \|\overline{m}_0 - \nabla\mathcal{L}_{\mathcal{H}}(\theta_0)\|^2 = \|\nabla\mathcal{L}_{\mathcal{H}}(\theta_0)\|^2.$$

Recall that $\mathcal{L}_{\mathcal{H}}$ is $L$-smooth. Thus, $\|\nabla\mathcal{L}_{\mathcal{H}}(\theta_0)\|^2 \leq 2L(\mathcal{L}_{\mathcal{H}}(\theta_0) - \mathcal{L}^*)$ (see (Nesterov et al., 2018), Theorem 2.1.5). Therefore, substituting $z_1 = \frac{1}{16}$, we have

$$V_0 - V_T \leq V_0 \leq \mathcal{L}_{\mathcal{H}}(\theta_0) - \mathcal{L}_\star + \frac{2L}{16L}(\mathcal{L}_{\mathcal{H}}(\theta_0) - \mathcal{L}_\star) = \frac{9}{8}(\mathcal{L}_{\mathcal{H}}(\theta_0) - \mathcal{L}_\star).$$

By plugging this bound back in (47), and denoting $a_4 := 24 \times 10 \times (\frac{9}{8}) = 270$ and $\mathcal{L}_0 := \mathcal{L}_{\mathcal{H}}(\theta_0) - \mathcal{L}_\star$, we obtain

$$\frac{1}{T}\sum_{t=0}^{T-1} \mathbb{E}\left[\|\nabla\mathcal{L}_{\mathcal{H}}(\theta_t)\|^2\right] \leq \frac{10 \times (\frac{9}{8})}{\gamma T}(\mathcal{L}_{\mathcal{H}}(\theta_0) - \mathcal{L}_\star) + a_3\gamma L\overline{\sigma}^2 + a_2\kappa G_{cov}^2$$

$$= \frac{a_4\mathcal{L}_0}{24\gamma T} + a_3\gamma L\overline{\sigma}^2 + a_2\kappa G_{cov}^2. \tag{48}$$

Recall that by definition

$$\gamma = \min\left\{\frac{1}{24L}, \frac{\sqrt{a_4\mathcal{L}_0}}{2\overline{\sigma}\sqrt{a_3 LT}}\right\},$$

and thus $\frac{1}{\gamma} = \max\left\{24L, \frac{2}{\sqrt{a_4\mathcal{L}_0}}\overline{\sigma}\sqrt{a_3 LT}\right\} \leq 24L + \frac{2}{\sqrt{a_4\mathcal{L}_0}}\overline{\sigma}\sqrt{a_3 LT}$. Therefore, we have

$$\frac{a_4\mathcal{L}_0}{24\gamma T} \leq \frac{a_4\mathcal{L}_0}{24T}\left(24L + \frac{2}{\sqrt{a_4\mathcal{L}_0}}\overline{\sigma}\sqrt{a_3 LT}\right) = \frac{a_4 L\mathcal{L}_0}{T} + \frac{\sqrt{a_3 a_4 L\mathcal{L}_0}\overline{\sigma}}{12\sqrt{T}}.$$

Upon using the above, and that $\gamma \leq \frac{\sqrt{a_4\mathcal{L}_0}}{2\overline{\sigma}\sqrt{a_3 LT}}$, in (48), we obtain that

$$\frac{1}{T}\sum_{t=0}^{T-1} \mathbb{E}\left[\|\nabla\mathcal{L}_{\mathcal{H}}(\theta_t)\|^2\right] \leq \frac{a_4 L\mathcal{L}_0}{T} + \frac{\sqrt{a_3 a_4 L\mathcal{L}_0}\overline{\sigma}}{12\sqrt{T}} + \frac{\sqrt{a_3 a_4 L\mathcal{L}_0}\overline{\sigma}}{2\sqrt{T}} + a_2\kappa G_{cov}^2 \leq a_2\kappa G_{cov}^2 + \frac{\sqrt{a_3 a_4 L\mathcal{L}_0}\overline{\sigma}}{\sqrt{T}} + \frac{a_4 L\mathcal{L}_0}{T}.$$

Finally, recall from Algorithm 1 that $\hat{\theta}$ is chosen randomly from the set of parameter vectors $(\theta_0, \ldots, \theta_{T-1})$. Thus, $\mathbb{E}\left[\left\|\nabla\mathcal{L}_{\mathcal{H}}\left(\hat{\theta}\right)\right\|^2\right] = \frac{1}{T}\sum_{t=0}^{T-1} \mathbb{E}\left[\|\nabla\mathcal{L}_{\mathcal{H}}(\theta_t)\|^2\right]$. Substituting this above proves the theorem. $\qquad\square$

## C.4. Proof of Corollary 4.1

We now state the proof of Corollary 4.1 below.

**Corollary 4.1.** *Let $\delta \in (0,1), \varepsilon \in (0, \log(1/\delta))$ and $n \geq (2+\eta)f$, for some absolute constant $\eta > 0$. Consider Algorithm 1 in the strongly convex setting of Theorem 4.2. If $\sigma_{cor}^2 = \sigma_{ind}^2 = \frac{32C^2 T \log(1/\delta)}{\varepsilon^2(n-f)}$, then Algorithm 1 is $(\varepsilon, \delta)$-SecLDP against an honest-but-curious server colluding with all malicious workers, and $(f, \varrho)$-robust where $\varrho = \mathcal{O}\left(\frac{(f+1)C^2 d \log(1/\delta)}{n^2\varepsilon^2} + \frac{fC^2 \log(1/\delta)}{n\varepsilon^2} + \frac{f}{n}G_{cov}^2\right)$, asymptotically in $T$ and ignoring absolute constants.*

*Proof.* The privacy claim follows directly from Theorem 4.1, given the choice of $\sigma_{\text{cor}}^2 = \sigma_{\text{ind}}^2 = \frac{32C^2 T \log(1/\delta)}{\varepsilon^2(n-f)}$.

We now prove the utility claim. First, we recall that $n - f \geq n/2$ since $f < n/2$, and $\kappa$ is bounded by $\frac{f}{n}$ up to a constant when $n \geq (2+\eta)f, \eta > 0$, by Proposition 4.1. Plugging the expressions of $\sigma_{\text{ind}}^2$ and $\sigma_{\text{cor}}^2$ in the strongly convex bound of Theorem 4.2 and rearranging terms yields

$$\mathbb{E}\left[\mathcal{L}_{\mathcal{H}}(\theta_T) - \mathcal{L}_\star\right] = \mathcal{O}(\frac{\kappa G_{\text{cov}}^2}{\mu} + \frac{L\overline{\sigma}^2}{\mu^2 T} + \frac{L^2 \mathcal{L}_0}{\mu^2 T^2})$$

$$= \mathcal{O}\left(\frac{\kappa G_{\text{cov}}^2}{\mu} + \frac{L}{\mu^2 T}\left[\frac{\sigma_b^2 + d(f\sigma_{\text{cor}}^2 + \sigma_{\text{ind}}^2)}{n - f} + \kappa\left(\sigma_b^2 + \left(n\sigma_{\text{cor}}^2 + \sigma_{\text{ind}}^2\right)\left(1 + \frac{d}{n-f}\right)\right)\right] + \frac{L^2 \mathcal{L}_0}{\mu^2 T^2}\right)$$

$$= \mathcal{O}\left(\frac{f G_{\text{cov}}^2}{n\mu} + \frac{C^2 d \log(1/\delta)}{\varepsilon^2 n} \cdot \frac{f+1}{n} + \frac{C^2 \log(1/\delta)}{\varepsilon^2} \cdot \frac{f}{n} + \frac{\sigma_b^2}{T} \cdot \frac{f+1}{n} + \frac{L^2 \mathcal{L}_0}{\mu^2 T^2}\right).$$

We conclude by ignoring asymptotically vanishing terms in $T$. $\qquad\square$

### C.5. Privacy-Utility Trade-off for Standard Aggregation Methods

We now analyze the privacy-utility trade-off of CAFCOR when using standard aggregation methods instead of CAF and present the corresponding proof below. Specifically, under the same setting as Corollary 4.1, but replacing CAF in Algorithm 1 with the coordinate-wise trimmed mean (proven order-optimal under the weaker robustness criterion of Allouah et al. (2023a))), we obtain $(f, \varrho)$-robustness with

$$\varrho = \mathcal{O}\left(\frac{f}{n} \cdot \frac{C^2 d \log(1/\delta)}{\varepsilon^2} + \frac{C^2 d \log(1/\delta)}{n^2 \varepsilon^2} + \frac{f}{n} d G_{\text{cov}}^2\right),$$

asymptotically in $T$ and ignoring absolute constants.

*Proof.* When using coordinate-wise trimmed mean, we have a weaker robustness guarantee than Proposition 4.1. Essentially, we can prove the latter but with the right-hand side of the inequality involving the trace instead of the maximum eigenvalue (of the empirical covariance of honest inputs), as shown by Allouah et al. (2023a). Therefore, the only change in the convergence analysis occurs in Lemma C.1, which would instead involve the trace operator instead of the maximum eigenvalue operator. Formally, denoting

$$\widetilde{\Delta}_t := \text{Tr}\left(\frac{1}{|\mathcal{H}|}\sum_{i\in\mathcal{H}}(m_t^{(i)} - \overline{m}_t)(m_t^{(i)} - \overline{m}_t)^\top\right) = \frac{1}{|\mathcal{H}|}\sum_{i\in\mathcal{H}}\left\|m_t^{(i)} - \overline{m}_t\right\|^2, \tag{49}$$

we would have

$$\mathbb{E}\left[\widetilde{\Delta}_{t+1}\right] \leq \beta_t \mathbb{E}\left[\widetilde{\Delta}_t\right] + 2(1-\beta_t)^2\left(\sigma_b^2 + 3d\left(n\sigma_{\text{cor}}^2 + \sigma_{\text{ind}}^2\right)\right) + (1-\beta_t)G_{\text{cov}}^2,$$

reusing the same notation as Lemma C.1. Therefore, leaving all other lemmas unchanged for the proof of Theorem 4.2, we would obtain the statement as the latter except that the expression of $\overline{\sigma}$ from Theorem 4.2 would become

$$\frac{\sigma_b^2 + d(f\sigma_{\text{cor}}^2 + \sigma_{\text{ind}}^2)}{n - f} + 4\kappa\left(\sigma_b^2 + 3d\left(n\sigma_{\text{cor}}^2 + \sigma_{\text{ind}}^2\right)\right).$$

We now follow the same steps as in the proof of Corollary 4.1. First, we recall that $n \geq (2+\eta)f$ for some constant value $\eta$, so that $\kappa$ is bounded by $f/n$ up to constants. Plugging the expressions of $\sigma_{\text{ind}}^2$ and $\sigma_{\text{cor}}^2$ in the strongly convex bound of Theorem 4.2 and rearranging terms yields

$$\mathbb{E}\left[\mathcal{L}_{\mathcal{H}}(\theta_T) - \mathcal{L}_\star\right] = \mathcal{O}(\frac{\kappa d G_{\text{cov}}^2}{\mu} + \frac{L\overline{\sigma}^2}{\mu^2 T} + \frac{L^2 \mathcal{L}_0}{\mu^2 T^2})$$

$$= \mathcal{O}\left(\frac{\kappa d G_{\text{cov}}^2}{\mu} + \frac{L}{\mu^2 T}\left[\frac{\sigma_b^2 + d(f\sigma_{\text{cor}}^2 + \sigma_{\text{ind}}^2)}{n - f} + \kappa\left(\sigma_b^2 + d\left(n\sigma_{\text{cor}}^2 + \sigma_{\text{ind}}^2\right)\right)\right] + \frac{L^2 \mathcal{L}_0}{\mu^2 T^2}\right)$$

$$= \mathcal{O}\left(\frac{f d G_{\text{cov}}^2}{n\mu} + \frac{C^2 d \log(1/\delta)}{\varepsilon^2 n} \cdot \frac{f+1}{n} + \frac{C^2 d \log(1/\delta)}{\varepsilon^2} \cdot \frac{f}{n} + \frac{\sigma_b^2}{T} \cdot \frac{f+1}{n} + \frac{L^2 \mathcal{L}_0}{\mu^2 T^2}\right)$$

$$= \mathcal{O}\left(\frac{f d G_{\text{cov}}^2}{n\mu} + \frac{C^2 d \log(1/\delta)}{\varepsilon^2 n^2} + \frac{C^2 d \log(1/\delta)}{\varepsilon^2} \cdot \frac{f}{n} + \frac{\sigma_b^2}{T} \cdot \frac{f+1}{n} + \frac{L^2 \mathcal{L}_0}{\mu^2 T^2}\right).$$

We conclude by ignoring asymptotically vanishing terms in $T$ and rearranging terms. □

## C.6. Proof of Supporting Lemmas

Before proving Lemmas C.1 to C.3 in Appendices C.6.2 to C.6.4, respectively, we first present some additional results in Appendix C.6.1 below. Most of the proofs follow along the lines of Allouah et al. (2023c), and we include them here for completeness.

### C.6.1. TECHNICAL LEMMAS

**Lemma C.4.** *Let $M \in \mathbb{R}^{d \times d}$ be a random real symmetric matrix and $g \colon \mathbb{R} \to \mathbb{R}$ an increasing function. It holds that*

$$\mathbb{E}\left[\sup_{\|v\| \leq 1} g(\langle v, \, Mv \rangle)\right] \leq 9^d \cdot \sup_{\|v\| \leq 1} \mathbb{E}\left[g(2 \langle v, \, Mv \rangle)\right].$$

*Proof.* Let $M \in \mathbb{R}^{d \times d}$ be a random real symmetric matrix and $g \colon \mathbb{R} \to \mathbb{R}$ a increasing function.

The proof follows the construction of (Section 5.2, (Vershynin, 2010)). Recall from standard covering net results (Vershynin, 2010) that we can construct $\mathcal{N}_{1/4}$ a finite $1/4$-net of the unit ball, i.e., for any vector $v$ in the unit ball, there exists $u_v \in \mathcal{N}_{1/4}$ such that $\|u_v - v\| \leq 1/4$. Moreover, we have the bound $|\mathcal{N}_{1/4}| \leq (1 + 2/(1/4))^d = 9^d$. Denote by $\|M\| := \sup_{\|v\| \leq 1} \|Mv\|$ the operator norm of $M$. By recalling that $M$ is symmetric, we obtain for any $v$ in the unit ball

$$|\langle v, \, Mv \rangle - \langle u_v, \, Mu_v \rangle| = |\langle v + u_v, \, M(v - u_v) \rangle| \leq \|v + u_v\| \|M(v - u_v)\| \leq (\|v\| + \|u_v\|) \|M(v - u_v)\|$$
$$\leq 2 \|M(v - u_v)\| \leq 2 \|M\| \|v - u_v\| \leq 2 \|M\| /4 = \|M\| /2.$$

Therefore, we have $\langle v, \, Mv \rangle - \langle u_v, \, Mu_v \rangle \leq \|M\| /2$, and $\langle v, \, Mv \rangle - \|M\| /2 \leq \langle u_v, \, Mu_v \rangle \leq \sup_{u \in \mathcal{N}_{1/4}} \langle u, \, Mu \rangle$. Recall that since $M$ is symmetric, its operator norm coincides with its maximum eigenvalue: $\|M\| = \sup_{\|v\| \leq 1} \langle v, \, Mv \rangle$. We therefore deduce that

$$\sup_{\|v\| \leq 1} \langle v, \, Mv \rangle \leq 2 \cdot \sup_{v \in \mathcal{N}_{1/4}} \langle v, \, Mv \rangle.$$

Upon composing with $g$, which is increasing, we get

$$\sup_{\|v\| \leq 1} g(\langle v, \, Mv \rangle) = g\left(\sup_{\|v\| \leq 1} \langle v, \, Mv \rangle\right) \leq g\left(2 \cdot \sup_{v \in \mathcal{N}_{1/4}} \langle v, \, Mv \rangle\right) = \sup_{v \in \mathcal{N}_{1/4}} g(2 \langle v, \, Mv \rangle).$$

Upon taking expectations and applying union bound, we finally conclude

$$\mathbb{E}\left[\sup_{\|v\| \leq 1} g(\langle v, \, Mv \rangle)\right] \leq \mathbb{E}\left[\sup_{v \in \mathcal{N}_{1/4}} g(2 \langle v, \, Mv \rangle)\right] \leq |\mathcal{N}_{1/4}| \cdot \sup_{v \in \mathcal{N}_{1/4}} \mathbb{E}\left[g(2 \langle v, \, Mv \rangle)\right] \leq 9^d \cdot \sup_{\|v\| \leq 1} \mathbb{E}\left[g(2 \langle v, \, Mv \rangle)\right].$$

□

**Lemma C.5.** *Suppose assumptions 2.2 and 2.3 hold. For any $t \in \{0, \ldots, T - 1\}$ and $i \in \mathcal{H}$, we have*

$$\mathbb{E}\left[\left\|\tilde{g}_t^{(i)} - \nabla \mathcal{L}_i(\theta_t)\right\|^2\right] \leq 2 \left(1 - \frac{b}{m}\right) \frac{\sigma^2}{b} + d \cdot \sigma_{\mathrm{DP}}^2.$$

*Proof.* Suppose assumptions 2.2 and 2.3 hold. Let $i \in \mathcal{H}$ and $t \in \{0, \ldots, T - 1\}$.

First recall from (16) that, since $\tilde{g}_t^{(i)} = g_t^{(i)} + \xi_t^{(i)}, \xi_t^{(i)} \overset{\text{i.i.d.}}{\sim} \mathcal{N}(0, \sigma_{\mathrm{DP}}^2 I_d)$, we have

$$\mathbb{E}_{\xi_t^{(i)}}\left[\left\|\tilde{g}_t^{(i)} - g_t^{(i)}\right\|^2\right] = \mathbb{E}\left[\left\|\xi_t^{(i)}\right\|^2\right] = d \cdot \sigma_{\mathrm{DP}}^2.$$

Next, we have

$$\left\| \tilde{g}_t^{(i)} - \nabla \mathcal{L}_i(\theta_t) \right\|^2 = \left\| \tilde{g}_t^{(i)} - g_t^{(i)} + g_t^{(i)} - \nabla \mathcal{L}_i(\theta_t) \right\|^2$$
$$= \left\| \tilde{g}_t^{(i)} - g_t^{(i)} \right\|^2 + \left\| g_t^{(i)} - \nabla \mathcal{L}_i(\theta_t) \right\|^2 + 2 \left\langle \tilde{g}_t^{(i)} - g_t^{(i)}, \, g_t^{(i)} - \nabla \mathcal{L}_i(\theta_t) \right\rangle.$$

Now taking expectation on the randomness of $\xi_t^{(i)}$ (independent of all other random variables), and since $\mathbb{E}\left[\xi_t^{(i)}\right] = 0$, we get

$$\mathbb{E}_{\xi_t^{(i)}}\left[\left\| \tilde{g}_t^{(i)} - \nabla \mathcal{L}_i(\theta_t) \right\|^2\right] = \mathbb{E}_{\xi_t^{(i)}}\left[\left\| \tilde{g}_t^{(i)} - g_t^{(i)} \right\|^2\right] + \left\| g_t^{(i)} - \nabla \mathcal{L}_i(\theta_t) \right\|^2 + 2 \left\langle \underbrace{\mathbb{E}_{\xi_t^{(i)}}\left[\tilde{g}_t^{(i)} - g_t^{(i)}\right]}_{=\mathbb{E}\left[\xi_t^{(i)}\right]=0}, \, g_t^{(i)} - \nabla \mathcal{L}_i(\theta_t) \right\rangle$$

$$= \mathbb{E}_{\xi_t^{(i)}}\left[\left\| \tilde{g}_t^{(i)} - g_t^{(i)} \right\|^2\right] + \left\| g_t^{(i)} - \nabla \mathcal{L}_i(\theta_t) \right\|^2.$$

Upon taking total expectation, we obtain

$$\mathbb{E}\left[\left\| \tilde{g}_t^{(i)} - \nabla \mathcal{L}_i(\theta_t) \right\|^2\right] = \mathbb{E}\left[\left\| \tilde{g}_t^{(i)} - g_t^{(i)} \right\|^2\right] + \mathbb{E}\left[\left\| g_t^{(i)} - \nabla \mathcal{L}_i(\theta_t) \right\|^2\right]$$

$$= \mathbb{E}\left[\left\| g_t^{(i)} - \nabla \mathcal{L}_i(\theta_t) \right\|^2\right] + d \cdot \sigma_{\mathrm{DP}}^2. \tag{50}$$

First observe that when $m = 1$, as $b \in [m]$, we must have $b = m$. Thus, the gradient is deterministic, i.e., $g_t^{(i)} = \nabla \mathcal{L}_i(\theta_t)$. Thus, the first term in the equation above is zero, and the claimed bound holds.

Else, when $m \geq 2$, recall that from Assumption 2.2, we have $\mathbb{E}_{x \sim \mathcal{U}(\mathcal{D}_i)}\left[\|\nabla_\theta \ell(\theta_t; x) - \nabla \mathcal{L}_i(\theta_t)\|^2\right] \leq \sigma^2$. From (Rice, 2006), the variance reduction due to subsampling without replacement gives

$$\mathbb{E}\left[\left\| g_t^{(i)} - \nabla \mathcal{L}_i(\theta_t) \right\|^2\right] \leq \left(1 - \frac{b-1}{m-1}\right)\frac{\sigma^2}{b}.$$

Plugging this bound back in Equation (50) yields

$$\mathbb{E}\left[\left\| \tilde{g}_t^{(i)} - \nabla \mathcal{L}_i(\theta_t) \right\|^2\right] \leq \left(1 - \frac{b-1}{m-1}\right)\frac{\sigma^2}{b} + d \cdot \sigma_{\mathrm{DP}}^2.$$

By observing, as $m \geq 2$, that $1 - \frac{b-1}{m-1} = \frac{m-b}{m-1} = \frac{m}{m-1} \cdot \frac{m-b}{m} = (1 + \frac{1}{m-1})(1 - \frac{b}{m}) \leq 2(1 - \frac{b}{m})$, we obtain the final result:

$$\mathbb{E}\left[\left\| \tilde{g}_t^{(i)} - \nabla \mathcal{L}_i(\theta_t) \right\|^2\right] \leq 2\left(1 - \frac{b}{m}\right)\frac{\sigma^2}{b} + d \cdot \sigma_{\mathrm{DP}}^2.$$

$\square$

**Lemma C.6.** *Let $\sigma_{\mathrm{DP}} \geq 0$ and $d, n \geq 1$. Consider $\xi^{(1)}, \ldots, \xi^{(n)}$ to be i.i.d. random variables drawn from the Gaussian distribution $\mathcal{N}(0, \sigma_{\mathrm{DP}}^2 I_d)$. We have*

$$\mathbb{E}\left[\sup_{\|v\|\leq 1} \frac{1}{n}\sum_{i=1}^n \left\langle v, \, \xi^{(i)} \right\rangle^2\right] \leq 36\sigma_{\mathrm{DP}}^2\left(1 + \frac{d}{n}\right).$$

*Proof.* Let $\sigma_{\mathrm{DP}} \geq 0$ and $d, n \geq 1$. Consider $\xi^{(1)}, \ldots, \xi^{(n)}$ to be i.i.d. random variables drawn from the Gaussian distribution $\mathcal{N}(0, \sigma_{\mathrm{DP}}^2 I_d)$.

If $\sigma_{\mathrm{DP}} = 0$, then $\xi^{(i)} = 0$ almost surely for every $i \in [n]$, and the remainder of the proof holds with $\sigma_{\mathrm{DP}} = 0$. Else, we assume $\sigma_{\mathrm{DP}} > 0$ in the remaining.

Thus, the law of the random variable $\xi^{(i)}/\sigma_{\mathrm{DP}}$ is $\mathcal{N}(0, I_d)$ for every $i \in [n]$. Thus, for every vector $v$ in the unit ball, the random variable $\langle v, \xi^{(i)}/\sigma_{\mathrm{DP}} \rangle$ is sub-Gaussian with variance equal to 1 (see Chapter 1, (Rigollet & Hütter, 2015)). Therefore, for every $i \in [n]$ and every vector $v$ in the unit ball, applying Theorem 2.1.1 in (Pauwels, 2020)), we have

$$\mathbb{E}\left[\exp\left(\frac{1}{8}\left\langle v,\, \xi^{(i)}/\sigma_{\mathrm{DP}}\right\rangle^2\right)\right] \leq 2.$$

As a result, by the independence of $\xi^{(i)}$'s, we obtain

$$\sup_{\|v\|\leq 1} \mathbb{E}\left[\exp\left(\frac{1}{8\sigma_{\mathrm{DP}}^2}\sum_{i=1}^{n}\left\langle v,\, \xi^{(i)}\right\rangle^2\right)\right] = \sup_{\|v\|\leq 1}\prod_{i=1}^{n}\mathbb{E}\left[\exp\left(\frac{1}{8}\left\langle v,\, \xi^{(i)}/\sigma_{\mathrm{DP}}\right\rangle^2\right)\right] \leq 2^n.$$

Now, observe that we can write $\sum_{i=1}^{n}\left\langle v,\, \xi^{(i)}\right\rangle^2$ as the quadratic form $\langle v,\, Mv \rangle$, where $M := \sum_{i=1}^{n}\xi^{(i)}\cdot\xi^{(i)^\top}$ is a random real symmetric matrix. Thus, applying Lemma C.4 with the increasing function $g(\cdot) = \exp\left(\frac{1}{16\sigma_{\mathrm{DP}}^2}\times\cdot\right)$, we have

$$\mathbb{E}\left[\sup_{\|v\|\leq 1}\exp\left(\frac{1}{16\sigma_{\mathrm{DP}}^2}\sum_{i=1}^{n}\left\langle v,\, \xi^{(i)}\right\rangle^2\right)\right] = \mathbb{E}\left[\sup_{\|v\|\leq 1} g(\langle v,\, Mv\rangle)\right] \leq 9^d \cdot \sup_{\|v\|\leq 1}\mathbb{E}\left[g(2\langle v,\, Mv\rangle)\right]$$

$$= 9^d \cdot \sup_{\|v\|\leq 1}\mathbb{E}\left[\exp\left(\frac{1}{8\sigma_{\mathrm{DP}}^2}\sum_{i=1}^{n}\left\langle v,\, \xi^{(i)}\right\rangle^2\right)\right] \leq 9^d \cdot 2^n.$$

We can now use this inequality to bound the term of interest. We apply Jensen's inequality thanks to $\exp$ being convex, and we also interchange $\exp$ and $\sup$ thanks to the former being increasing:

$$\exp\left(\frac{1}{16\sigma_{\mathrm{DP}}^2}\mathbb{E}\left[\sup_{\|v\|\leq 1}\sum_{i=1}^{n}\left\langle v,\, \xi^{(i)}\right\rangle^2\right]\right) \leq \mathbb{E}\left[\exp\left(\frac{1}{16\sigma_{\mathrm{DP}}^2}\sup_{\|v\|\leq 1}\sum_{i=1}^{n}\left\langle v,\, \xi^{(i)}\right\rangle^2\right)\right]$$

$$= \mathbb{E}\left[\sup_{\|v\|\leq 1}\exp\left(\frac{1}{16\sigma_{\mathrm{DP}}^2}\sum_{i=1}^{n}\left\langle v,\, \xi^{(i)}\right\rangle^2\right)\right] \leq 9^d \cdot 2^n.$$

Upon taking $\ln$ and multiplying by $16\sigma_{\mathrm{DP}}^2/n$ on both sides, we obtain that

$$\mathbb{E}\left[\sup_{\|v\|\leq 1}\frac{1}{n}\sum_{i=1}^{n}\left\langle v,\, \xi^{(i)}\right\rangle^2\right] \leq 16\frac{\sigma_{\mathrm{DP}}^2}{n}(d\ln 9 + n\ln 2) \leq 36\frac{\sigma_{\mathrm{DP}}^2}{n}(d+n) = 36\sigma_{\mathrm{DP}}^2\left(1 + \frac{d}{n}\right).$$

The above concludes the lemma. □

### C.6.2. PROOF OF LEMMA C.1

**Lemma C.1.** *Suppose that assumptions 2.2 and 2.3 hold. Consider Algorithm 1. For every $t \in \{0, \ldots, T-1\}$, we have*

$$\mathbb{E}\left[\Delta_{t+1}\right] \leq \beta_t\,\mathbb{E}\left[\Delta_t\right] + 2(1-\beta_t)^2\left(\sigma_b^2 + 108\left(n\sigma_{\mathrm{cor}}^2 + \sigma_{\mathrm{ind}}^2\right)\left(1 + \frac{d}{n-f}\right)\right) + (1-\beta_t)G_{\mathrm{cov}}^2,$$

*where $\overline{m}_t := \frac{1}{|\mathcal{H}|}\sum_{i\in\mathcal{H}}m_t^{(i)}$, $\sigma_b^2 := 2(1 - \frac{b}{m})\frac{\sigma^2}{b}$, and $G_{\mathrm{cov}}^2 := \sup_{\theta\in\mathbb{R}^d}\sup_{\|v\|\leq 1}\frac{1}{|\mathcal{H}|}\sum_{i\in\mathcal{H}}\left\langle v,\, \nabla\mathcal{L}_i(\theta) - \nabla\mathcal{L}_{\mathcal{H}}(\theta)\right\rangle^2$.*

*Proof.* Let $t \in \{0, \ldots, T-1\}$. Suppose that Assumption 2.2 holds. Recall that the alternate definition of maximum eigenvalue implies, following the definition of $\Delta_t$ in Equation (19), that

$$\Delta_t = \lambda_{\max}\left(\frac{1}{|\mathcal{H}|}\sum_{i\in\mathcal{H}}(m_t^{(i)} - \overline{m}_t)(m_t^{(i)} - \overline{m}_t)^\top\right) = \sup_{\|v\|\leq 1}\frac{1}{|\mathcal{H}|}\sum_{i\in\mathcal{H}}\left\langle v,\, m_t^{(i)} - \overline{m}_t\right\rangle^2.$$

We will use the latter expression above for $\Delta_t$ throughout this lemma.

For every $i \in \mathcal{H}$, by definition of $m_t^{(i)}$, given in Equation (15), we have

$$m_{t+1}^{(i)} = \beta_t m_t^{(i)} + (1 - \beta_t) \tilde{g}_{t+1}^{(i)}.$$

We also denote $\overline{m}_t := \frac{1}{|\mathcal{H}|} \sum_{i \in \mathcal{H}} m_t^{(i)}$ and $\overline{\tilde{g}}_{t+1} := \frac{1}{|\mathcal{H}|} \sum_{i \in \mathcal{H}} \tilde{g}_{t+1}^{(i)}$. Therefore, we have $\overline{m}_{t+1} = \beta_t \overline{m}_t + (1 - \beta_t) \overline{\tilde{g}}_{t+1}$. As a result, we can write for every $i \in \mathcal{H}$

$$\begin{aligned}
m_{t+1}^{(i)} - \overline{m}_{t+1} &= \beta_t (m_t^{(i)} - \overline{m}_t) + (1 - \beta_t)(\tilde{g}_{t+1}^{(i)} - \overline{\tilde{g}}_{t+1}) \\
&= \beta_t (m_t^{(i)} - \overline{m}_t) + (1 - \beta_t)(\nabla \mathcal{L}_i(\theta_{t+1}) - \nabla \mathcal{L}_\mathcal{H}(\theta_{t+1})) \\
&\quad + (1 - \beta_t)(\tilde{g}_{t+1}^{(i)} - \nabla \mathcal{L}_i(\theta_{t+1}) - \overline{\tilde{g}}_{t+1} + \nabla \mathcal{L}_\mathcal{H}(\theta_{t+1})).
\end{aligned}$$

By projecting the above expression on an arbitrary vector $v$ and then taking squares, we obtain

$$\begin{aligned}
\left\langle v, m_{t+1}^{(i)} - \overline{m}_{t+1} \right\rangle^2 &= \Big[ \beta_t \left\langle v, m_t^{(i)} - \overline{m}_t \right\rangle + (1 - \beta_t) \left\langle v, \nabla \mathcal{L}_i(\theta_{t+1}) - \nabla \mathcal{L}_\mathcal{H}(\theta_{t+1}) \right\rangle \\
&\quad + (1 - \beta_t) \left\langle v, \tilde{g}_{t+1}^{(i)} - \nabla \mathcal{L}_i(\theta_{t+1}) - \overline{\tilde{g}}_{t+1} + \nabla \mathcal{L}_\mathcal{H}(\theta_{t+1}) \right\rangle \Big]^2 \\
&= \beta_t^2 \left\langle v, m_t^{(i)} - \overline{m}_t \right\rangle^2 + (1 - \beta_t)^2 \left\langle v, \nabla \mathcal{L}_i(\theta_{t+1}) - \nabla \mathcal{L}_\mathcal{H}(\theta_{t+1}) \right\rangle^2 \\
&\quad + (1 - \beta_t)^2 \left\langle v, \tilde{g}_{t+1}^{(i)} - \nabla \mathcal{L}_i(\theta_{t+1}) - \overline{\tilde{g}}_{t+1} + \nabla \mathcal{L}_\mathcal{H}(\theta_{t+1}) \right\rangle^2 \\
&\quad + 2\beta_t(1 - \beta_t) \left\langle v, m_t^{(i)} - \overline{m}_t \right\rangle \left\langle v, \nabla \mathcal{L}_i(\theta_{t+1}) - \nabla \mathcal{L}_\mathcal{H}(\theta_{t+1}) \right\rangle \\
&\quad + 2\beta_t(1 - \beta_t) \left\langle v, m_t^{(i)} - \overline{m}_t \right\rangle \left\langle v, \tilde{g}_{t+1}^{(i)} - \nabla \mathcal{L}_i(\theta_{t+1}) - \overline{\tilde{g}}_{t+1} + \nabla \mathcal{L}_\mathcal{H}(\theta_{t+1}) \right\rangle \\
&\quad + 2\beta_t(1 - \beta_t) \left\langle v, \nabla \mathcal{L}_i(\theta_{t+1}) - \nabla \mathcal{L}_\mathcal{H}(\theta_{t+1}) \right\rangle \left\langle v, \tilde{g}_{t+1}^{(i)} - \nabla \mathcal{L}_i(\theta_{t+1}) - \overline{\tilde{g}}_{t+1} + \nabla \mathcal{L}_\mathcal{H}(\theta_{t+1}) \right\rangle.
\end{aligned}$$

Upon averaging over $i \in \mathcal{H}$, taking the supremum over the unit ball, and then total expectations, we get

$$\begin{aligned}
\mathbb{E}\left[ \sup_{\|v\| \le 1} \frac{1}{|\mathcal{H}|} \sum_{i \in \mathcal{H}} \left\langle v, m_{t+1}^{(i)} - \overline{m}_{t+1} \right\rangle^2 \right] &= \beta_t^2 \mathbb{E}\left[ \sup_{\|v\| \le 1} \frac{1}{|\mathcal{H}|} \sum_{i \in \mathcal{H}} \left\langle v, m_t^{(i)} - \overline{m}_t \right\rangle^2 \right] \\
&+ (1 - \beta_t)^2 \mathbb{E}\left[ \sup_{\|v\| \le 1} \frac{1}{|\mathcal{H}|} \sum_{i \in \mathcal{H}} \left\langle v, \nabla \mathcal{L}_i(\theta_{t+1}) - \nabla \mathcal{L}_\mathcal{H}(\theta_{t+1}) \right\rangle^2 \right] \\
&+ (1 - \beta_t)^2 \mathbb{E}\left[ \sup_{\|v\| \le 1} \frac{1}{|\mathcal{H}|} \sum_{i \in \mathcal{H}} \left\langle v, \tilde{g}_{t+1}^{(i)} - \nabla \mathcal{L}_i(\theta_{t+1}) - \overline{\tilde{g}}_{t+1} + \nabla \mathcal{L}_\mathcal{H}(\theta_{t+1}) \right\rangle^2 \right] \\
&+ 2\beta_t(1 - \beta_t) \mathbb{E}\left[ \sup_{\|v\| \le 1} \frac{1}{|\mathcal{H}|} \sum_{i \in \mathcal{H}} \left\langle v, m_t^{(i)} - \overline{m}_t \right\rangle \left\langle v, \nabla \mathcal{L}_i(\theta_{t+1}) - \nabla \mathcal{L}_\mathcal{H}(\theta_{t+1}) \right\rangle \right] \\
&+ 2\beta_t(1 - \beta_t) \mathbb{E}\left[ \sup_{\|v\| \le 1} \frac{1}{|\mathcal{H}|} \sum_{i \in \mathcal{H}} \left\langle v, m_t^{(i)} - \overline{m}_t \right\rangle \left\langle v, \tilde{g}_{t+1}^{(i)} - \nabla \mathcal{L}_i(\theta_{t+1}) - \overline{\tilde{g}}_{t+1} + \nabla \mathcal{L}_\mathcal{H}(\theta_{t+1}) \right\rangle \right] \\
&+ 2\beta_t(1 - \beta_t) \mathbb{E}\left[ \sup_{\|v\| \le 1} \frac{1}{|\mathcal{H}|} \sum_{i \in \mathcal{H}} \left\langle v, \nabla \mathcal{L}_i(\theta_{t+1}) - \nabla \mathcal{L}_\mathcal{H}(\theta_{t+1}) \right\rangle \left\langle v, \tilde{g}_{t+1}^{(i)} - \nabla \mathcal{L}_i(\theta_{t+1}) - \overline{\tilde{g}}_{t+1} + \nabla \mathcal{L}_\mathcal{H}(\theta_{t+1}) \right\rangle \right].
\end{aligned}$$
$$(51)$$

We now show that the last two terms on the RHS of Equation (51) are non-positive. We show it for the first one, as the second one can be shown to be non-positive in the same way.

First, note that we can write the inner expression as a quadratic form. Precisely, we have for any vector $v$ and any $i \in \mathcal{H}$ that

$$2 \sum_{i \in \mathcal{H}} \left\langle v, m_t^{(i)} - \overline{m}_t \right\rangle \left\langle v, \tilde{g}_{t+1}^{(i)} - \nabla \mathcal{L}_i(\theta_{t+1}) - \overline{\tilde{g}}_{t+1} + \nabla \mathcal{L}_\mathcal{H}(\theta_{t+1}) \right\rangle = \left\langle v, Mv \right\rangle,$$

where we have introduced the $d \times d$ matrix $M := N + N^\top$, such that $N := \sum_{i \in \mathcal{H}} (m_t^{(i)} - \overline{m}_t)(\tilde{g}_{t+1}^{(i)} - \nabla \mathcal{L}_i(\theta_{t+1}) - \overline{\tilde{g}}_{t+1} + \nabla \mathcal{L}_\mathcal{H}(\theta_{t+1}))^\top$. By observing that $M$ is symmetric, we can apply Lemma C.4 with $g$ being the identity mapping:

$$\mathbb{E}\left[\sup_{\|v\|\leq 1} 2 \sum_{i \in \mathcal{H}} \left\langle v, m_t^{(i)} - \overline{m}_t \right\rangle \left\langle v, \tilde{g}_{t+1}^{(i)} - \nabla \mathcal{L}_i(\theta_{t+1}) - \overline{\tilde{g}}_{t+1} + \nabla \mathcal{L}_\mathcal{H}(\theta_{t+1}) \right\rangle\right] = \mathbb{E}\left[\sup_{\|v\|\leq 1} \langle v, Mv \rangle\right]$$

$$\leq 9^d \cdot \sup_{\|v\|\leq 1} \mathbb{E}\left[2 \langle v, Mv \rangle\right]. \quad (52)$$

However, the last term is zero by the total law of expectation. Indeed, recall that stochastic gradients are unbiased (Assumption 2.2) and that $\theta_{t+1}$ and $m_t^{(i)}$ are deterministic when given history $\mathcal{P}_{t+1}$. This gives

$$\mathbb{E}\left[\langle v, Mv \rangle\right] = \mathbb{E}\left[2 \sum_{i \in \mathcal{H}} \left\langle v, m_t^{(i)} - \overline{m}_t \right\rangle \left\langle v, \tilde{g}_{t+1}^{(i)} - \nabla \mathcal{L}_i(\theta_{t+1}) - \overline{\tilde{g}}_{t+1} + \nabla \mathcal{L}_\mathcal{H}(\theta_{t+1}) \right\rangle\right]$$

$$= \mathbb{E}\left[\mathbb{E}_{t+1}\left[2 \sum_{i \in \mathcal{H}} \left\langle v, m_t^{(i)} - \overline{m}_t \right\rangle \left\langle v, \tilde{g}_{t+1}^{(i)} - \nabla \mathcal{L}_i(\theta_{t+1}) - \overline{\tilde{g}}_{t+1} + \nabla \mathcal{L}_\mathcal{H}(\theta_{t+1}) \right\rangle\right]\right]$$

$$= \mathbb{E}\left[2 \sum_{i \in \mathcal{H}} \left\langle v, m_t^{(i)} - \overline{m}_t \right\rangle \left\langle v, \underbrace{\mathbb{E}_{t+1}\left[\tilde{g}_{t+1}^{(i)} - \nabla \mathcal{L}_i(\theta_{t+1})\right]}_{=0} - \underbrace{\mathbb{E}_{t+1}\left[\overline{\tilde{g}}_{t+1} - \nabla \mathcal{L}_\mathcal{H}(\theta_{t+1})\right]}_{=0} \right\rangle\right] = 0.$$

Moreover, going back to Equation (52), we obtain

$$\mathbb{E}\left[\sup_{\|v\|\leq 1} 2 \sum_{i \in \mathcal{H}} \left\langle v, m_t^{(i)} - \overline{m}_t \right\rangle \left\langle v, \tilde{g}_{t+1}^{(i)} - \nabla \mathcal{L}_i(\theta_{t+1}) - \overline{\tilde{g}}_{t+1} + \nabla \mathcal{L}_\mathcal{H}(\theta_{t+1}) \right\rangle\right] \leq 9^d \cdot \sup_{\|v\|\leq 1} \mathbb{E}\left[2 \langle v, Mv \rangle\right] = 0.$$

As mentioned previously, we can prove in the same way that

$$\mathbb{E}\left[\sup_{\|v\|\leq 1} 2 \sum_{i \in \mathcal{H}} \langle v, \nabla \mathcal{L}_i(\theta_{t+1}) - \nabla \mathcal{L}_\mathcal{H}(\theta_{t+1}) \rangle \left\langle v, \tilde{g}_{t+1}^{(i)} - \nabla \mathcal{L}_i(\theta_{t+1}) - \overline{\tilde{g}}_{t+1} + \nabla \mathcal{L}_\mathcal{H}(\theta_{t+1}) \right\rangle\right] \leq 0.$$

Plugging the two previous bounds back in Equation (51), we have thus proved that

$$\mathbb{E}\left[\sup_{\|v\|\leq 1} \frac{1}{|\mathcal{H}|} \sum_{i \in \mathcal{H}} \left\langle v, m_{t+1}^{(i)} - \overline{m}_{t+1} \right\rangle^2\right] = \beta_t^2 \mathbb{E}\left[\sup_{\|v\|\leq 1} \frac{1}{|\mathcal{H}|} \sum_{i \in \mathcal{H}} \left\langle v, m_t^{(i)} - \overline{m}_t \right\rangle^2\right]$$

$$+ (1 - \beta_t)^2 \mathbb{E}\left[\sup_{\|v\|\leq 1} \frac{1}{|\mathcal{H}|} \sum_{i \in \mathcal{H}} \langle v, \nabla \mathcal{L}_i(\theta_{t+1}) - \nabla \mathcal{L}_\mathcal{H}(\theta_{t+1}) \rangle^2\right]$$

$$+ (1 - \beta_t)^2 \mathbb{E}\left[\sup_{\|v\|\leq 1} \frac{1}{|\mathcal{H}|} \sum_{i \in \mathcal{H}} \left\langle v, \tilde{g}_{t+1}^{(i)} - \nabla \mathcal{L}_i(\theta_{t+1}) - \overline{\tilde{g}}_{t+1} + \nabla \mathcal{L}_\mathcal{H}(\theta_{t+1}) \right\rangle^2\right]$$

$$+ 2\beta_t(1 - \beta_t) \mathbb{E}\left[\sup_{\|v\|\leq 1} \frac{1}{|\mathcal{H}|} \sum_{i \in \mathcal{H}} \left\langle v, m_t^{(i)} - \overline{m}_t \right\rangle \langle v, \nabla \mathcal{L}_i(\theta_{t+1}) - \nabla \mathcal{L}_\mathcal{H}(\theta_{t+1}) \rangle\right]. \quad (53)$$

We now bound the two last terms on the RHS of Equation (53).

First, by using the fact that $2ab \leq a^2 + b^2$, we have for any vector $v$ that

$$\frac{2}{|\mathcal{H}|} \sum_{i \in \mathcal{H}} \left\langle v, m_t^{(i)} - \overline{m}_t \right\rangle \langle v, \nabla \mathcal{L}_i(\theta_{t+1}) - \nabla \mathcal{L}_\mathcal{H}(\theta_{t+1}) \rangle \leq \frac{1}{|\mathcal{H}|} \sum_{i \in \mathcal{H}} \left[\left\langle v, m_t^{(i)} - \overline{m}_t \right\rangle^2 + \langle v, \nabla \mathcal{L}_i(\theta_{t+1}) - \nabla \mathcal{L}_\mathcal{H}(\theta_{t+1}) \rangle^2\right]$$

$$= \frac{1}{|\mathcal{H}|} \sum_{i \in \mathcal{H}} \left\langle v, m_t^{(i)} - \overline{m}_t \right\rangle^2 + \frac{1}{|\mathcal{H}|} \sum_{i \in \mathcal{H}} \langle v, \nabla \mathcal{L}_i(\theta_{t+1}) - \nabla \mathcal{L}_\mathcal{H}(\theta_{t+1}) \rangle^2. \quad (54)$$

Taking the supremum over the unit ball and then total expectations on both sides yields

$$2\, \mathbb{E}\left[\sup_{\|v\|\leq 1} \frac{1}{|\mathcal{H}|} \sum_{i\in\mathcal{H}} \left\langle v,\, m_t^{(i)} - \overline{m}_t \right\rangle \langle v,\, \nabla\mathcal{L}_i(\theta_{t+1}) - \nabla\mathcal{L}_{\mathcal{H}}(\theta_{t+1})\rangle\right]$$

$$\leq \mathbb{E}\left[\sup_{\|v\|\leq 1} \frac{1}{|\mathcal{H}|} \sum_{i\in\mathcal{H}} \left\langle v,\, m_t^{(i)} - \overline{m}_t \right\rangle^2\right] + \mathbb{E}\left[\sup_{\|v\|\leq 1} \frac{1}{|\mathcal{H}|} \sum_{i\in\mathcal{H}} \langle v,\, \nabla\mathcal{L}_i(\theta_{t+1}) - \nabla\mathcal{L}_{\mathcal{H}}(\theta_{t+1})\rangle^2\right]. \qquad (55)$$

Second, recall that $\tilde{g}_{t+1}^{(i)} = g_{t+1}^{(i)} + \xi_{t+1}^{(i)}$, where we denote $\xi_{t+1}^{(i)} := \sum_{\substack{j=1 \\ j\neq i}}^{n} v_{t+1}^{(ij)} + \overline{v}_{t+1}^{(i)}$, where $\overline{v}_{t+1}^{(i)} \sim \mathcal{N}(0, \sigma_{\mathrm{ind}}^2 I_d)$ and $v_{t+1}^{(ij)} = -v_{t+1}^{(ji)} \sim \mathcal{N}(0, \sigma_{\mathrm{cor}}^2 I_d)$. Denote $\overline{\xi}_{t+1} := \frac{1}{|\mathcal{H}|} \sum_{i\in\mathcal{H}} \xi_{t+1}^{(i)}$. Therefore, by applying Jensen's inequality, we have

$$\mathbb{E}\left[\sup_{\|v\|\leq 1} \frac{1}{|\mathcal{H}|} \sum_{i\in\mathcal{H}} \left\langle v,\, \tilde{g}_{t+1}^{(i)} - \nabla\mathcal{L}_i(\theta_{t+1}) - \overline{\overline{g}}_{t+1} + \nabla\mathcal{L}_{\mathcal{H}}(\theta_{t+1}) \right\rangle^2\right]$$

$$= \mathbb{E}\left[\sup_{\|v\|\leq 1} \frac{1}{|\mathcal{H}|} \sum_{i\in\mathcal{H}} \left\langle v,\, g_{t+1}^{(i)} - \nabla\mathcal{L}_i(\theta_{t+1}) - \overline{g}_{t+1} + \nabla\mathcal{L}_{\mathcal{H}}(\theta_{t+1}) + \xi_{t+1}^{(i)} - \overline{\xi}_{t+1} \right\rangle^2\right]$$

$$\leq 2\, \mathbb{E}\left[\sup_{\|v\|\leq 1} \frac{1}{|\mathcal{H}|} \sum_{i\in\mathcal{H}} \left[\left\langle v,\, g_{t+1}^{(i)} - \nabla\mathcal{L}_i(\theta_{t+1}) - \overline{g}_{t+1} + \nabla\mathcal{L}_{\mathcal{H}}(\theta_{t+1}) \right\rangle^2 + \left\langle v,\, \xi_{t+1}^{(i)} - \overline{\xi}_{t+1} \right\rangle^2\right]\right]$$

Recall the following bias-variance decomposition: for any $x_1, \ldots, x_n \in \mathbb{R}$, denoting $\overline{x} := \frac{1}{n}\sum_{i=1}^{n} x_i$, we have $\frac{1}{n}\sum_{i=1}^{n}(x_i - \overline{x})^2 = \frac{1}{n}\sum_{i=1}^{n} x_i^2 - \overline{x}^2 \leq \sum_{i=1}^{n} x_i^2$. Applying this fact above yields

$$\mathbb{E}\left[\sup_{\|v\|\leq 1} \frac{1}{|\mathcal{H}|} \sum_{i\in\mathcal{H}} \left\langle v,\, \tilde{g}_{t+1}^{(i)} - \nabla\mathcal{L}_i(\theta_{t+1}) - \overline{\overline{g}}_{t+1} + \nabla\mathcal{L}_{\mathcal{H}}(\theta_{t+1}) \right\rangle^2\right]$$

$$\leq 2\, \mathbb{E}\left[\sup_{\|v\|\leq 1} \frac{1}{|\mathcal{H}|} \sum_{i\in\mathcal{H}} \left[\left\langle v,\, g_{t+1}^{(i)} - \nabla\mathcal{L}_i(\theta_{t+1}) \right\rangle^2 + \left\langle v,\, \xi_{t+1}^{(i)} \right\rangle^2\right]\right]$$

$$\leq 2\, \mathbb{E}\left[\frac{1}{|\mathcal{H}|} \sum_{i\in\mathcal{H}} \left\|g_{t+1}^{(i)} - \nabla\mathcal{L}_i(\theta_{t+1})\right\|^2\right] + 2\, \mathbb{E}\left[\sup_{\|v\|\leq 1} \frac{1}{|\mathcal{H}|} \sum_{i\in\mathcal{H}} \left\langle v,\, \xi_{t+1}^{(i)} \right\rangle^2\right], \qquad (56)$$

where the last inequality is due to the Cauchy-Schwartz inequality. Recall that, by Assumption 2.2 and Lemma C.5 applied with zero privacy noise, we have for every $i \in \mathcal{H}$ that $\mathbb{E}\left[\left\|g_{t+1}^{(i)} - \nabla\mathcal{L}_i(\theta_{t+1})\right\|^2\right] \leq 2(1 - \frac{b}{m})\frac{\sigma^2}{b} =: \sigma_b^2$. Therefore, upon averaging over $i \in \mathcal{H}$, we have

$$\mathbb{E}\left[\frac{1}{|\mathcal{H}|} \sum_{i\in\mathcal{H}} \left\|g_{t+1}^{(i)} - \nabla\mathcal{L}_i(\theta_{t+1})\right\|^2\right] \leq \sigma_b^2. \qquad (57)$$

We now bound the remaining (last) term on the RHS of (56).

$$
\mathbb{E}\left[\sup_{\|v\|\le 1} \frac{1}{|\mathcal{H}|}\sum_{i\in\mathcal{H}}\left\langle v, \xi_{t+1}^{(i)}\right\rangle^2\right] = \mathbb{E}\sup_{\|v\|\le 1}\frac{1}{|\mathcal{H}|}\sum_{i\in\mathcal{H}}\left\langle v, \sum_{\substack{j=1\\j\ne i}}^{n} v_{t+1}^{(ij)} + \overline{v}_{t+1}^{(i)}\right\rangle^2 = \mathbb{E}\sup_{\|v\|\le 1}\frac{1}{|\mathcal{H}|}\sum_{i\in\mathcal{H}}\left(\sum_{\substack{j=1\\j\ne i}}^{n}\left\langle v, v_{t+1}^{(ij)}\right\rangle + \left\langle v, \overline{v}_{t+1}^{(i)}\right\rangle\right)^2
$$

$$
= \mathbb{E}\sup_{\|v\|\le 1}\frac{1}{|\mathcal{H}|}\sum_{i\in\mathcal{H}}\left(\sum_{\substack{j=1\\j\ne i}}^{n}\left\langle v, v_{t+1}^{(ij)}\right\rangle^2 + \left\langle v, \overline{v}_{t+1}^{(i)}\right\rangle^2 + 2\sum_{\substack{j=1\\j\ne i}}^{n}\left\langle v, v_{t+1}^{(ij)}\right\rangle\left\langle v, \overline{v}_{t+1}^{(i)}\right\rangle + 2\sum_{\substack{j=1\\k<j\ne i}}^{n}\left\langle v, v_{t+1}^{(ij)}\right\rangle\left\langle v, v_{t+1}^{(ik)}\right\rangle\right)
$$

$$
\le \mathbb{E}\sup_{\|v\|\le 1}\frac{1}{|\mathcal{H}|}\sum_{i\in\mathcal{H}}\left(\sum_{\substack{j=1\\j\ne i}}^{n}\left\langle v, v_{t+1}^{(ij)}\right\rangle^2 + \left\langle v, \overline{v}_{t+1}^{(i)}\right\rangle^2\right)
$$

$$
+ 2\mathbb{E}\sup_{\|v\|\le 1}\frac{1}{|\mathcal{H}|}\sum_{i\in\mathcal{H}}\left(\sum_{\substack{j=1\\j\ne i}}^{n}\left\langle v, v_{t+1}^{(ij)}\right\rangle\left\langle v, \overline{v}_{t+1}^{(i)}\right\rangle + \sum_{\substack{j=1\\k<j\ne i}}^{n}\left\langle v, v_{t+1}^{(ij)}\right\rangle\left\langle v, v_{t+1}^{(ik)}\right\rangle\right)
$$

We observe that the second term above is non-positive as we have shown for (52) using Lemma C.4 and the fact that for a fixed $i \in \mathcal{H}$, all the noise terms in $\xi_{t+1}^{(i)}$ are statistically independent and are zero in expectation. Therefore, we have from above that

$$
\mathbb{E}\left[\sup_{\|v\|\le 1}\frac{1}{|\mathcal{H}|}\sum_{i\in\mathcal{H}}\left\langle v, \xi_{t+1}^{(i)}\right\rangle^2\right] \le \mathbb{E}\sup_{\|v\|\le 1}\frac{1}{|\mathcal{H}|}\sum_{i\in\mathcal{H}}\left(\sum_{\substack{j=1\\j\ne i}}^{n}\left\langle v, v_{t+1}^{(ij)}\right\rangle^2 + \left\langle v, \overline{v}_{t+1}^{(i)}\right\rangle^2\right).
$$

We observe here, since $v_{t+1}^{(ij)} = -v_{t+1}^{(ji)}$ for all $i, j \in \mathcal{H}$, that $\sum_{i\in\mathcal{H}}\sum_{\substack{j=1\\j\ne i}}^{n}\left\langle v, v_{t+1}^{(ij)}\right\rangle^2 = \sum_{i\in\mathcal{H}}\sum_{j\in\mathcal{H}\setminus\{i\}}\left\langle v, v_{t+1}^{(ij)}\right\rangle^2 +$
$\sum_{i\in\mathcal{H}}\sum_{j\in[n]\setminus\mathcal{H}}\left\langle v, v_{t+1}^{(ij)}\right\rangle^2 = 2\sum_{i\in\mathcal{H}}\sum_{j\in\mathcal{H},j<i}\left\langle v, v_{t+1}^{(ij)}\right\rangle^2 + \sum_{i\in\mathcal{H}}\sum_{j\in[n]\setminus\mathcal{H}}\left\langle v, v_{t+1}^{(ij)}\right\rangle^2$. Therefore, we have

$$
\mathbb{E}\left[\sup_{\|v\|\le 1}\frac{1}{|\mathcal{H}|}\sum_{i\in\mathcal{H}}\left\langle v, \xi_{t+1}^{(i)}\right\rangle^2\right] \le \mathbb{E}\sup_{\|v\|\le 1}\frac{1}{|\mathcal{H}|}\sum_{i\in\mathcal{H}}\left(2\sum_{\substack{j\in\mathcal{H}\\j<i}}\left\langle v, v_{t+1}^{(ij)}\right\rangle^2 + \sum_{j\in[n]\setminus\mathcal{H}}\left\langle v, v_{t+1}^{(ij)}\right\rangle^2 + \left\langle v, \overline{v}_{t+1}^{(i)}\right\rangle^2\right)
$$

$$
\le 2\mathbb{E}\sup_{\|v\|\le 1}\frac{1}{|\mathcal{H}|}\sum_{i\in\mathcal{H}}\sum_{\substack{j\in\mathcal{H}\\j<i}}\left\langle v, v_{t+1}^{(ij)}\right\rangle^2 + \mathbb{E}\sup_{\|v\|\le 1}\frac{1}{|\mathcal{H}|}\sum_{i\in\mathcal{H}}\sum_{j\in[n]\setminus\mathcal{H}}\left\langle v, v_{t+1}^{(ij)}\right\rangle^2
$$

$$
+ \mathbb{E}\sup_{\|v\|\le 1}\frac{1}{|\mathcal{H}|}\sum_{i\in\mathcal{H}}\left\langle v, \overline{v}_{t+1}^{(i)}\right\rangle^2.
$$

Now, for each of the three suprema in the right-hand side above, all random variables involved are independent Gaussians

with respective variances, so that applying Lemma C.6 separately for each of the three suprema yields:

$$
\begin{aligned}
\mathbb{E}\left[\sup_{\|v\|\leq 1}\frac{1}{|\mathcal{H}|}\sum_{i\in\mathcal{H}}\left\langle v,\xi_{t+1}^{(i)}\right\rangle^2\right] &\leq 2\mathbb{E}\sup_{\|v\|\leq 1}\frac{1}{|\mathcal{H}|}\sum_{i\in\mathcal{H}}\sum_{\substack{j\in\mathcal{H}\\ j<i}}\left\langle v,v_{t+1}^{(ij)}\right\rangle^2 + \mathbb{E}\sup_{\|v\|\leq 1}\frac{1}{|\mathcal{H}|}\sum_{i\in\mathcal{H}}\sum_{j\in[n]\setminus\mathcal{H}}\left\langle v,v_{t+1}^{(ij)}\right\rangle^2 \\
&\quad + \mathbb{E}\sup_{\|v\|\leq 1}\frac{1}{|\mathcal{H}|}\sum_{i\in\mathcal{H}}\left\langle v,\bar{v}_{t+1}^{(i)}\right\rangle^2 \\
&\leq 36\frac{2}{|\mathcal{H}|}\frac{|\mathcal{H}|\,(|\mathcal{H}|-1)}{2}\sigma_{\mathrm{cor}}^2(1+\frac{2d}{|\mathcal{H}|\,(|\mathcal{H}|-1)}) \\
&\quad + 36\frac{1}{|\mathcal{H}|}\frac{|\mathcal{H}|\,(n-|\mathcal{H}|)}{2}\sigma_{\mathrm{cor}}^2(1+\frac{2d}{|\mathcal{H}|\,(n-|\mathcal{H}|)}) + 36\sigma_{\mathrm{ind}}^2(1+\frac{d}{|\mathcal{H}|}) \\
&= 36(n-\frac{f}{2}-1)\sigma_{\mathrm{cor}}^2(1+\frac{3d}{n-f}) + 36\sigma_{\mathrm{ind}}^2(1+\frac{d}{n-f}) \\
&\leq 108\left(n\sigma_{\mathrm{cor}}^2+\sigma_{\mathrm{ind}}^2\right)(1+\frac{d}{n-f}). 
\end{aligned}
\tag{58}
$$

Plugging the bounds obtained in (57) and (58) back in (56), we get

$$
\mathbb{E}\left[\sup_{\|v\|\leq 1}\frac{1}{|\mathcal{H}|}\sum_{i\in\mathcal{H}}\left\langle v,\tilde{g}_{t+1}^{(i)}-\nabla\mathcal{L}_i(\theta_{t+1})-\bar{\tilde{g}}_{t+1}+\nabla\mathcal{L}_\mathcal{H}(\theta_{t+1})\right\rangle^2\right] \leq 2\left(\sigma_b^2+108\left(n\sigma_{\mathrm{cor}}^2+\sigma_{\mathrm{ind}}^2\right)\left(1+\frac{d}{n-f}\right)\right).
\tag{59}
$$

We use the above bounds in (55) and (59) to bound the RHS of (53), which yields

$$
\begin{aligned}
\mathbb{E}\left[\sup_{\|v\|\leq 1}\frac{1}{|\mathcal{H}|}\sum_{i\in\mathcal{H}}\left\langle v,m_{t+1}^{(i)}-\overline{m}_{t+1}\right\rangle^2\right] &\leq \beta_t^2\,\mathbb{E}\left[\sup_{\|v\|\leq 1}\frac{1}{|\mathcal{H}|}\sum_{i\in\mathcal{H}}\left\langle v,m_t^{(i)}-\overline{m}_t\right\rangle^2\right] \\
&\quad + (1-\beta_t)^2\,\mathbb{E}\left[\sup_{\|v\|\leq 1}\frac{1}{|\mathcal{H}|}\sum_{i\in\mathcal{H}}\langle v,\nabla\mathcal{L}_i(\theta_{t+1})-\nabla\mathcal{L}_\mathcal{H}(\theta_{t+1})\rangle^2\right] + 2(1-\beta_t)^2\left(\sigma_b^2+108\left(n\sigma_{\mathrm{cor}}^2+\sigma_{\mathrm{ind}}^2\right)\left(1+\frac{d}{n-f}\right)\right) \\
&\quad + \beta_t(1-\beta_t)\,\mathbb{E}\left[\sup_{\|v\|\leq 1}\frac{1}{|\mathcal{H}|}\sum_{i\in\mathcal{H}}\left\langle v,m_t^{(i)}-\overline{m}_t\right\rangle^2 + \sup_{\|v\|\leq 1}\frac{1}{|\mathcal{H}|}\sum_{i\in\mathcal{H}}\langle v,\nabla\mathcal{L}_i(\theta_{t+1})-\nabla\mathcal{L}_\mathcal{H}(\theta_{t+1})\rangle^2\right].
\end{aligned}
$$

By rearranging terms, and noticing that $\beta_t^2+\beta_t(1-\beta_t)=\beta_t$ and $(1-\beta_t)^2+\beta_t(1-\beta_t)=1-\beta_t$, we obtain

$$
\begin{aligned}
\mathbb{E}\left[\sup_{\|v\|\leq 1}\frac{1}{|\mathcal{H}|}\sum_{i\in\mathcal{H}}\left\langle v,m_{t+1}^{(i)}-\overline{m}_{t+1}\right\rangle^2\right] &\leq \beta_t\,\mathbb{E}\left[\sup_{\|v\|\leq 1}\frac{1}{|\mathcal{H}|}\sum_{i\in\mathcal{H}}\left\langle v,m_t^{(i)}-\overline{m}_t\right\rangle^2\right] \\
&\quad + (1-\beta_t)\,\mathbb{E}\left[\sup_{\|v\|\leq 1}\frac{1}{|\mathcal{H}|}\sum_{i\in\mathcal{H}}\langle v,\nabla\mathcal{L}_i(\theta_{t+1})-\nabla\mathcal{L}_\mathcal{H}(\theta_{t+1})\rangle^2\right] + 2(1-\beta_t)^2\left(\sigma_b^2+108\left(n\sigma_{\mathrm{cor}}^2+\sigma_{\mathrm{ind}}^2\right)\left(1+\frac{d}{n-f}\right)\right).
\end{aligned}
$$

Denote $G_{\mathrm{cov}}^2 := \sup_{\theta\in\mathbb{R}^d}\sup_{\|v\|\leq 1}\frac{1}{|\mathcal{H}|}\sum_{i\in\mathcal{H}}\langle v,\nabla\mathcal{L}_i(\theta)-\nabla\mathcal{L}_\mathcal{H}(\theta)\rangle^2$. Then, the above bound implies

$$
\begin{aligned}
\mathbb{E}\left[\sup_{\|v\|\leq 1}\frac{1}{|\mathcal{H}|}\sum_{i\in\mathcal{H}}\left\langle v,m_{t+1}^{(i)}-\overline{m}_{t+1}\right\rangle^2\right] &\leq \beta_t\,\mathbb{E}\left[\sup_{\|v\|\leq 1}\frac{1}{|\mathcal{H}|}\sum_{i\in\mathcal{H}}\left\langle v,m_t^{(i)}-\overline{m}_t\right\rangle^2\right] \\
&\quad + 2(1-\beta_t)^2\left(\sigma_b^2+108\left(n\sigma_{\mathrm{cor}}^2+\sigma_{\mathrm{ind}}^2\right)\left(1+\frac{d}{n-f}\right)\right) + (1-\beta_t)G_{\mathrm{cov}}^2.
\end{aligned}
$$

The above inequality concludes the proof. $\qquad\square$

C.6.3. PROOF OF LEMMA C.2

**Lemma C.2.** *Suppose that assumptions 2.2 and 2.3 hold and that $\mathcal{L}_{\mathcal{H}}$ is L-smooth. Consider Algorithm 1. For all $t \in \{0, \dots, T-1\}$, we have*

$$\mathbb{E}\left[\|\delta_{t+1}\|^2\right] \leq \beta_t^2(1+\gamma_t L)(1+4\gamma_t L)\,\mathbb{E}\left[\|\delta_t\|^2\right] + 4\gamma_t L(1+\gamma_t L)\beta_t^2\,\mathbb{E}\left[\|\nabla\mathcal{L}_{\mathcal{H}}(\theta_t)\|^2\right]$$
$$+ (1-\beta_t)^2\frac{\overline{\sigma}_{\mathrm{DP}}^2}{n-f} + 2\gamma_t L(1+\gamma_t L)\beta_t^2\,\mathbb{E}\left[\|\epsilon_t\|^2\right],$$

*where $\overline{\sigma}_{\mathrm{DP}}^2 := \left(1-\frac{b}{m}\right)\frac{\sigma^2}{b} + df\sigma_{\mathrm{cor}}^2 + d\sigma_{\mathrm{ind}}^2$.*

*Proof.* Let $t \in \{0, \dots, T-1\}$. Suppose that assumptions 2.2 and 2.3 hold and that $\mathcal{L}_{\mathcal{H}}$ is L-smooth.

Recall from (20) that

$$\delta_{t+1} := \overline{m}_{t+1} - \nabla\mathcal{L}_{\mathcal{H}}\left(\theta_{t+1}\right).$$

Denote $\overline{\tilde{g}}_t := \frac{1}{|\mathcal{H}|}\sum_{i\in\mathcal{H}}\tilde{g}_t^{(i)}$. Substituting from (15) and recalling that $\overline{m}_t = \frac{1}{|\mathcal{H}|}\sum_{i\in\mathcal{H}}m_t^{(i)}$, we obtain

$$\delta_{t+1} = \beta_t\,\overline{m}_t + (1-\beta_t)\overline{\tilde{g}}_{t+1} - \nabla\mathcal{L}_{\mathcal{H}}\left(\theta_{t+1}\right).$$

Upon adding and subtracting $\beta_t\nabla\mathcal{L}_{\mathcal{H}}(\theta_t)$ and $\beta_t\nabla\mathcal{L}_{\mathcal{H}}(\theta_{t+1})$ on the R.H.S. above we obtain that

$$\delta_{t+1} = \beta_t\,\overline{m}_t - \beta_t\nabla\mathcal{L}_{\mathcal{H}}(\theta_t) + (1-\beta_t)\overline{\tilde{g}}_{t+1} - \nabla\mathcal{L}_{\mathcal{H}}\left(\theta_{t+1}\right) + \beta_t\nabla\mathcal{L}_{\mathcal{H}}(\theta_{t+1}) + \beta_t\nabla\mathcal{L}_{\mathcal{H}}(\theta_t) - \beta_t\nabla\mathcal{L}_{\mathcal{H}}(\theta_{t+1})$$
$$= \beta_t\left(\overline{m}_t - \nabla\mathcal{L}_{\mathcal{H}}(\theta_t)\right) + (1-\beta_t)\overline{\tilde{g}}_{t+1} - (1-\beta_t)\nabla\mathcal{L}_{\mathcal{H}}\left(\theta_{t+1}\right) + \beta_t\left(\nabla\mathcal{L}_{\mathcal{H}}(\theta_t) - \nabla\mathcal{L}_{\mathcal{H}}(\theta_{t+1})\right).$$

As $\overline{m}_t - \nabla\mathcal{L}_{\mathcal{H}}(\theta_t) = \delta_t$ (by (20)), from above we obtain that

$$\delta_{t+1} = \beta_t\delta_t + (1-\beta_t)\left(\overline{\tilde{g}}_{t+1} - \nabla\mathcal{L}_{\mathcal{H}}\left(\theta_{t+1}\right)\right) + \beta_t\left(\nabla\mathcal{L}_{\mathcal{H}}(\theta_t) - \nabla\mathcal{L}_{\mathcal{H}}(\theta_{t+1})\right).$$

Therefore,

$$\|\delta_{t+1}\|^2 = \beta_t^2\|\delta_t\|^2 + (1-\beta_t)^2\left\|\overline{\tilde{g}}_{t+1} - \nabla\mathcal{L}_{\mathcal{H}}\left(\theta_{t+1}\right)\right\|^2$$
$$+ \beta_t^2\left\|\nabla\mathcal{L}_{\mathcal{H}}(\theta_t) - \nabla\mathcal{L}_{\mathcal{H}}(\theta_{t+1})\right\|^2 + 2\beta_t(1-\beta_t)\left\langle\delta_t,\,\overline{\tilde{g}}_{t+1} - \nabla\mathcal{L}_{\mathcal{H}}\left(\theta_{t+1}\right)\right\rangle$$
$$+ 2\beta_t^2\left\langle\delta_t,\,\nabla\mathcal{L}_{\mathcal{H}}(\theta_t) - \nabla\mathcal{L}_{\mathcal{H}}(\theta_{t+1})\right\rangle + 2\beta_t(1-\beta_t)\left\langle\overline{\tilde{g}}_{t+1} - \nabla\mathcal{L}_{\mathcal{H}}\left(\theta_{t+1}\right),\,\nabla\mathcal{L}_{\mathcal{H}}(\theta_t) - \nabla\mathcal{L}_{\mathcal{H}}(\theta_{t+1})\right\rangle.$$

By taking conditional expectation $\mathbb{E}_{t+1}\left[\cdot\right]$ on both sides, and recalling that $\delta_t$, $\theta_{t+1}$ and $\theta_t$ are deterministic values when the history $\mathcal{P}_{t+1}$ is given, we obtain that

$$\mathbb{E}_{t+1}\left[\|\delta_{t+1}\|^2\right] = \beta_t^2\|\delta_t\|^2 + (1-\beta_t)^2\mathbb{E}_{t+1}\left[\left\|\overline{\tilde{g}}_{t+1} - \nabla\mathcal{L}_{\mathcal{H}}\left(\theta_{t+1}\right)\right\|^2\right] + \beta_t^2\|\nabla\mathcal{L}_{\mathcal{H}}(\theta_t) - \nabla\mathcal{L}_{\mathcal{H}}(\theta_{t+1})\|^2 +$$
$$2\beta_t(1-\beta_t)\left\langle\delta_t,\,\mathbb{E}_{t+1}\left[\overline{\tilde{g}}_{t+1}\right] - \nabla\mathcal{L}_{\mathcal{H}}\left(\theta_{t+1}\right)\right\rangle + 2\beta_t^2\left\langle\delta_t,\,\nabla\mathcal{L}_{\mathcal{H}}(\theta_t) - \nabla\mathcal{L}_{\mathcal{H}}(\theta_{t+1})\right\rangle$$
$$+ 2\beta_t(1-\beta_t)\left\langle\mathbb{E}_{t+1}\left[\overline{\tilde{g}}_{t+1}\right] - \nabla\mathcal{L}_{\mathcal{H}}\left(\theta_{t+1}\right),\,\nabla\mathcal{L}_{\mathcal{H}}(\theta_t) - \nabla\mathcal{L}_{\mathcal{H}}(\theta_{t+1})\right\rangle.$$

Recall that $\overline{\tilde{g}}_{t+1} := \frac{1}{(n-f)}\sum_{j\in\mathcal{H}}\tilde{g}_{t+1}^{(i)}$. Thus, as we ignore clipping by Assumption 2.3, we have $\mathbb{E}_{t+1}\left[\overline{\tilde{g}}_{t+1}\right] = \nabla\mathcal{L}_{\mathcal{H}}(\theta_{t+1})$. Using this above we obtain that

$$\mathbb{E}_{t+1}\left[\|\delta_{t+1}\|^2\right] = \beta_t^2\|\delta_t\|^2 + (1-\beta_t)^2\mathbb{E}_{t+1}\left[\left\|\overline{\tilde{g}}_{t+1} - \nabla\mathcal{L}_{\mathcal{H}}\left(\theta_{t+1}\right)\right\|^2\right] + \beta_t^2\|\nabla\mathcal{L}_{\mathcal{H}}(\theta_t) - \nabla\mathcal{L}_{\mathcal{H}}(\theta_{t+1})\|^2$$
$$+ 2\beta_t^2\left\langle\delta_t,\,\nabla\mathcal{L}_{\mathcal{H}}(\theta_t) - \nabla\mathcal{L}_{\mathcal{H}}(\theta_{t+1})\right\rangle.$$

Also, we bound the expected distance between the average of full honest gradients and the average of mini-batch gradients with additive correlated noise. We first write

$$\overline{\overline{g}}_{t+1} = \frac{1}{|\mathcal{H}|} \sum_{i \in \mathcal{H}} \left( g_{t+1}^{(i)} + \sum_{\substack{j=1 \\ j \neq i}}^{n} v_{t+1}^{(ij)} + \overline{v}_{t+1}^{(i)} \right) = \frac{1}{|\mathcal{H}|} \sum_{i \in \mathcal{H}} g_{t+1}^{(i)} + \frac{1}{|\mathcal{H}|} \sum_{i \in \mathcal{H}} \sum_{\substack{j=1 \\ j \neq i}}^{n} v_{t+1}^{(ij)} + \frac{1}{|\mathcal{H}|} \sum_{i \in \mathcal{H}} \overline{v}_{t+1}^{(i)}$$

$$= \frac{1}{|\mathcal{H}|} \sum_{i \in \mathcal{H}} g_{t+1}^{(i)} + \frac{1}{|\mathcal{H}|} \sum_{i,j \in \mathcal{H}} v_{t+1}^{(ij)} + \frac{1}{|\mathcal{H}|} \sum_{i \in \mathcal{H}} \sum_{j \in [n] \setminus \mathcal{H}} v_{t+1}^{(ij)} + \frac{1}{|S|} \sum_{i \in \mathcal{H}} \overline{v}_{t+1}^{(i)}.$$

We remark that the second term is zero by construction of the correlated noise terms, while all remaining random variables in the third and fourth terms are independently sampled from centered Gaussians with covariance $\sigma_{\text{cor}}^2 I_d$ or $\sigma_{\text{ind}}^2 I_d$. We directly deduce the following:

$$\mathbb{E}\left\| \overline{\overline{g}}_{t+1} - \nabla \mathcal{L}_{\mathcal{H}}(\theta_{t+1}) \right\|^2 = \mathbb{E}\left\| \frac{1}{|\mathcal{H}|} \sum_{i \in \mathcal{H}} g_{t+1}^{(i)} - \nabla \mathcal{L}_{\mathcal{H}}(\theta_{t+1}) + \frac{1}{|\mathcal{H}|} \sum_{i \in \mathcal{H}} \sum_{j \in [n] \setminus \mathcal{H}} v_{t+1}^{(ij)} + \frac{1}{|\mathcal{H}|} \sum_{i \in \mathcal{H}} \overline{v}_{t+1}^{(i)} \right\|$$

$$= \mathbb{E}\left\| \frac{1}{|\mathcal{H}|} \sum_{i \in \mathcal{H}} g_{t+1}^{(i)} - \nabla \mathcal{L}_{\mathcal{H}}(\theta_{t+1}) \right\|^2 + \mathbb{E}\left\| \frac{1}{|\mathcal{H}|} \sum_{i \in \mathcal{H}} \sum_{j \in [n] \setminus \mathcal{H}} v_{t+1}^{(ij)} + \frac{1}{|S|} \sum_{i \in \mathcal{H}} \overline{v}_{t+1}^{(i)} \right\|^2$$

$$= \mathbb{E}\left\| \frac{1}{|\mathcal{H}|} \sum_{i \in \mathcal{H}} g_{t+1}^{(i)} - \nabla \mathcal{L}_{\mathcal{H}}(\theta_{t+1}) \right\|^2 + \frac{df(n-f)\sigma_{\text{cor}}^2}{(n-f)^2} + \frac{d\sigma_{\text{ind}}^2}{n-f}$$

$$\leq \frac{2}{n-f}\left(1 - \frac{b}{m}\right)\frac{\sigma^2}{b} + \frac{df\sigma_{\text{cor}}^2}{n-f} + \frac{d\sigma_{\text{ind}}^2}{n-f},$$

where the last step is due to Lemma C.5 thanks to assumptions 2.2 and 2.3. Now, denote $\overline{\sigma}_{\text{DP}}^2 := 2\left(1 - \frac{b}{m}\right)\frac{\sigma^2}{b} + df\sigma_{\text{cor}}^2 + d\sigma_{\text{ind}}^2$. Thus, we have

$$\mathbb{E}_{t+1}\left[\|\delta_{t+1}\|^2\right] \leq \beta_t^2 \|\delta_t\|^2 + (1-\beta_t)^2 \frac{\overline{\sigma}_{\text{DP}}^2}{(n-f)} + \beta_t^2 \left\|\nabla\mathcal{L}_{\mathcal{H}}(\theta_t) - \nabla\mathcal{L}_{\mathcal{H}}(\theta_{t+1})\right\|^2 + 2\beta_t^2 \left\langle \delta_t,\, \nabla\mathcal{L}_{\mathcal{H}}(\theta_t) - \nabla\mathcal{L}_{\mathcal{H}}(\theta_{t+1})\right\rangle.$$

By the Cauchy-Schwartz inequality, $\left\langle \delta_t,\, \nabla\mathcal{L}_{\mathcal{H}}(\theta_t) - \nabla\mathcal{L}_{\mathcal{H}}(\theta_{t+1})\right\rangle \leq \|\delta_t\| \left\|\nabla\mathcal{L}_{\mathcal{H}}(\theta_t) - \nabla\mathcal{L}_{\mathcal{H}}(\theta_{t+1})\right\|$. Since $\mathcal{L}_{\mathcal{H}}$ is $L$-smooth, we have $\left\|\nabla\mathcal{L}_{\mathcal{H}}(\theta_t) - \nabla\mathcal{L}_{\mathcal{H}}(\theta_{t+1})\right\| \leq L\left\|\theta_{t+1} - \theta_t\right\|$. Recall from (18) that $\theta_{t+1} = \theta_t - \gamma_t R_t$. Thus, $\left\|\nabla\mathcal{L}_{\mathcal{H}}(\theta_t) - \nabla\mathcal{L}_{\mathcal{H}}(\theta_{t+1})\right\| \leq \gamma_t L \|R_t\|$. Using this above we obtain that

$$\mathbb{E}_{t+1}\left[\|\delta_{t+1}\|^2\right] \leq \beta_t^2 \|\delta_t\|^2 + (1-\beta_t)^2 \frac{\overline{\sigma}_{\text{DP}}^2}{(n-f)} + \gamma_t^2 \beta_t^2 L^2 \|R_t\|^2 + 2\gamma_t \beta_t^2 L \|\delta_t\| \|R_t\|.$$

As $2ab \leq a^2 + b^2$, from above we obtain that

$$\mathbb{E}_{t+1}\left[\|\delta_{t+1}\|^2\right] \leq \beta_t^2 \|\delta_t\|^2 + (1-\beta_t)^2 \frac{\overline{\sigma}_{\text{DP}}^2}{(n-f)} + \gamma_t^2 \beta_t^2 L^2 \|R_t\|^2 + \gamma_t L \beta_t^2 \left(\|\delta_t\|^2 + \|R_t\|^2\right)$$

$$= (1 + \gamma_t L)\beta_t^2 \|\delta_t\|^2 + (1-\beta_t)^2 \frac{\overline{\sigma}_{\text{DP}}^2}{(n-f)} + \gamma_t L(1 + \gamma_t L)\beta_t^2 \|R_t\|^2. \tag{60}$$

By definition of $\epsilon_t$ in (21), we have $R_t = \epsilon_t + \overline{m}_t$. Thus, owing to the triangle inequality and the fact that $2ab \leq a^2 + b^2$, we have $\|R_t\|^2 \leq 2\|\epsilon_t\|^2 + 2\|\overline{m}_t\|^2$. Similarly, by definition of $\delta_t$ in (20), we have $\|\overline{m}_t\|^2 \leq 2\|\delta_t\|^2 + 2\|\nabla\mathcal{L}_{\mathcal{H}}(\theta_t)\|^2$. Thus, $\|R_t\|^2 \leq 2\|\epsilon_t\|^2 + 4\|\delta_t\|^2 + 4\|\nabla\mathcal{L}_{\mathcal{H}}(\theta_t)\|^2$. Using this in (60) we obtain that

$$\mathbb{E}_{t+1}\left[\|\delta_{t+1}\|^2\right] \leq (1 + \gamma_t L)\beta_t^2 \|\delta_t\|^2 + (1-\beta_t)^2 \frac{\overline{\sigma}_{\text{DP}}^2}{(n-f)}$$

$$+ 2\gamma_t L(1 + \gamma_t L)\beta_t^2 \left(\|\epsilon_t\|^2 + 2\|\delta_t\|^2 + 2\|\nabla\mathcal{L}_{\mathcal{H}}(\theta_t)\|^2\right).$$

By rearranging the terms on the R.H.S., we get

$$\mathbb{E}_{t+1}\left[\|\delta_{t+1}\|^2\right] \le \beta_t^2(1+\gamma_t L)(1+4\gamma_t L)\|\delta_t\|^2 + 4\gamma_t L(1+\gamma_t L)\beta_t^2\|\nabla\mathcal{L}_{\mathcal{H}}(\theta_t)\|^2 + (1-\beta_t)^2\frac{\overline{\sigma}_{\mathrm{DP}}^2}{(n-f)}$$
$$+ 2\gamma_t L(1+\gamma_t L)\beta_t^2\|\epsilon_t\|^2.$$

The proof concludes upon taking total expectation on both sides. $\qquad\square$

### C.6.4. PROOF OF LEMMA C.3

**Lemma C.3.** *Assume that $\mathcal{L}_{\mathcal{H}}$ is $L$-smooth. Consider Algorithm 1. For any $t \in [T]$, we have*

$$\mathbb{E}\left[\mathcal{L}_{\mathcal{H}}(\theta_{t+1}) - \mathcal{L}_{\mathcal{H}}(\theta_t)\right] \le -\frac{\gamma_t}{2}(1-4\gamma_t L)\,\mathbb{E}\left[\|\nabla\mathcal{L}_{\mathcal{H}}(\theta_t)\|^2\right] + \gamma_t(1+2\gamma_t L)\,\mathbb{E}\left[\|\delta_t\|^2\right] + \gamma_t(1+\gamma_t L)\,\mathbb{E}\left[\|\epsilon_t\|^2\right].$$

*Proof.* Let $t \in \{0, \ldots, T-1\}$. Assuming $\mathcal{L}_{\mathcal{H}}$ is $L$-smooth, we have (see Lemma 1.2.3 (Nesterov et al., 2018))

$$\mathcal{L}_{\mathcal{H}}(\theta_{t+1}) - \mathcal{L}_{\mathcal{H}}(\theta_t) \le \langle\theta_{t+1}-\theta_t, \nabla\mathcal{L}_{\mathcal{H}}(\theta_t)\rangle + \frac{L}{2}\|\theta_{t+1}-\theta_t\|^2.$$

Substituting from (22), i.e., $\theta_{t+1} = \theta_t - \gamma_t\overline{m}_t - \gamma_t\epsilon_t$, we obtain that

$$\mathcal{L}_{\mathcal{H}}(\theta_{t+1}) - \mathcal{L}_{\mathcal{H}}(\theta_t) \le -\gamma_t\langle\overline{m}_t, \nabla\mathcal{L}_{\mathcal{H}}(\theta_t)\rangle - \gamma_t\langle\epsilon_t, \nabla\mathcal{L}_{\mathcal{H}}(\theta_t)\rangle + \gamma_t^2\frac{L}{2}\|\overline{m}_t+\epsilon_t\|^2$$
$$= -\gamma_t\langle\overline{m}_t - \nabla\mathcal{L}_{\mathcal{H}}(\theta_t)+\nabla\mathcal{L}_{\mathcal{H}}(\theta_t), \nabla\mathcal{L}_{\mathcal{H}}(\theta_t)\rangle - \gamma_t\langle\epsilon_t, \nabla\mathcal{L}_{\mathcal{H}}(\theta_t)\rangle + \gamma_t^2\frac{L}{2}\|\overline{m}_t+\epsilon_t\|^2.$$

By Definition (20), $\overline{m}_t - \nabla\mathcal{L}_{\mathcal{H}}(\theta_t) = \delta_t$. Thus, from above we obtain

$$\mathcal{L}_{\mathcal{H}}(\theta_{t+1}) - \mathcal{L}_{\mathcal{H}}(\theta_t) \le -\gamma_t\|\nabla\mathcal{L}_{\mathcal{H}}(\theta_t)\|^2 - \gamma_t\langle\delta_t, \nabla\mathcal{L}_{\mathcal{H}}(\theta_t)\rangle - \gamma_t\langle\epsilon_t, \nabla\mathcal{L}_{\mathcal{H}}(\theta_t)\rangle + \frac{1}{2}\gamma_t^2 L\|\overline{m}_t+\epsilon_t\|^2. \quad (61)$$

Now, we consider the last three terms on the R.H.S. separately. Using Cauchy-Schwartz inequality, and the fact that $2ab \le \frac{1}{c}a^2 + cb^2$ for any $c > 0$, we obtain that (by substituting $c = 2$)

$$2|\langle\delta_t, \nabla\mathcal{L}_{\mathcal{H}}(\theta_t)\rangle| \le 2\|\delta_t\|\|\nabla\mathcal{L}_{\mathcal{H}}(\theta_t)\| \le \frac{2}{1}\|\delta_t\|^2 + \frac{1}{2}\|\nabla\mathcal{L}_{\mathcal{H}}(\theta_t)\|^2. \quad (62)$$

Similarly,

$$2|\langle\epsilon_t, \nabla\mathcal{L}_{\mathcal{H}}(\theta_t)\rangle| \le 2\|\epsilon_t\|\|\nabla\mathcal{L}_{\mathcal{H}}(\theta_t)\| \le \frac{2}{1}\|\epsilon_t\|^2 + \frac{1}{2}\|\nabla\mathcal{L}_{\mathcal{H}}(\theta_t)\|^2. \quad (63)$$

Finally, using triangle inequality and the fact that $2ab \le a^2 + b^2$ we have

$$\|\overline{m}_t+\epsilon_t\|^2 \le 2\|\overline{m}_t\|^2 + 2\|\epsilon_t\|^2 = 2\|\overline{m}_t - \nabla\mathcal{L}_{\mathcal{H}}(\theta_{t+1})+\nabla\mathcal{L}_{\mathcal{H}}(\theta_t)\|^2 + 2\|\epsilon_t\|^2$$
$$\le 4\|\delta_t\|^2 + 4\|\nabla\mathcal{L}_{\mathcal{H}}(\theta_t)\|^2 + 2\|\epsilon_t\|^2. \qquad [\text{since } \overline{m}_t - \nabla\mathcal{L}_{\mathcal{H}}(\theta_t) = \delta_t] \quad (64)$$

Substituting from (62), (63) and (64) in (61) we obtain that

$$\mathcal{L}_{\mathcal{H}}(\theta_{t+1}) - \mathcal{L}_{\mathcal{H}}(\theta_t) \le -\gamma_t\|\nabla\mathcal{L}_{\mathcal{H}}(\theta_t)\|^2 + \frac{1}{2}\gamma_t\left(2\|\delta_t\|^2 + \frac{1}{2}\|\nabla\mathcal{L}_{\mathcal{H}}(\theta_t)\|^2\right) + \frac{1}{2}\gamma_t\left(2\|\epsilon_t\|^2 + \frac{1}{2}\|\nabla\mathcal{L}_{\mathcal{H}}(\theta_t)\|^2\right)$$
$$+ \frac{1}{2}\gamma_t^2 L\left(4\|\delta_t\|^2 + 4\|\nabla\mathcal{L}_{\mathcal{H}}(\theta_t)\|^2 + 2\|\epsilon_t\|^2\right).$$

Upon rearranging the terms in the R.H.S., we obtain that

$$\mathcal{L}_{\mathcal{H}}(\theta_{t+1}) - \mathcal{L}_{\mathcal{H}}(\theta_t) \le -\frac{\gamma_t}{2}(1-4\gamma_t L)\|\nabla\mathcal{L}_{\mathcal{H}}(\theta_t)\|^2 + \gamma_t(1+2\gamma_t L)\|\delta_t\|^2 + \gamma_t(1+\gamma_t L)\|\epsilon_t\|^2.$$

This concludes the proof. $\qquad\square$

# D. Additional Experimental Results

In this section, we provide additional results that were not included in the main paper due to space constraints. Appendix D.1 contains the complete set of results for the evaluation of CAFCOR across the four privacy threat models, under all four attack types, and for both values of $f = 5, 10$, on the MNIST and Fashion-MNIST datasets. Additionally, Appendix D.2 presents the full results comparing CAF with other robust aggregation methods under all four attack types, across various values of $f$, data heterogeneity settings, and on both MNIST and Fashion-MNIST.

## D.1. CAFCOR Under Three Privacy Threat Models

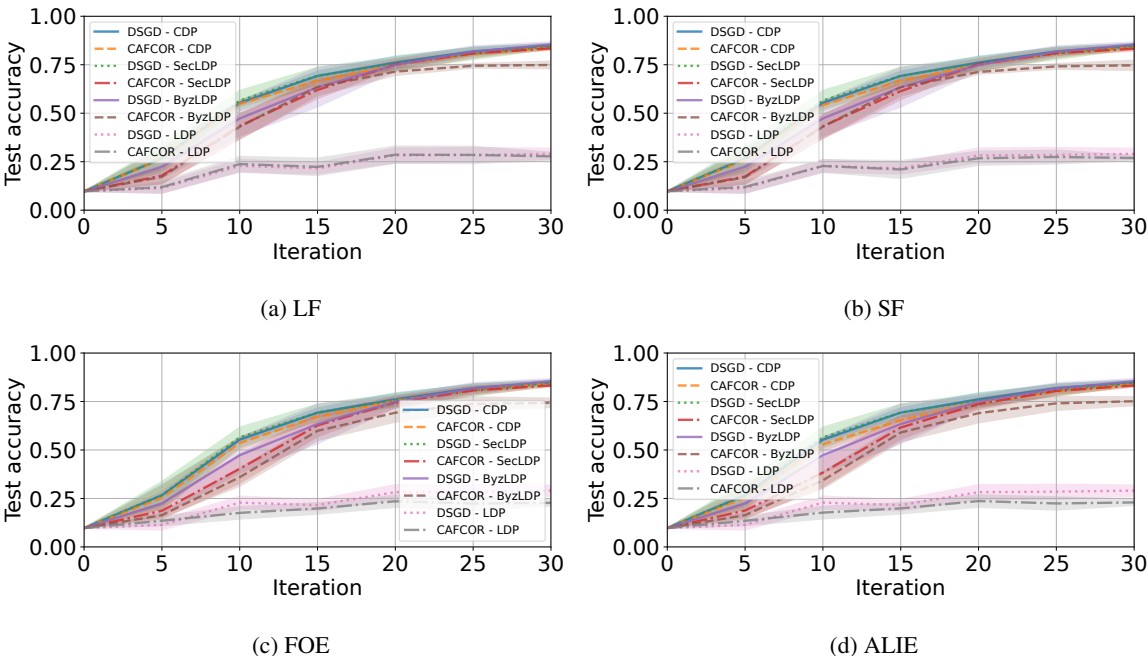

(a) LF

(b) SF

(c) FOE

(d) ALIE

Figure 3: Comparison of CAFCOR and DSGD under four privacy threat models on MNIST. There are $n = 100$ workers, including $f = 5$ malicious workers executing four attacks. A homogeneous data distribution is considered among honest workers. User-level DP is used and the aggregate privacy budget is $(\epsilon, \delta) = (27.8, 10^{-4})$.

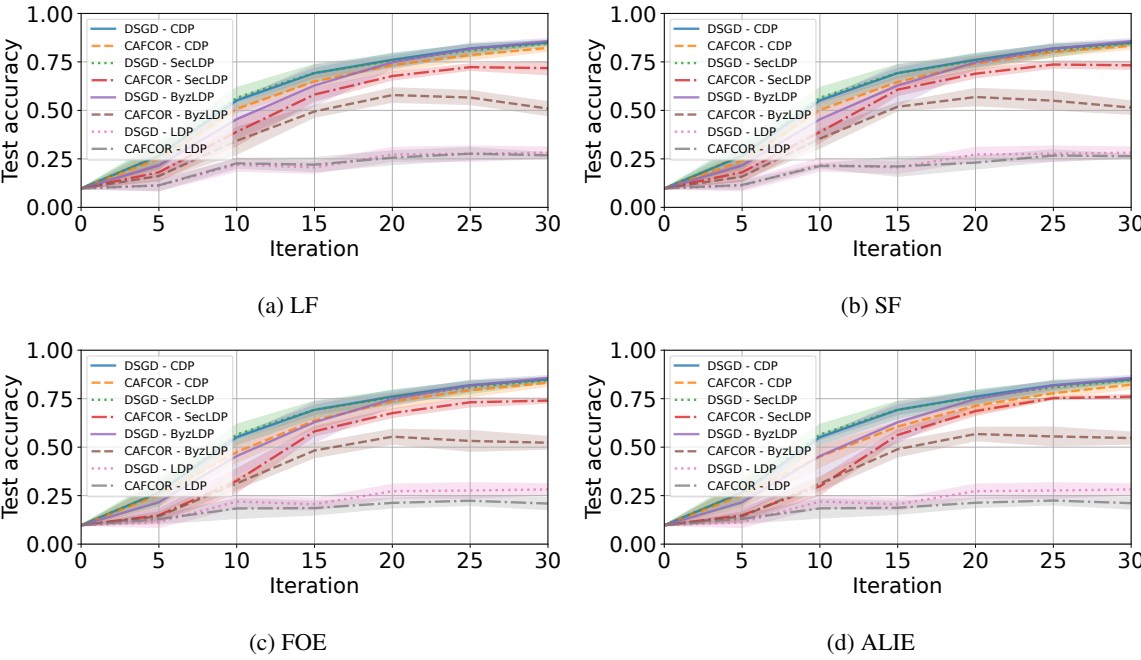

(a) LF

(b) SF

(c) FOE

(d) ALIE

Figure 4: Comparison of CAFCOR and DSGD under four privacy threat models on MNIST. There are $n = 100$ workers, including $f = 10$ malicious workers executing four attacks. A homogeneous data distribution is considered among honest workers. User-level DP is used and the aggregate privacy budget is $(\epsilon, \delta) = (26.4, 10^{-4})$.

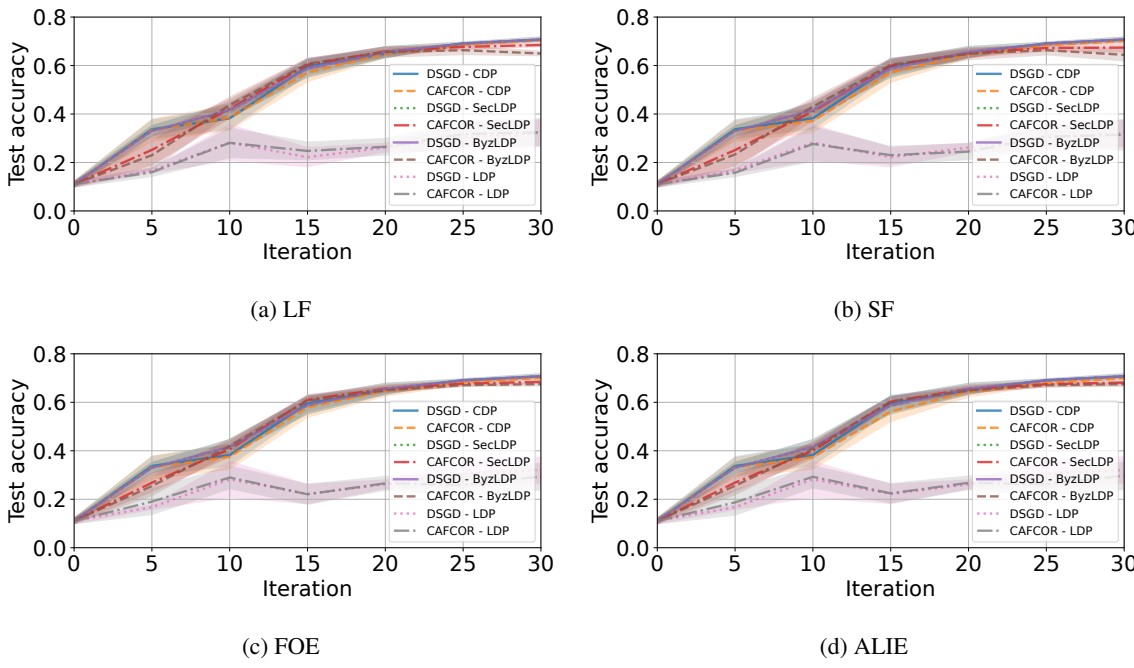

(a) LF

(b) SF

(c) FOE

(d) ALIE

Figure 5: Comparison of CAFCOR and DSGD under four privacy threat models on Fashion-MNIST. There are $n = 100$ workers, including $f = 5$ malicious workers executing four attacks. A homogeneous data distribution is considered among honest workers. User-level DP is used and the aggregate privacy budget is $(\epsilon, \delta) = (39.6, 10^{-4})$.

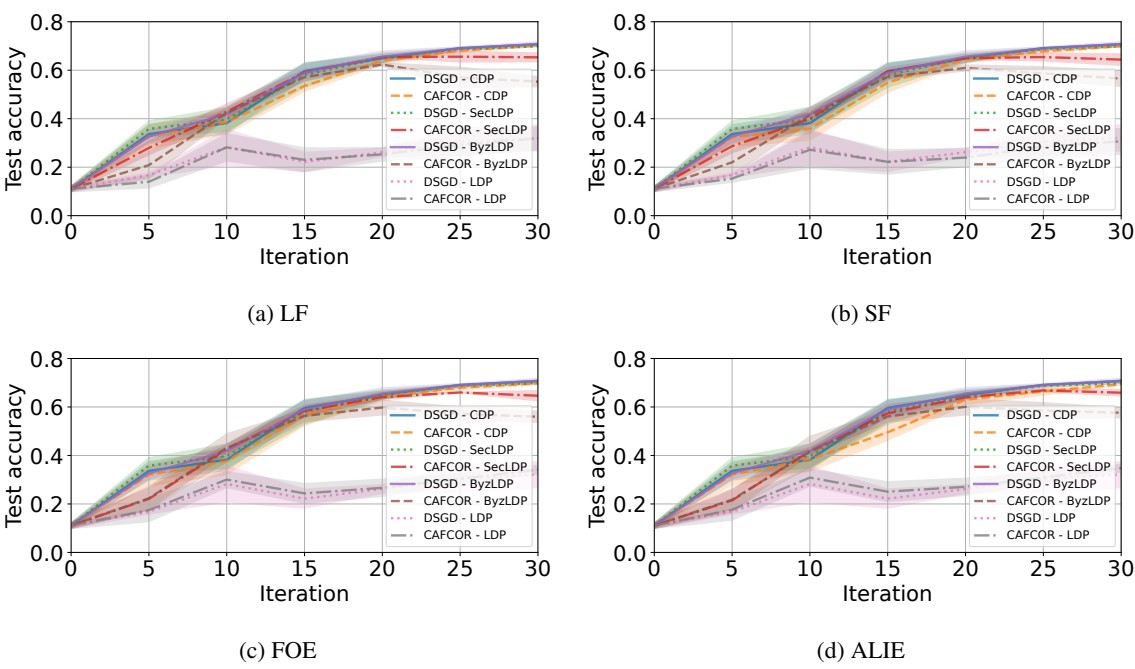

(a) LF

(b) SF

(c) FOE

(d) ALIE

Figure 6: Comparison of CAFCOR and DSGD under four privacy threat models on Fashion-MNIST. There are $n = 100$ workers, including $f = 10$ malicious workers executing four attacks. A homogeneous data distribution is considered among honest workers. User-level DP is used and the aggregate privacy budget is $(\epsilon, \delta) = (39.6, 10^{-4})$.

## D.2. CAF vs. Robust Aggregators

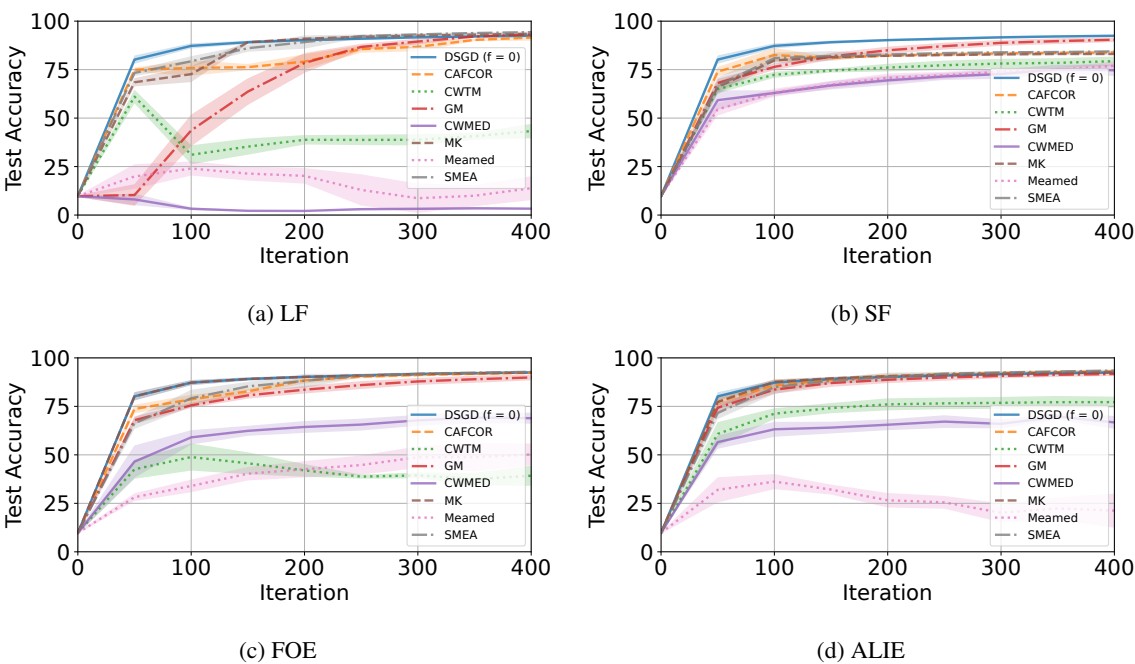

(a) LF

(b) SF

(c) FOE

(d) ALIE

Figure 7: Performance of CAFCOR versus standard robust algorithms on MNIST. There is $f = 1$ malicious worker among $n = 11$ workers, and we consider extreme heterogeneity among honest workers. The CDP privacy model is considered under example-level DP, and the privacy budget is $(\varepsilon, \delta) = (13.7, 10^{-4})$ throughout.

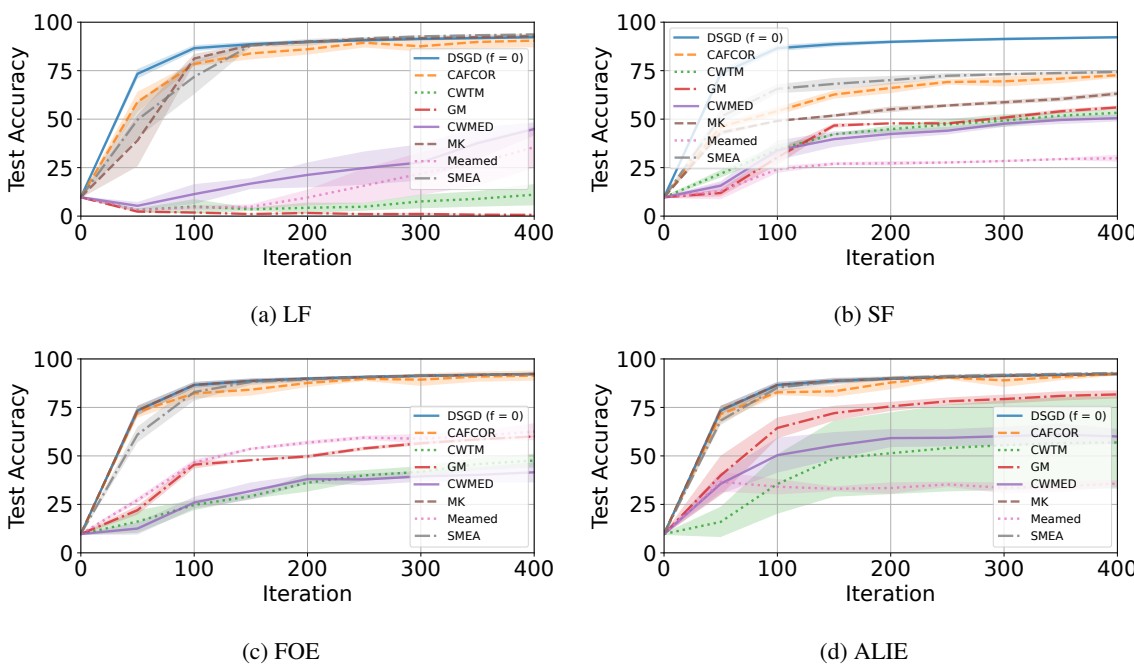

(a) LF

(b) SF

(c) FOE

(d) ALIE

Figure 8: Performance of CAFCOR versus standard robust algorithms on MNIST. There are $f = 3$ malicious workers among $n = 13$ and $\alpha = 0.1$. The CDP privacy model is considered under example-level DP, and the privacy budget is $(\varepsilon, \delta) = (13.7, 10^{-4})$ throughout.

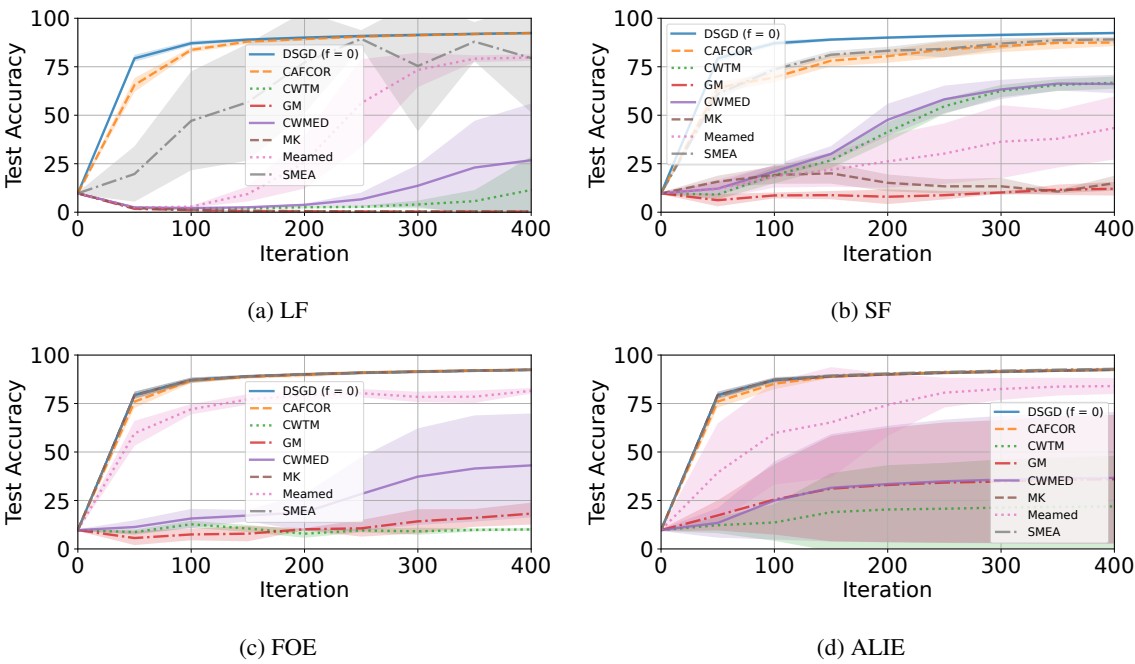

Figure 9: Performance of CAFCOR versus standard robust algorithms on MNIST. There are $f = 5$ malicious workers among $n = 15$ and $\alpha = 1$. The CDP privacy model is considered under example-level DP, and the privacy budget is $(\varepsilon, \delta) = (13.7, 10^{-4})$ throughout.

**Comparison with SMEA (Allouah et al., 2023c)** We also included in our experiments a comparison with the SMEA robust aggregation method on the MNIST dataset (see Figures 7, 8, and 9). These results show that SMEA matches CAF's accuracy, significantly outperforming other standard robust methods, and even approaching the averaging baseline. This observation aligns with our theoretical findings (see Proposition 4.1 and subsequent discussion).

However, we emphasize that SMEA incurs a substantial computational cost, which further supports the empirical advantage of CAF, despite the two methods offering similarly strong robustness in practice. To highlight SMEA's prohibitive computational complexity, we report runtime ratios relative to simple averaging (lower is better), with $n = 30$ workers and $f = 3$ malicious workers, on models of varying dimension $d$.

| Dimension ($d$) | SMEA | CAF | Meamed | GM |
|---|---|---|---|---|
| $0.25 \times 10^7$ | 30,251 | **28** | 112 | 62 |
| $0.5 \times 10^7$ | 61,142 | **51** | 197 | 78 |
| $1 \times 10^7$ | 117,255 | **100** | 378 | 126 |

Table 1: Computation time (relative to simple averaging) across different dimensions for various aggregation methods.

As seen in Table 1, CAF dramatically outperforms SMEA in terms of runtime, and also maintains a clear advantage over other robust aggregation baselines such as Meamed and GM. Combined with CAF's superior performance on test accuracy, these results underscore CAFCOR's practical relevance for trustworthy federated learning.

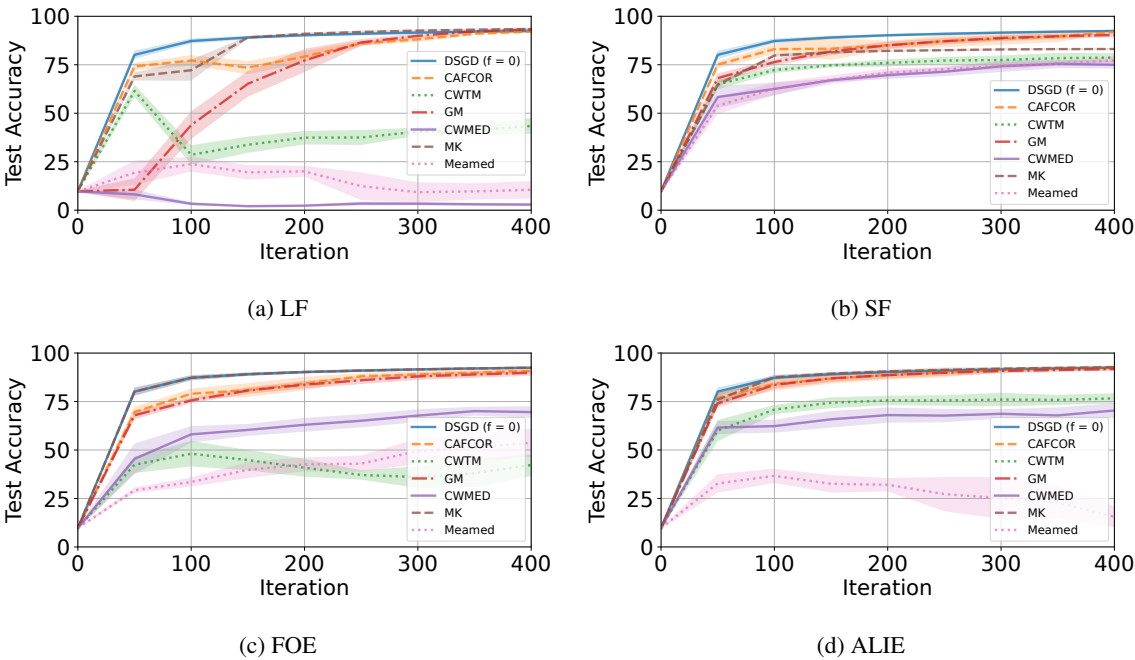

(a) LF

(b) SF

(c) FOE

(d) ALIE

Figure 10: Performance of CAFCOR versus standard robust algorithms on Fashion-MNIST. There are $f = 1$ malicious worker among $n = 11$, and we consider extreme heterogeneity among honest workers. The CDP privacy model is considered under example-level DP, and the privacy budget is $(\varepsilon, \delta) = (13.7, 10^{-4})$ throughout.

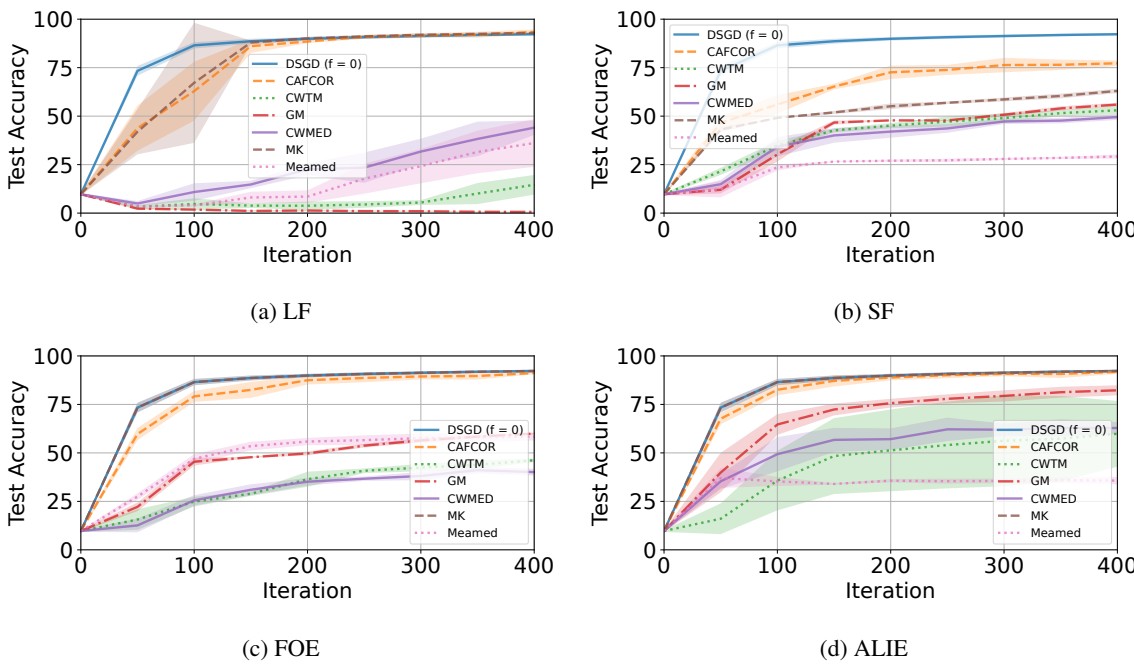

(a) LF

(b) SF

(c) FOE

(d) ALIE

Figure 11: Performance of CAFCOR versus standard robust algorithms on Fashion-MNIST. There are $f = 3$ malicious workers among $n = 13$ and $\alpha = 0.1$. The CDP privacy model is considered under example-level DP, and the privacy budget is $(\varepsilon, \delta) = (13.7, 10^{-4})$ throughout.

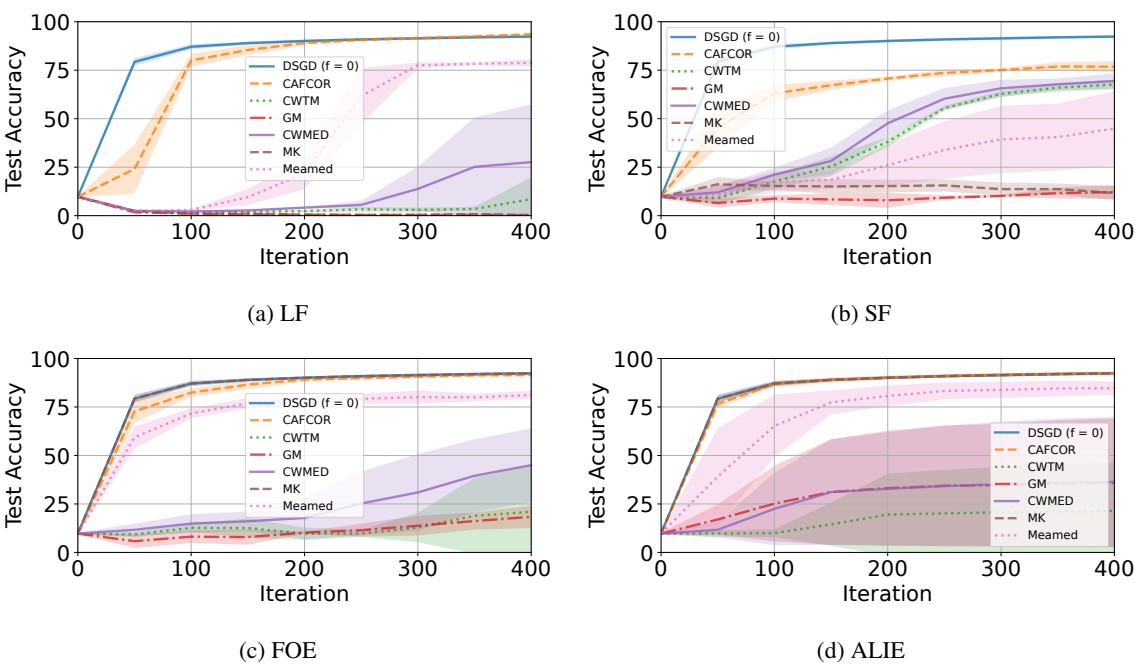

(a) LF

(b) SF

(c) FOE

(d) ALIE

Figure 12: Performance of CAFCOR versus standard robust algorithms on Fashion-MNIST. There are $f = 5$ malicious workers among $n = 15$ and $\alpha = 1$. The CDP privacy model is considered under example-level DP, and the privacy budget is $(\varepsilon, \delta) = (13.7, 10^{-4})$ throughout.

