# OpenReview forum: "Towards Trustworthy Federated Learning with Untrusted Participants"
_ICML.cc/2025/Conference — ICML 2025 poster_

### Official Review · Reviewer_3HsQ · 2025-03-10

**Overall Recommendation:** 2

**Summary:**

Achieving robustness against Byzantine workers and preserving data privacy are two important objectives in distributed learning. Existing work primarily studies each problem separately, and achieving both simultaneously is a challenging task. In this paper, the authors propose CAFCOR, an algorithm designed to achieve both robustness and data privacy without relying on a fully trusted central server. The CAFCOR algorithm combines robust gradient aggregation with correlated noise injection, using shared randomness between workers. In this setting, each worker perturbs its gradient update using a combination of independent noise and correlated noise. The server then applies CAF (covariance-bound agnostic filter), which adjusts weights based on variance contributions to suppress the influence of Byzantine workers. The authors demonstrate that CAFCOR achieves privacy guarantees comparable to central differential privacy while maintaining robustness against Byzantine attacks.

**Claims And Evidence:**

I think the claims are generally well-supported. However, please see the Experimental Designs or Analyses section.

## update after rebuttal

I thank the authors for their responses.
While some of my concerns have been addressed, my main concern regarding the limited scope of the experiments remains unresolved. I think this paper has the potential to make a significant contribution, but it would require a major revision to reach that stage. I have decided to lower my initial rating.

**Essential References Not Discussed:**

N/A

**Experimental Designs Or Analyses:**

- The experiments are weak. Even considering the theoretical contributions of the paper, evaluating on MNIST and Fashion-MNIST is insufficient. To demonstrate the practical utility of the algorithm, the authors should perform experiments on more challenging real-world datasets (e.g., ImageNet, CIFAR-100).

**Methods And Evaluation Criteria:**

- The authors introduce a refined version of secret-based local differential privacy (SecLDP), which extends traditional LDP by assuming that workers share randomness. This relaxation of the trust assumption in CDP seems to improve utility over LDP models while still maintaining privacy.
- The authors provide a theoretical analysis of the privacy-utility trade-off and robustness against Byzantine workers achieved by CAFCOR.
- The proposed CAF algorithm appears to be sensible and with theoretical guarantees. The requirement of shared randomness, achieved through a one-time encrypted communication round, seems reasonable.

- The complexity of the proposed CAF aggregation appears to be high, making it impractical for high-dimensional models, e.g., deep learning. While the authors mention a power method-based approximation, its practical applicability in large-scale problems needs to be verified.
- The assumption of bounded heterogeneity seems to limit the applicability. In real-world distributed learning scenarios, data can be highly heterogeneous.

**Other Comments Or Suggestions:**

- Gradient clipping appears to be performed at the client-side, it is not clear why this is necessary. Given that gradient clipping could potentially be handled at the server-side, requiring additional client-side operations beyond momentum generation may reduce the practical applicability of CAFCOR.
- Please explicitly define the nature of Byzantine workers. Do they have access to the entire learning process, or are there any limitations on their capabilities?
- Please clarify how data heterogeneity is modeled in the experiments. Specifically, how are the datasets distributed among clients, and what level of heterogeneity is introduced?

**Other Strengths And Weaknesses:**

N/A

**Questions For Authors:**

N/A

**Relation To Broader Scientific Literature:**

Please see the Summary section.

**Theoretical Claims:**

I checked briefly the main theoretical results but not their proofs.

---

> ### Author Rebuttal · Authors · 2025-03-31
>
> We thank the reviewer for their insightful feedback. Below, we address the key points raised:
>
> ### **Experimental Scope**
>
> > evaluating on MNIST and Fashion-MNIST is insufficient [...]
>
> The reviewer’s suggestion to evaluate additional datasets is highly valid. We stress, however, that our current evaluation on MNIST and Fashion-MNIST is already extensive, rigorously testing our algorithm CafCor across multiple threat models, robust aggregation methods, and varying data heterogeneity levels (Section 5). Crucially, our setting is quite *challenging* even on these standard datasets, since we enforce **both** differential privacy without trusting the server and Byzantine resilience under state-of-the-art attacks. Indeed, as Figure 2 clearly illustrates, prior state-of-the-art methods suffer substantial accuracy losses under these combined constraints. Also, the SOTA method by Allouah et al. (2023b, ICML) could only scale to small logistic regression tasks with weaker theoretical guarantees, highlighting the novelty and practical significance of our results.
>
> We recall that our core contribution is a rigorous theoretical framework that significantly advances the resolution of the privacy–robustness–utility trilemma (Allouah et al. 2023b, ICML), with minimal trust assumptions. Theoretically, we approach the minimax-optimal privacy-utility trade-off achievable in the ideal scenario (trusted server, no Byzantine adversaries—Corollary 4.1). Empirically, CafCor closely matches this optimal baseline even under Byzantine threats (Figure 1). Extending to larger datasets such as ImageNet is an important and exciting future step, which we will explicitly mention.
>
> ### **Computational Complexity**
>
> > The complexity of the proposed CAF aggregation appears to be high, making it impractical for high-dimensional models, e.g., deep learning
>
> We thank the reviewer for highlighting this important aspect. CafCor’s aggregation complexity, while higher than simple averaging, is significantly lower than previous state-of-the-art methods like SMEA, whose runtime complexity is exponential in the number of Byzantine workers $f$. Specifically, SMEA performs an exhaustive subset search across subsets of size $n-f$, making it computationally infeasible. In contrast, our covariance-based CAF aggregation with power-method approximation achieves an efficient runtime of $\mathcal{O}(f n d \log d)$, enabling scalability to high-dimensional models, far beyond the closest prior work (Allouah et al. 2023b), which only scaled to small logistic regression tasks.
>
> We refer the reviewer to new experiments, included in our response to Reviewer 6PZT due to space constraints, that explicitly demonstrate this complexity advantage over SMEA, and will include these new experiments in the revision.
>
> ### **Clarifications on Definitions and Methods**
>
> > Gradient clipping appears to be performed at the client-side, it is not clear why this is necessary.
>
> > Please explicitly define the nature of Byzantine workers
>
> > Please clarify how data heterogeneity is modeled in the experiments
>
> We thank the reviewer for this helpful request for clarity.
> Byzantine workers are defined as workers capable of arbitrary deviation from the protocol and collusion, with full knowledge of the algorithm. Data heterogeneity is modeled through a Dirichlet distribution (following Hsu et al. 2019, see Section 5), simulating realistic conditions.
> Besides, bounded heterogeneity is a common assumption in distributed learning for convergence analyses (Karimireddy et al. 2022, Farhadkhani et al. 2022, Allouah et al. 2023b), although heterogeneity is not the focus of our current work.
> Finally, gradient clipping at the client side is standard practice, essential for differential privacy guarantees, and carries low computational overhead.

---

### Official Review · Reviewer_6PZT · 2025-03-10

**Overall Recommendation:** 4

**Summary:**

The paper proposes a technique to perform distributed mean estimation with differential privacy guarantees and robustness to byzantine participants. To achieve privacy, if first adopts the anti-correlated noise method of [1,2]. To achieve robustness, it uses the empirical covariance matrix of the contributions to filter byzantine inputs, a technique which is specially tailored to work under correlated DP noise. The resulting protocol inherits the accuracy improvement that approaches central DP of [1,2] while improving the resilience to malicious participants of related byzantine aggregation techniques that are less prepared to work under DP noise.

References (will be also used below):

[1] Sabater, César, Aurélien Bellet, and Jan Ramon. "An accurate, scalable and verifiable protocol for federated differentially private averaging." Machine Learning 111.11 (2022): 4249-4293.

[2] Allouah, Youssef, et al. "The privacy power of correlated noise in decentralized learning." arXiv preprint arXiv:2405.01031 (2024).

[3] Allouah, Youssef, et al. "On the privacy-robustness-utility trilemma in distributed learning." International Conference on Machine Learning. PMLR, 2023.

## update

The authors have successfully addressed all my concerns and therefore I raise my score. As long as the discussed aspects are clarified, I think the paper deserves to be accepted.

**Claims And Evidence:**

The main claim of the paper is that the proposed byzantine aggregation technique outperforms previous techniques for FL updates that are hidden under correlated DP noise. This is theoretically and empirically backed for standard aggregation techniques.

However, it is not clear why the SMEA aggregation method [3] is not included in the comparison. Theoretically, it seems that SMEA has the same robustness as CafCor. However, it is not present in the comparison of the empirical evaluation.

Therefore, the claim is not completely backed for related previous aggregation techniques.

**Essential References Not Discussed:**

To the best of my knowledge, I don't see any key reference that has been ignored.

**Experimental Designs Or Analyses:**

The experimental design is reasonable in general. However there are a few issues:
- The number of runs of the protocol to get the confidence intervals on the reported accuracy is rather small (5 seeds).
- I am not really sure why Opacus is used to estimate the privacy budget if the paper provides a clear privacy analysis which gives explicit $\epsilon$ and $\delta$ after execution

**Methods And Evaluation Criteria:**

The paper provides a theoretical analysis and an experimental evaluation.

The experimental analysis is performed for using MNIST and fashion-MNIST datasets under homogeneous and heterogeneous distributions, which is reasonable. Attacks also seem reasonable.

The theoretical analysis of utility/robustness (i.e. Proposition 4.1) is reasonable as it provides clear comparisons with previous techniques. The rest of the theoretical results have a less clear message. First, I am not sure why a new privacy analysis is necessary, as current trade-offs are inherited from [1,2] and the novel part of the contribution (i.e. the CAF filter) is a post processing of such protocols. Second, I do not see a clear take-away from the convergence analysis that relates to the claims of the paper, whose results and techniques seem standard.

**Other Comments Or Suggestions:**

I have no further comments.

**Other Strengths And Weaknesses:**

- The communication cost is not discussed: implementing the pairwise shared randomness requires that all parties communicate with each other to minimize $\sigma_{cor}^2$. This incurs in a high communication cost ($O(n^2)$ messages). Considering that the authors claim efficiency as a feature of the protocol, this should be brought to the table.

**Questions For Authors:**

Please address the raised points with respect to the comparison with SMEA [3] and [1,2] raised in my review.

**Relation To Broader Scientific Literature:**

- The contributions are well positioned with respect to what are called within the paper as standard byzantine aggregation techniques: coordinate-wise trimmed mean and median (Yin et al., 2018), geometric median (Chen et al., 2017), Multi-Krum (Blanchard et al., 2017), and mean around median (Xie et al., 2018). However, as said before, the contribution is not well positioned with respect to the resilience of SMEA [3]. In addition, it is not sufficiently clear how different is the proposed technique with respect to SMEA.

- The paper states that previous literature has not studied the resilience of correlated DP noise to malicious participants. This does not seem to be true, as [1] studies cryptographic techniques (in particular, zero knowledge proofs) to deal with malicious participants under privacy constraints. The paper should broadly position to such techniques.

- The paper claims that [1] does not use Renyi divergences for their privacy guarantees. I am not sure why that is a disadvantage. Renyi divergences are a tool for privacy accounting but it is not mandatory to use it as long as guarantees are provided. What makes the protocol private are the obtained with $(\epsilon, \delta)$ parameters and their trade-off with respect to $\sigma_{cor}^2$ and $\sigma_{ind}^2$. Moreover, [1] provides equal or better trade-offs between privacy and the variance of the correlated noise $\sigma_{cor}^2$ than [2] for the extreme cases of communication topologies, which suggests that the privacy analysis is tighter.

**Theoretical Claims:**

I have checked the proofs theoretical claims of Theorem 4.1 and Proposition 4.1 and I did not see problems. Proofs of Theorems 4.2 and Corollary 4.1 seem in shape, but I have not checked all the derivations of the appendix related to them.

---

> ### Author Rebuttal · Authors · 2025-03-31
>
> We thank the reviewer for their detailed and thoughtful feedback. Below, we address the points raised:
>
> ###  **Empirical Comparison with SMEA**
>
> > it is not clear why [...] SMEA [3] is not included in the comparison. Theoretically, it seems that SMEA has the same robustness as CafCor.
>
> We thank the reviewer for this valuable suggestion. As noted, SMEA (Allouah et al., 2023b) was initially excluded from our main experiments due to its exponential complexity. Specifically, SMEA exhaustively searches all subsets of size $n - f$ for the minimal covariance top eigenvalue, incurring prohibitive computational costs. In contrast, our CAF aggregation method achieves a runtime of $\mathcal{O}(f n d \log d)$ using an iterative spectral reweighting scheme.
>
> **1\. Computational Complexity Comparison**
>
> To explicitly demonstrate SMEA's prohibitive computational complexity, we report runtime ratios relative to simple averaging (smaller is better), with $n=30$ workers and $f=3$ Byzantine workers, on models of varying dimension $d$.
>
> | Dimension ($d$) | SMEA         | CAF   | MeaMed | Geometric Median (GM) |
> |---------------------|--------------|-------|--------|------------------|
> | $2.5 \times 10^6$ | 30,251       | **28**| 112    | 62               |
> | $5 \times 10^6$   | 61,142       | **51**| 197    | 78               |
> | $10^7$            | 117,255      | **100**| 378   | 126              |
>
> CAF clearly achieves a dramatically lower complexity than SMEA and can even be more efficient than standard aggregations such as MeaMed and Geometric Median.
>
> **2\. Utility Comparison**
>
> We also execute SMEA in the setting of Figure 2 (MNIST), where we consider a small system comprised only of $n=15$ and $f=5$ Byzantine workers.
>
> | Method                 | Accuracy (\%)        |
> |------------------------|----------------------|
> | Averaging (no attack)  | $93.05$     |
> | SMEA                   | $89.96$     |
> | **CAF**                | ${\bf 89.60}$     |
> | MeaMed                 | $44.81$     |
> | Median (CWMED)                 | $29.66$      |
> | Trimmed mean (CWTM)          | $12.74$     |
> | Geometric Median (GM)                 | $9.73$     |
>
>
> CAF matches SMEA's accuracy, significantly outperforming other standard robust methods, and approaching the averaging baseline (thanks also to our correlated noise scheme). This result aligns with our theory (Proposition 4.1), as the reviewer helpfully pointed out.
>
> We will clearly highlight these new results in the revised manuscript.
>
> ###  **Theoretical Analysis**
>
> > The theoretical analysis [...] is reasonable [...]  First, I am not sure why a new privacy analysis is necessary [...]. Second, I do not see a clear take-away from the convergence analysis [...]
>
> This comment is insightful. Our privacy analysis (Theorem 4.1) is distinct from [1,2], addressing a new threat model where Byzantine adversaries partially collude with the untrusted server. A notable insight (Section 4.1) is that honest workers require no independent noise injection when a non-colluding Byzantine worker exists. Further, Proposition 4.1 alone cannot derive our final privacy-utility result (Corollary 4.1); Theorem 4.2 is essential to quantify correlated noise effects across iterations, controlled by CAF aggregation and local momentum in CafCor. These dependencies were not analyzed in [1,2]. We will explicitly clarify these points.
>
> ###  **Positioning relative to Reference [1]**
>
> > [1] studies cryptographic techniques [...] to deal with malicious participants [...]. The paper should broadly position to such techniques.
>
> Reference [1], included in our paper, uses cryptographic checks (e.g., correct computation, bounded inputs) to verify malicious participants' messages but provides no explicit utility guarantees under arbitrary Byzantine attacks passing these checks. In contrast, CafCor addresses a more challenging scenario: adversaries crafting inputs that pass such verification yet harm model performance. Thus, cryptographic verification alone is insufficient for our Byzantine robustness objective. CafCor explicitly guarantees privacy and utility even under these stronger adversarial conditions (Corollary 4.1). We agree the privacy analysis in [1] may be tighter than [2], though both focus on decentralized topologies unlike our federated setting. We will highlight this explicitly.
>
> ###  **Communication Cost**
>
> > The communication cost is not discussed [...]
>
> Pairwise shared randomness (exchanging at most $n(n-1)$ integers) occurs offline once, incurring negligible quadratic communication compared to repeated training communications (model weights of size $d$ over many iterations). We will clarify this explicitly.
>
> ### **Additional Clarifications**
>
> > I am not really sure why Opacus is used to estimate the privacy budget [...]
>
> We use Opacus solely for tighter practical composition of per-round privacy budgets (using per-round theoretical guarantees of Theorem 4.1 with $T=1$). We will explicitly clarify this usage.

---

> > ### Comment · Reviewer_6PZT · 2025-04-07
> >
> > Dear Authors,
> >
> > Thank you for your clarifications. They were very helpful. Some of my concerns have been clarified. Here are some additional comments:
> >
> > - Privacy analysis: You say that "Our privacy analysis (Theorem 4.1) is distinct from [1,2], addressing a new threat model where Byzantine adversaries partially collude with the untrusted server. ". However, [1] *does* take into account adversarial parties that collude with the server. Therefore I am not sure what is the difference. I will revise the paper and take it into account in my final score.
> >
> > - Regarding the use of Opacus: I have no complain on the use of Opacus on the experiments as long as the used composition technique is cited in priority to the software.

---

> > > ### Author Response · Authors · 2025-04-07
> > >
> > > Thank you very much for your feedback. We hope that our clarifications address your concerns and that you will consider increasing your score.
> > >
> > > **Privacy Analysis Comment**
> > >
> > > Thank you for this helpful comment. We agree that [1] considers adversaries that can collude with the server, and we will update the paper to give [1] proper credit and clarify the positioning.
> > >
> > > Our setting includes a new aspect: we model not only honest users and malicious users who collude with the server, but also *malicious users who do not collude* with the server to violate privacy.
> > >
> > > A key implication is that when such non-colluding malicious users are present, honest users do not need to add independent noise for privacy. The correlated noise shares from these users act as a mask (see discussion following Theorem 4.1). This leads to improved empirical performance and, to our knowledge, was not addressed in prior work. Specifically, Figure 1 shows a significant performance gap between the case where all malicious users collude with the server (noted "ByzLDP" in the legend) and where none do (noted "SecLDP" in the legend). This difference is captured thanks to the privacy analysis in Theorem 4.1.
> > >
> > > While the privacy analysis in Theorem 4.1 is not our main contribution, we agree it is important to clarify. We will update the paper to make this distinction clearer and to acknowledge the contributions of [1] more explicitly.
> > >
> > > We also recall, as mentioned in the initial rebuttal, that [1] focuses on cryptographic verification (e.g., checking message correctness and bounded inputs), but does not offer explicit utility guarantees under general malicious behavior. In contrast, our work combines privacy guarantees with a novel robust aggregation to ensure utility even under stronger adversarial conditions.
> > >
> > > **Opacus Comment**
> > >
> > > Thank you for the suggestion. We confirm that our privacy accounting uses the RDP framework of Balle et al., “Hypothesis testing interpretations and Rényi differential privacy”, AISTATS 2020. We will cite this directly as the main reference for the composition method.

---

### Official Review · Reviewer_aCu9 · 2025-03-13

**Overall Recommendation:** 3

**Summary:**

This paper proposes a novel methodology to achieve resilience in the face of malicious parties colluding with an untrusted server in the distributed learning framework as well as privacy guarantees, based on a weak assumption that each pair of communicating workers secretly share a seed of randomness, used to inject correlated noise when aggregating gradients. As experiments show, the proposed method achieves stronger guarantees than methods based on local differential privacy and even approach those of centralized differential privacy, which do rely on an assumption of a fully trusted server. Empirical results validate.

**Claims And Evidence:**

The claims are supported theoretically and experimentally.

**Essential References Not Discussed:**

The claim that [Lamport 1982] labeled misbehaving workers in a certain way is false. That paper narrates a parable and applies a certain label to all workers (i.e., generals), not only misbehaving ones.

**Ethical Review Concerns:**

The paper continuously alludes to a prejudice that labels people of a certain cultural heritage as dishonest and untrustworthy.

**Ethical Review Flag:**

Flag this paper for an ethics review.

**Ethics Expertise Needed:**

["Discrimination / Bias / Fairness Concerns", "Other expertise"]

**Experimental Designs Or Analyses:**

The proposed method builds upon DSGD, and also compares against its plain version as a baseline.

**Methods And Evaluation Criteria:**

Methods and data are well-chosen.

**Other Comments Or Suggestions:**

NA

**Other Strengths And Weaknesses:**

NA

**Questions For Authors:**

Is there a reason for the main body of the paper to continuously invoke a cultural bias that labels people of a certain cultural heritage as dishonest or malicious?

**Relation To Broader Scientific Literature:**

First analysis of secret-based local differential privacy in adversarial distributed learning, considering an untrusted server and workers who aim to disrupt the learning as well as to compromise the privacy of honest workers by colluding with the server. Most of the works is based on previous works by Mironov and Allouah et al.

**Theoretical Claims:**

Theorems 4.1 and 4.2 seem sound.

---

> ### Author Rebuttal · Authors · 2025-03-31
>
> We clarify our intent and recall our contributions below following the reviewer's comments, and we welcome further discussion to refine our presentation.
>
>
> ### **Use of "Byzantine” Terminology**
>
> This is a standard term in distributed computing literature (Lamport et al., 1982), referring strictly to arbitrary malicious behavior, with no cultural implications. Nevertheless, we remain open to adopting alternative phrasing.
>
> We recall that our core contribution is a rigorous theoretical framework that significantly advances the resolution of the privacy–robustness–utility trilemma (Allouah et al. 2023b, ICML), with minimal trust assumptions. Theoretically, we approach the minimax-optimal privacy-utility trade-off achievable in the ideal scenario (trusted server and no attacks, see Corollary 4.1). Empirically, CafCor closely matches this optimal baseline even under state-of-the-art attacks (Figure 1).

---

> > ### Comment · Reviewer_aCu9 · 2025-04-01
> >
> > 1st comment:
> >
> > A term that is an exonym for a people or culture cannot be 'with no cultural implications'. This term was chosen because of its pejorative sense recorded in dictionaries, which reflects a historically ingrained cultural bias disparaging a group of people as devious, treacherous, and potentially malicious. If the goal is to refer to devious workers without invoking a cultural prejudice, how about calling them 'devious workers'?
> >
> > 2nd comment:
> >
> > The decision made here and the accompanying arguments about the contribution are appreciated.
> > Considering the cited references, it is worth highlighting the following key points:
> >
> > (A) [Lamport et al., 1982] was published at a time when there was less awareness around cultural stereotyping, and the review process did not include ethical considerations; besides, it does not endorse the use cultural terms for concepts like deviousness or dishonesty; it merely recounts a historical parable involving a group of generals.
> >
> > (B) [Blanchard et al., 2017] presents a troubling example of cultural bias, endorsing the use of a cultural term as a synonym for 'devious' and 'dishonest', and naming the proposed function after a Bulgarian khan who "undertook offensive attacks against the Byzantine empire" and is known for acts such as fashioning drinking vessels from, and toasting with, the skulls of defeated leaders [1]. Regardless of the historical accuracy of such accounts, which all derive from Greek sources, technical scientific terminology should not rest on specific interpretations of historical events and personalities. As historical understanding evolves, any given narrative may come to seem outdated and even embarrassing. In any case, the glorification of violence and the reinforcement of stereotypes distract from scientific objectives and contribute to an unhealthy narrative within the field.
> >
> > To foster an ethically responsible and inclusive research environment, we must critically reflect on the implications of such historical references and ensure that the language used in our field is technically precise and culturally sensitive. Unlike mythology, which invites creative interpretation, history demands careful, evidence-based refinement and bears directly on people’s identities. It is therefore inappropriate to single out any people as a symbol of deceit. Mythological figures are more fitting for such symbolic purposes. For instance, Norse mythology offers a useful term for describing something arbitrary, deceptive, and devious: Lokian.
> >
> > [1] E. N. Luttwak, The Grand Strategy of the Byzantine Empire, Harvard University Press, 2009.

---

> > > ### Author Response · Authors · 2025-04-03
> > >
> > > We appreciate your feedback regarding the use of the term “Byzantine.” Although this term has long been established in distributed computing and federated learning (e.g., Lamport et al., 1982; Blanchard et al., 2017), we understand your concerns.
> > >
> > > We have decided to adopt alternative phrasing in the revised manuscript (changing “Byzantine” everywhere to “adversarial”).
> > >
> > > We trust that this modification, alongside our theoretical and empirical contributions, addresses your concerns.

---

### Official Review · Reviewer_g8Bp · 2025-03-14

**Overall Recommendation:** 3

**Summary:**

The paper introduces an algorithm (CAFCOR) to achieve privacy and robustness in distributed learning without relying on a trusted central server. In particular, it employs correlated noise injection inspired by secret sharing and combines it with a robust aggregation technique to mitigate Byzantine workers' impact. The algorithm achieves near-central DP (CDP) utility by leveraging secret-based local differential privacy. Extensive experiments on MNIST and Fashion-MNIST datasets demonstrate CAFCOR's superior performance under various Byzantine attack scenarios compared to existing methods.

**Claims And Evidence:**

- The CAF aggregation method is shown to be resilient to a significant number of Byzantine workers.
- The experimental results support the theoretical claims, demonstrating improved accuracy compared to Local Differential Privacy (LDP) baselines.
- However, the experiments are limited to only MNISST and Fashion-MNIST.
- The explanation of how it practically handles high-dimensional data could be clarified.

**Essential References Not Discussed:**

The related works discussed are sufficient.

**Experimental Designs Or Analyses:**

The experimental design is limited.

**Methods And Evaluation Criteria:**

- The use of shared randomness and correlated noise injection can improve the privacy-utility trade-off.
- It is necessary to compare against more existing state-of-the-art methods beyond LDP and CDP.

**Other Comments Or Suggestions:**

NA

**Other Strengths And Weaknesses:**

### Strengths:
- Effective handling of Byzantine workers through robust aggregation.
- Empirical results support the theoretical findings.

### Weaknesses:
- Limited evaluation datasets.
- The complexity of CAF aggregation may hinder real-time deployment in large-scale systems.
- Lack of comparison with recent privacy-preserving FL methods like DP-FL or DP-SGD.

**Questions For Authors:**

1. How does CAFCOR perform on datasets with real-world heterogeneity and natural label noise?
2. What is the computational overhead of the CAF aggregation technique, particularly for large-scale models?
3. Can CAFCOR be integrated with gradient compression or sparsification techniques to improve communication efficiency?

**Relation To Broader Scientific Literature:**

The paper discusses related work in differential privacy, Byzantine robustness, and secret sharing.

**Theoretical Claims:**

The theoretical claims are thoroughly developed. However, to improve clarity, more intuitive explanations of the theoretical results should be included.

---

> ### Author Rebuttal · Authors · 2025-03-31
>
> We thank the reviewer for their insightful feedback. Below, we address the key points raised:
>
> ### **Experimental Scope**
>
> > Limited evaluation datasets.
>
> The reviewer’s suggestion to evaluate additional datasets is highly valid. We stress, however, that our current evaluation on MNIST and Fashion-MNIST is already extensive, rigorously testing our algorithm CafCor across multiple threat models, robust aggregation methods, and varying data heterogeneity levels (Section 5). Crucially, our setting is quite *challenging* even on these standard datasets, since we enforce **both** differential privacy without trusting the server and Byzantine resilience under state-of-the-art attacks. Indeed, as Figure 2 clearly illustrates, prior state-of-the-art methods suffer substantial accuracy losses under these combined constraints. Also, the SOTA method by Allouah et al. (2023b, ICML) could only scale to small logistic regression tasks with weaker theoretical guarantees, highlighting the novelty and practical significance of our results.
>
> We recall that our core contribution is a rigorous theoretical framework that significantly advances the resolution of the privacy–robustness–utility trilemma (Allouah et al. 2023b, ICML), with minimal trust assumptions. Theoretically, we approach the minimax-optimal privacy-utility trade-off achievable in the ideal scenario (trusted server, no Byzantine adversaries—Corollary 4.1). Empirically, CafCor closely matches this optimal baseline even under Byzantine threats (Figure 1). Extending to larger datasets such as ImageNet is an important and exciting future step, which we will explicitly mention.
>
> ### **Comparison with State-of-the-Art Methods**
>
> > Lack of comparison with recent privacy-preserving FL methods like DP-FL or DP-SGD.
>
> We believe there is a misunderstanding regarding our comparisons. In fact, we already benchmark CafCor against strong, theoretically optimal baselines in the paper: DP-SGD (pointed out by the reviewer) adapted for federated learning under local (LDP) and central (CDP) differential privacy. Figures 1 and 2, where "DSGD" denotes FL-adapted DP-SGD, demonstrate that CafCor outperforms LDP and closely approaches CDP (i.e., trusted server) utility. Moreover, in Section 5.2, we compare to numerous standard Byzantine-robust defenses. We will clarify this in the paper.
>
> ### **Computational Complexity**
>
> > The complexity of CAF aggregation may hinder real-time deployment in large-scale systems.
>
> We thank the reviewer for highlighting this important aspect. CafCor’s aggregation complexity, while higher than simple averaging, is significantly lower than previous state-of-the-art methods like SMEA, whose runtime complexity is exponential in the number of Byzantine workers $f$. Specifically, SMEA performs an exhaustive subset search across subsets of size $n-f$, making it computationally infeasible. In contrast, our covariance-based CAF aggregation with power-method approximation achieves an efficient runtime of $\mathcal{O}(f n d \log d)$, enabling scalability to high-dimensional models, far beyond the closest prior work (Allouah et al. 2023b), which only scaled to small logistic regression tasks.
> Finally, compression is feasible but beyond our current focus on privacy and robustness.
>
> We refer the reviewer to new experiments, included in our response to Reviewer 6PZT due to space constraints, that explicitly demonstrate this complexity advantage over SMEA, and will include these new experiments in the revision.

---

> > ### Comment · Reviewer_g8Bp · 2025-04-09
> >
> > I want to thank the authors for responding to my concerns with more explicit clarification and a new experiment. Hence, I decided to change my recommendation for this paper.

---

### Decision · Program_Chairs · 2025-05-01

**Decision:**

Accept (poster)

**Comment:**

This paper tackles the challenging trilemma of privacy, robustness, and utility in federated learning without assuming a trusted central server. Inspired by recent advances in the privacy literature (Sabater et al., 2022; Allouah et al., 2024), the authors slightly relaxes the local DP threat model to be able to leverage pairwise shared randomness for correlated noise injection. The paper also proposes CAF, a novel robust aggregation rule that is significantly more computationally efficient than the state-of-the-art SMEA. Theoretical results demonstrate that the proposed approach achieves near–central‑DP utility while bounding the influence of adversarial participants.

The author response and subsequent discussion clarified key concerns, through additional experiments and improved positioning relative to prior work. With these clarifications, the majority of reviewers now view the paper favorably. I recommend acceptance.

That said, I ask the authors to follow through on the revisions they committed to during the discussion phase, particularly:
However, I ask the authors to commit to making the revisions they promised during the discussion, in particular:
- Adding the new experimental comparisons;
- Updating the discussion of Sabater et al. (2022) to appropriately acknowledge their contributions to privacy analysis and the ability to handle malicious workers;
- Replacing "Byzantine" with "adversarial" throughout the manuscript to address cultural sensitivity concerns.